# Generalization over Memorization in In-Context Learning

## Abstract

Transformers exhibit remarkable in-context learning capabilities, solving new tasks without requiring explicit model weight updates. However, existing training paradigms for in-context learners rely on vast, unstructured datasets, which are costly to use in training and challenging to collect and analyze. Inspired by processes that drive human learning and motivated by these limitations, we propose a paradigm shift: training on multiple smaller, domain-specific datasets to improve generalization. We investigate this paradigm by leveraging meta-learning to train an in-context learner across diverse, small-scale datasets using the Meta-Album benchmark. We further investigate realistic scenarios, including domain streaming with curriculum learning strategies and settings where training data is entirely unlabeled. Our experiments demonstrate that this multi-dataset approach promotes broader generalization, enhances robustness in streaming scenarios, and achieves competitive performance even under unsupervised conditions.

## 1 Introduction

In-context learning (ICL) has emerged as a transformative paradigm in artificial intelligence, particularly with the development of large language models (LLMs). Unlike traditional machine learning approaches that rely on explicit weight updates or fine-tuning for adapting to new tasks, ICL enables models to generalize and solve tasks on the fly given only a few examples in the form of demonstrations (Brown et al., 2020). These demonstrations act as contextual information that helps the model infer the objective of a given task and make the right prediction without altering its internal parameters. This dynamic adaptability makes ICL a powerful framework for few-shot and even zero-shot learning, positioning it as a versatile tool for tackling diverse tasks in real time (Olsson et al., 2022).

Despite these advancements, the mechanisms underlying ICL remain an active area of investigation. Recent work has sought to draw connections between ICL and meta-learning (Min et al., 2022a; Kirsch et al., 2022; Fifty et al., 2024). Meta-learning approaches are explicitly trained to adapt to new tasks by leveraging previously learned knowledge and information extracted from a small set of data (context) (Vettoruzzo et al., 2024; Hospedales et al., 2021; Son et al., 2024; Finn et al., 2017). While meta-learning algorithms are explicitly trained for this purpose, e.g., by meta-learning a generalizable feature extractor as in Snell et al. (2017); Vinyals et al. (2016), in-context learners acquire this ability implicitly during the training phase (Akyürek et al., 2022), relying on large-scale datasets and large architectures to uncover patterns that enable generalization. Training such large models on vast, uncurated language corpora, such as the Common Crawl dataset (Raffel et al., 2020), is prohibitively expensive, lacks interpretability, and deviates significantly from how humans learn. Children are immersed in complex, unstructured environments from early on, but studies have shown that their learning is initially focused and repetitive (Farzin et al., 2010; Barrett, 1985) —they first observe a small set of familiar objects and faces before expanding to more diverse stimuli (Jayaraman et al., 2017; Clerkin et al., 2017). The same applies while learning to read, where the brain first learns to recognize and differentiate lines, curves, and strokes, before being able to generalize to single characters and entire words (Smith, 2024; Vong et al., 2024; Dehaene, 2010). This structured, incremental exposure supports the gradual development of generalization, mirroring how certain model training strategies may also benefit from domain-specific focus (Maurer et al., 2007). This highlights an important difference between human learning and traditional LLM training: while LLMs rely on vast amounts of unfiltered data to achieve

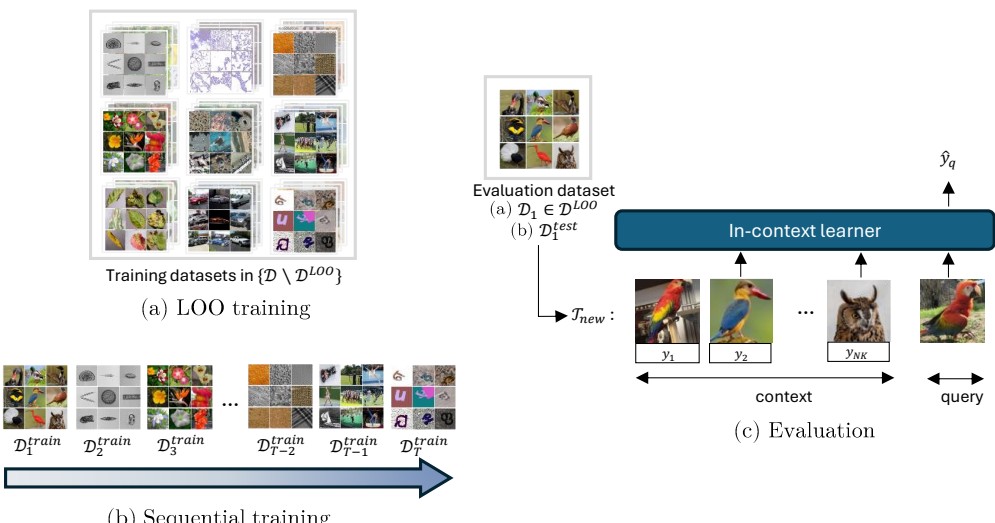

Figure 1: Overview of GEOM. The left side illustrates two training paradigms: (a) a leave-one-out (LOO) approach where the model is trained on all domains except one (e.g., Large Animals), and a dataset from the excluded domain is used for evaluation; and (b) a sequential approach, where datasets are introduced to the model in a sequential order and the model is evaluated on the test set of a previously seen dataset. The right side depicts the model evaluation process. A new task $\mathcal{T}_{new}$ is sampled from a dataset, either an entire dataset from $\mathcal{D}^{LOO}$ in (a) or the test split of a previously seen dataset in (b). This task is then organized into a non-causal sequence as described in Sect. 3. An in-context learner processes this sequence, using the context to infer and predict the query label.

generalization, a child can generalize with fewer, more meaningful examples (Smith, 2003). This suggests that the key to generalization may depend less on the sheer volume of data and more on its quality and the sequence in which it is presented (Bambach et al., 2018).

Motivated by these limitations, we propose an alternative perspective: ***training on multiple smaller, domain-specific datasets to foster generalization***. To investigate this, we analyze the performance of an in-context learner trained on visual tasks sampled from Meta-Album, a multi-domain meta-dataset designed specifically for few-shot image classification (Ullah et al., 2022). By evaluating performance across distinct visual domains, we can assess whether ICL possesses an intrinsic ability to generalize beyond its training domain. More specifically, we adopt a meta-learning approach to train a transformer model from scratch, reframing meta-learning as a sequence modeling problem. We organize tasks into non-causal sequences (Fifty et al., 2024; Vettoruzzo et al., 2025), where each instance is concatenated with its corresponding label to form the context, while query data is used for prediction. These sequences are fed into a transformer encoder, which processes the task context to predict the query label. By leveraging this formulation, we aim to train a model that favors **ge**neralization **o**ver **m**emorization, a capability we emphasize in the name of our approach: **GEOM**. Although this strategy does not *universally* outperform training on large, uncurated corpora, it can match — and in some cases exceed — the performance of large-scale pre-training, while offering additional benefits such as modularity, interpretability, and adaptability. Keeping datasets separate enables sequential training and simplifies the integration of new data as it becomes available. Similar approaches have been explored in recent work on language models (Xie et al., 2024; Chowdhery et al., 2023), showing that access to more balanced and curated web data can enhance the learning process.

Beyond investigating ICL with multiple domain-specific datasets, we further explore how GEOM can emulate human learning processes through two complementary strategies: sequential learning and unsupervised learning. Sequential learning mirrors the structured, incremental progression of human learning, where knowledge is acquired over time in an ordered manner (Sheybani et al., 2024b; Wang et al., 2024b). This paradigm introduces unique challenges related to dataset ordering such as how the order of datasets influ-

ences learning and the risk of forgetting earlier tasks (Lopez-Paz & Ranzato, 2017; Wang et al., 2024b). We employ curriculum learning strategies (Bengio et al., 2009; Soviany et al., 2022; Liu et al., 2024a) that organize datasets based on increasing levels of difficulty, either using a transfer learning (TL)-based approach (Faber et al., 2024) or optimal transport (OT) (Peyré et al., 2019; Chang et al., 2023; Alvarez-Melis & Fusi, 2020). These methods, together with our decision to use multiple, small datasets instead of having direct access to the whole knowledge, enable the model to adapt gradually, improving its generalization and study its resilience to forgetting. In addition, this confirms the generalization abilities of the model, as it acquires a general knowledge, rather than memorizing previously seen examples. An illustration of this variant, which we will refer to as **GEOM-S** (GEOM-*Sequential*), is presented in Fig. 1.

In contrast, unsupervised meta-learning reflects the human ability to derive meaningful patterns from raw, unlabeled experiences (Bambach et al., 2018). To explore this, we experiment with an unsupervised meta-learning approach, where tasks are generated through data augmentation and data mixtures, following the method proposed in Vettoruzzo et al. (2025). The resulting variant, denoted as **GEOM-U**, achieves remarkable generalization across tasks, further underscoring the benefits of leveraging small-scale datasets from diverse domains.

To summarize, our study (1) highlights the advantages of training on multiple small-scale, domain-specific datasets, emphasizing its practical relevance; (2) demonstrates that this approach fosters improved generalization compared to training on a single, large-scale dataset; (3) it proves even more effective in ordered sequential scenarios, achieving continuous improvement as additional datasets are introduced without catastrophic forgetting; (4) it showcases remarkable generalization across tasks, even in the absence of labeled data. In conclusion, by revisiting the training process of in-context learners, we propose an approach that draws inspiration from human learning processes, potentially bringing AI closer to natural and efficient learning.

The remainder of this paper is organized as follows. We provide an overview of the existing literature in Sect. 2 and we formally define the method and the datasets used in our experiments in Sect. 3 and Sect. 4, respectively. We then present the results across three different multi-domain scenarios: the supervised (offline) scenario in Sect. 5, the sequential scenario in Sect. 6, and the unsupervised scenario in Sect. 7. Finally, Sect. 8 concludes the paper and outlines potential directions for future work.

## 2 Related Work

**Meta-learning for in-context learning.** The term "in-context learning", introduced by Brown et al. (2020), describes the ability of LLMs to solve tasks based solely on contextual examples provided during inference, without requiring explicit weight updates or fine-tuning. Initially thought to be exclusive to large-scale language models (Radford et al., 2019; Hendrycks et al., 2020), thus trained on vast datasets (Raffel et al., 2020; Gao et al., 2020; Penedo et al., 2023), subsequent studies have shown similar behavior could be achieved also in smaller models (Schaeffer et al., 2023; Du et al., 2024), trained on more compact image datasets (Chan et al., 2022; Singh et al., 2023) like Omniglot (Lake et al., 2015). This capability has been compared with meta-learning, which explicitly trains models to adapt to new tasks by leveraging prior knowledge (Schmidhuber, 1987; Vettoruzzo et al., 2024; Hospedales et al., 2021). Unlike meta-learning, where task generalization is explicitly encouraged during training, ICL emerges implicitly during the pre-training stage. Recent studies have combined these paradigms by integrating meta-learning into ICL training, improving few-shot performance and model generalization (Santoro et al., 2016; Min et al., 2022a; Chen et al., 2022; Kirsch et al., 2022; Wang et al., 2024a; Fifty et al., 2024; Vettoruzzo et al., 2025). In particular, CAML (Fifty et al., 2024) and CAMeLU (Vettoruzzo et al., 2025) reframe meta-learning as a non-causal sequence modeling problem and demonstrate superior cross-domain performance, respectively in supervised and unsupervised settings.

**Multi-domain training paradigm.** The training paradigm in LLMs usually relies on unstructured, large-scale text corpora scraped from the entire web (Brown et al., 2020; Raffel et al., 2020; Gao et al., 2020; Penedo et al., 2023). However, the sheer scale and lack of curation in these datasets introduce challenges related to data quality, redundancy, and potential biases. To address these issues, recent efforts have focused

on improving dataset quality by weighting different data sources based on their quality (Chowdhery et al., 2023) or balancing model weights during training (Xie et al., 2024). These methods, though effective, diverge from learning by analogy typical of humans (Winston, 1980), where learning occurs through analogies across diverse domains—a concept tied to meta-learning principles. Multi-domain datasets (Triantafillou et al., 2020; Bornschein et al., 2024; Zhai et al., 2019; Koh et al., 2021) provide a structured way to emulate such processes facilitating the model adaptation and generalization to diverse tasks (Fifty et al., 2024; Vettoruzzo et al., 2025). However, these benchmarks are constrained to relatively similar domains or suffer from overlaps with commonly used datasets in transfer learning and meta-learning research. Meta-Album (Ullah et al., 2022) overcomes these limitations by offering a well-curated collection of datasets, systematically organized across ten distinct domains, with minimal overlap and balanced representation.

**Sequential learning.** Sequential learning, also called continual, lifelong, or streaming learning, represents a more human-like learning process, where concepts are introduced to a model sequentially, and each of them is available to the model only for a limited time before it progresses to the next (Wang et al., 2024b). A significant challenge in sequential learning is balancing two competing goals: ensuring robust generalization to future tasks by reusing prior knowledge and mitigating catastrophic forgetting of previously learned information (Lopez-Paz & Ranzato, 2017). To address these challenges, various methods have been proposed in the literature. These include memory-based methods (Buzzega et al., 2020; Rebuffi et al., 2017; Lopez-Paz & Ranzato, 2017), architectural-based methods (Sokar et al., 2021; Hemati et al., 2023; Kang et al., 2022), regularization-based methods (Kirkpatrick et al., 2017; Zenke et al., 2017), and meta-learning-based approaches (Gupta et al., 2020; Javed & White, 2019; Son et al., 2024; Irie et al., 2022; Lee et al., 2023). However, these strategies typically evaluate model performance using hand-crafted task streams, often derived by splitting a single dataset into subsets or applying manually designed data augmentations. Such synthetic streams fail to capture the complexity of real-world scenarios and suffer from issues such as poorly defined domain separation, arbitrary task orders, and an absence of structured progression. Curriculum learning offers a promising solution to these limitations by organizing tasks in a structured manner, typically based on increasing difficulty (Soviany et al., 2022; Bengio et al., 2009). Studies like Faber et al. (2024); Sheybani et al. (2024b); Chang et al. (2023); Liu et al. (2024b) propose various techniques for ordering datasets by complexity level.

## 3 Method

In this section, we begin by defining the concepts of meta-learning and ICL, highlighting their key differences in a comparative table (Tab. 1). As the former is well-known and extensively studied Schmidhuber (1987), the latter has been formally introduced in recent years Brown et al. (2020). To better align with the objectives of our study, firstly we revise both concepts to make their own purpose more evident; secondly, we describe GEOM as a meta-trained in-context learner specifically designed to adapt to diverse tasks by leveraging context examples during inference, and outline the training details used in our experiments.

### 3.1 Definitions

Meta-learning, often referred to as "learning-to-learn" explicitly utilizes the task's context (also referred to as *support set* in meta-learning) in a structured and well-defined manner. It explicitly encodes how the context is leveraged, typically through a dedicated adaptation step. This step systematically adapts the model to the task by enforcing specific algorithms for utilizing and "learning from" the context information.

In-context learning (ICL), on the other hand, involves providing the task context as part of the input (e.g., concatenated context examples and queries). However, it does not explicitly define or enforce how the context should be used to learn from it and produce task-specific outputs. Instead, the model exploits the broad and diverse knowledge accumulated during the training phase and only leverages the attention mechanism during inference.

Therefore, although both ICL and meta-learning utilize demonstration contexts for task adaptation, they differ fundamentally in their approach (see Tab. 1). ICL arises implicitly during the pre-training phase of attention-based models, requiring no additional design to enable adaptation, while meta-learning is a strategy

Table 1: Differences between in-context learning (ICL) and meta-learning.

| Aspect | In-context learning | Meta-learning |
|---|---|---|
| Training data | Trained on vast datasets, often leading to broad generalization. | Relies on tasks sampled from meta-datasets for simulating adaptation. |
| Training objective | Emerges implicitly from standard objectives (e.g., next-token prediction or classification). | Explicitly optimized for task adaptation during meta-training. |
| Adaptation process | Adapts during inference solely through task context; no parameter updates required. | May require task-specific adaptation (e.g., gradient updates) during inference. |
| Generalization | Relies on patterns learned during pre-training. | Optimized to generalize quickly across tasks. |
| Applications | Commonly used in LLMs. | Widely applied in scenarios requiring rapid task adaptation (e.g., robotics, reinforcement learning, few-shot classification). |

aimed at designing models that rapidly adapt to new tasks or domains through explicit task conditioning and optimization. Given these distinctions, our approach meta-learns an in-context learner to combine the learning to learn strategy typical of meta-learning with the implicit task inference and generalization capabilities of ICL, resulting in a flexible yet systematic framework for generalization across diverse tasks.

### 3.2 GEOM

In this section, we provide a general overview of GEOM. The architecture will be further expanded in the coming sections to fit the specific setting. Specifically, in Sect. 5, GEOM is trained with a leave-one-out (LOO) approach, where one domain is excluded from the training pipeline, to evaluate cross-domain generalization. In Sect. 6, sequential training is performed on the training split of each dataset, and performance is evaluated on the test split. Finally, Sect. 7 discusses an unsupervised scenario where no labels are available during training. An illustration of our approach both in the LOO and sequential setting is presented in Fig. 1.

We formalize the general pipeline for GEOM by following the same principle of several ICL methods (Brown et al., 2020; Kirsch et al., 2022; Chan et al., 2022) and inheriting the non-causal nature of the transformer encoder as in Fifty et al. (2024) and Vettoruzzo et al. (2025). Let $\mathcal{D} = \{\mathcal{D}_a \mid a = 1, \ldots, A\}$ be the set of all available datasets containing image-label pairs. Following the common rationale of meta-learning, we split each dataset into two parts $\mathcal{D}_a = \{\mathcal{D}_a^{train}, \mathcal{D}_a^{test}\}$ such that the classes in the training set do not overlap with those in the test set, i.e., $\{y^{train}\} \cap \{y^{test}\} = \emptyset$. At training time, we sample a task $\mathcal{T}_i$ from a randomly chosen dataset $\mathcal{D}_a^{train}$. Each task corresponds to a data generating distribution $\mathcal{T}_i \triangleq \{p_i(x), p_i(y|x)\}$ and consists of data from $N$ distinct classes. We reserve a small number of $K$ labeled examples per class to form the *task context* or demonstrations, while the remaining $Q$ examples are used as *queries* to evaluate the predictions. As a result, for each task, we construct $Q$ sequences as the concatenation of the full context and a single unlabeled query $x_q$. This sequence is defined as follows:

$$S_{i,q} = ((x_1, y_1), \ldots, (x_{NK}, y_{NK}), x_q) \quad q = 1, \ldots, Q, \tag{1}$$

where $NK$ is the total number of context examples. It is worth noting that this sequence is permutation invariant, or *non-causal*, as the order of context examples does not affect the query classification. This property is inherent in visual meta-learners (Fifty et al., 2024; Garnelo et al., 2018; Müller et al., 2022) and differs from the causal sequence model typical of LLMs.

To enable the model to learn from these non-causal sequences, GEOM consists of three components: (1) a feature extractor $f_\psi$ that maps each image into an embedding space; (2) a single-layer linear class encoder $g_\phi$ that maps the value of each label $y_k \in \{1, \ldots, N\}$ to a high-dimensional space; and (3) a non-causal transformer encoder $M_\theta$ with a classification layer on top that performs the classification. In particular, each sequence is formed by concatenating the output of the feature extractor for each image with its corresponding encoded label. Since the class of the query image is unknown, a randomly initialized learnable vector is appended to each query representation. This results in the following sequence $S_{i,q} = ((f_\psi(x_1), g_\phi(y_1)), \ldots, (f_\psi(x_{NK}), g_\phi(y_{NK})), f_\psi(x_q)), q = 1, \ldots, Q$, which resembles the format in Eq. 1. This sequence is fed into the transformer encoder, and only the output corresponding to the query sample is selected and passed through a classification layer to predict the query label. This process iterates for all queries in the task, and the aggregated loss is employed for model training. The resulting training objective is formulated as follows:

$$\min_{\theta, \phi} \ \mathbb{E}_{S_i} \left[ \frac{1}{Q} \sum_{q=1}^{Q} \mathcal{L}(M_\theta(S_{i,q}), y_q) \right] \tag{2}$$

where $S_i = \{S_{i,q}\}_{q=1}^{Q}$ represents the set of sequences associated to each task $\mathcal{T}_i \sim \mathcal{D}_a^{train}$, $\mathcal{L}$ is the cross-entropy loss function, and $y_q \in \{1, \ldots, N\}$ is the true label of the query $x_q$ within the context window.

During evaluation, a new task $\mathcal{T}_{new}$ with $N$ classes is sampled from a dataset $\mathcal{D}_a^{test}$ (with $a \in \{1, \ldots, A\}$), and the task context, consisting of $K$ labeled examples per class, is used to guide the classification of each query sample into one of the $N$ classes.

### 3.3 Training details

For all our experiments, we build each training episode as an $N$-way $K$-shot classification task, where $N$ and $K$ are fixed to 5. Following the same model architecture as in Vettoruzzo et al. (2025), we use a ResNet-50 (He et al., 2016) feature extractor $f_\psi$ pre-trained on ImageNet-1k and a class encoder $g_\phi$ consisting of a single learnable layer that maps the $N$ class labels to a dimensionality of 256. The non-causal transformer consists of eight encoder layers, each incorporating a multi-head self-attention block with eight attention heads, an MLP, and a single-layer classifier that maps the transformer output to the predicted category. The episodic training is performed for $300\,000$ iterations with Adam optimizer, an initial learning rate set at $10^{-5}$, and a warmup cosine scheduler. For future evaluation, the best-performing model is saved as the one resulting in the highest validation accuracy across $50\,000$ new tasks, sampled from $\mathcal{D}_a^{test}, a = 1, \ldots, A$. The code is written in Python and the experiments are run on an NVIDIA GeForce RTX 3070 Ti Laptop GPU and on an NVIDIA A100-SXM4 GPU with 40GB of VRAM, to speed up the execution. More details about the training settings can be found in Appendix A.2, while the code will be released upon acceptance of the paper.

## 4 Dataset

Meta-Album (Ullah et al., 2022) serves as the primary benchmark for this study. Although this approach could be expanded to other collections, we chose Meta-Album as it offers a diverse and comprehensive suite of datasets tailored for few-shot learning, transfer learning, and meta-learning research, in addition to its well-curated nature, wide range of domains included and balance across datasets. It includes 30 image classification datasets (as of writing), spanning ten distinct domains. Each domain comprises three datasets made available in three successive **releases**, as outlined in Tab. 2. The datasets are uniformly preprocessed and are available in three **sizes** (Micro, Mini, and Extended) to accommodate varying computational requirements. For our experiments, we

Table 2: Dataset IDs in Meta-Album Mini.

| Domain name | First release | Second release | Third release |
|---|---|---|---|
| Large Animals | 44285 | 44298 | 44305 |
| Small Animals | 44282 | 44292 | 44306 |
| Plants | 44283 | 44293 | 44302 |
| Plant Diseases | 44286 | 44299 | 44303 |
| Microscopy | 44281 | 44297 | 44308 |
| Remote Sensing | 44290 | 44300 | 44307 |
| Vehicles | 44289 | 44295 | 44309 |
| Manufacturing | 44288 | 44294 | 44304 |
| Human Actions | 44284 | 44291 | 44301 |
| OCR | 44287 | 44296 | 44310 |

primarily focus on the Mini size, which includes all original classes from the 30 datasets (up to 706 classes per dataset), and 40 examples per class. We refer to the datasets by their dataset IDs, detailed in Tab. 2, unless otherwise stated.

Since ImageNet-1k (Deng et al., 2009) has been widely used when pre-training model backbones for visual recognition and identification tasks, it is crucial to assess the potential overlap between Meta-Album and ImageNet-1k. Such overlap could lead to data leakage, where models trained on ImageNet-1k may inadvertently benefit from prior exposure to similar data, resulting in enhanced performance on Meta-Album. To ensure a fair evaluation, we perform an analysis to identify any overlaps, both in terms of class names and underlying concepts, between Meta-Album and ImageNet-1k. We use two complementary approaches for this investigation:

1. **Label matching**: Class names in Meta-Album and ImageNet-1k are compared by identifying matching words. A pre-processing step is applied to remove special characters and convert all names to lowercase, ensuring consistency in the comparison.

2. **Concept similarity**: Using CLIP (Radford et al., 2021) embeddings, we calculate cosine similarity scores between Meta-Album and ImageNet-1k labels to identify overlapping concepts. Scores above a certain threshold are considered indicative of overlap. The threshold is computed considering the distribution of cosine similarity values for each dataset, identifying the $90^{th}$ percentile of the distribution, and calculating the median value across all datasets. The resulting global threshold is set to 0.83. Fig. 3 illustrates the cosine similarity distributions for all datasets.

Three domains—Small Animals, Microscopy, and OCR—are excluded from the concept similarity analysis due to their unique characteristics and label formats, which make a direct comparison with ImageNet-1k impractical. Specifically, Microscopy and OCR feature concepts differ significantly from those in natural images (as in ImageNet-1k), while Small Animals, with its reliance on Latin names, introduces ambiguity and confusion in the matching process, leading to unreliable results. The results, illustrated in Fig. 2, reveal a substantial degree of similarity, exceeding 50%, for the Large Animals datasets (with dataset IDs 44285, 44289, 44305). Significant similarities with ImageNet-1k are identified also in the Remote Sensing and Human Actions domains, highlighting the possibility of data leakage when models pre-trained on ImageNet-1k are evaluated on these datasets. More details about this analysis and the other datasets used in this work are described in Appendix A.2.

## 5 Supervised (offline) learning

In this section, we investigate whether training on multiple small-scale datasets across diverse domains can improve model generalization when tested on an entirely different domain. This setting offers practical advantages as small datasets are easy to curate, update, and maintain allowing individual datasets to be replaced or excluded without disrupting the overall training pipeline. This modular approach ensures flexibility in handling potentially biased or outdated data (Bourtoule et al., 2021; Menon et al., 2020), making it easier to refine and adapt the dataset composition over time. To address this question, we consider a standard supervised learning scenario where all training data are accessible at the start of the training phase, and evaluation is performed cross-domain, on a domain excluded from training. We adopt a LOO approach, where datasets from nine randomly selected domains are used for training, while the remaining domain is reserved for evaluation. Specifically, we define the evaluation datasets as $\mathcal{D}^{\text{LOO}} = \{\mathcal{D}_l^{\text{LOO}} \mid l = 1, 2, 3\}$, representing the three datasets from the left-out domain and the training datasets as $\{\mathcal{D}\backslash\mathcal{D}^{\text{LOO}}\}$, which include all other datasets. As the focus here is on cross-domain evaluation, datasets are not split into $\mathcal{D}_a^{train}$ and $\mathcal{D}_a^{test}$, but all data are used during meta-training if they belong to $\{\mathcal{D}\backslash\mathcal{D}^{LOO}\}$, or during evaluation if they are part of $\mathcal{D}^{LOO}$. Depending on the baseline used, tasks may consist of examples from a single dataset or a mixture of datasets, as described in the subsequent section. All other methodological aspects align with those described in Sect. 3.

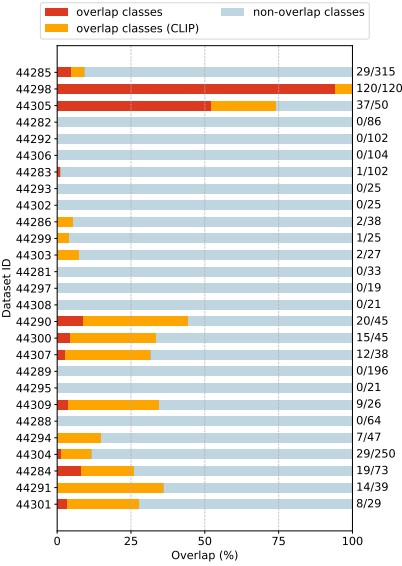

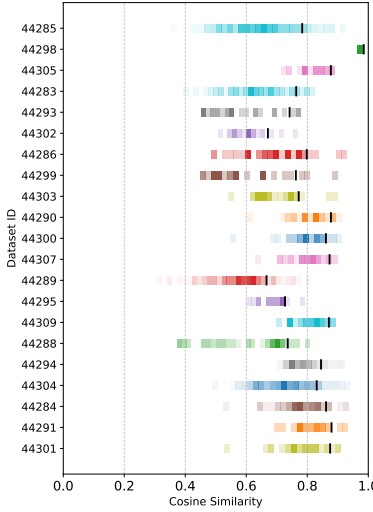

Figure 2: Class overlap between ImageNet-1k and Meta-Album Mini datasets. The red color shows the exact label matching analysis and the orange color indicates the result of the concept similarity analysis computed with CLIP embeddings (Radford et al., 2021). On the right side, we report the number of overlapping classes.

Figure 3: Cosine similarity distribution between CLIP (Radford et al., 2021) embeddings of ImageNet-1k labels and the Meta-Album labels that have no exact match. Horizontal bars represent the $90^{th}$ percentile of similarity values for each dataset. Datasets from the Small Animals, Microscopy, and OCR domains are excluded from the analysis.

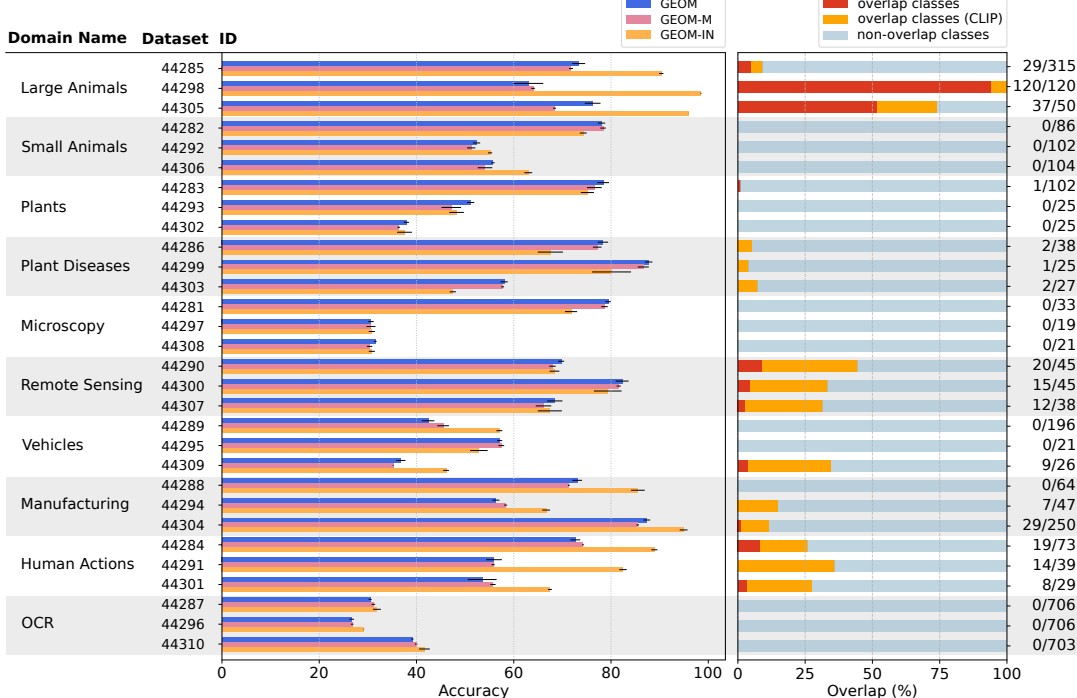

Figure 4: (Left) Accuracy comparison between GEOM, GEOM-M, and GEOM-IN for all the Meta-Album datasets. The training is performed using the LOO approach detailed in Sect. 5, and the performance is evaluated on the datasets from the left-out domain. (Right) Corresponding class overlapping between ImageNet-1k and Meta-Album as shown in Fig. 2.

### 5.1 Multi-dataset training

Building on the cross-domain LOO scenario described earlier, we evaluate the generalization performance of three distinct baselines. The goal of this section is to determine whether training on multiple, distinct small-scale datasets from different domains provides greater benefits for model generalization than relying on a single large-scale dataset. The baselines analyzed are as follows:

- **GEOM**: each Meta-Album dataset is treated as a distinct entity, and each training task consists exclusively of images sampled from a single dataset.

- **GEOM-M (GEOM-*Merged*)**: all Meta-Album datasets are combined to resemble a large-scale dataset, where each training task can include samples from multiple datasets and domains.

- **GEOM-IN (GEOM-*ImageNet-1k*)**: training tasks are sampled from ImageNet-1k (Deng et al., 2009), a large-scale benchmark widely used in computer vision.

Both GEOM and GEOM-M are trained across ten distinct combinations of Meta-Album domains, ensuring all possible LOO scenarios are covered. The performance for all baselines is evaluated on the left-out domain, and the results are summarized in Fig. 4. Overall, GEOM performs comparably or even better than GEOM-M across the Meta-Album benchmark, although tasks in GEOM can be considered more challenging as they usually involve a fine-grained classification. One possible explanation for this is the burstiness of GEOM's training data (Chan et al., 2022; Singh et al., 2023; Zhao et al., 2024; Chan et al., 2025). In GEOM, tasks are sampled within domain-specific datasets, leading to naturally clustered, or bursty, task distributions. In contrast, tasks in GEOM-M are constructed by uniformly sampling from the merged Meta-Album datasets, resulting in "less bursty" distribution. While the differences in performance are not always substantial, GEOM offers several advantages, including improved modularity and adaptability to new domains without requiring a large-scale, merged dataset. This highlights the benefit of preserving domain-specific boundaries during training, rather than merging datasets into a single corpus. These findings contrast with the common training paradigm for LLMs, where massive, unstructured datasets, often combining text from a wide variety of domains, are leveraged to improve generalization (Brown et al., 2020). Our results suggest that focusing on domain-specific training can yield comparable or improved cross-domain generalization, providing additional benefits. This aligns with human learning, which often benefits from focused task-level learning before generalizing across domains (O'hearn, 2005; Feldman, 2003; Gagné, 1985). Additional evidences supporting this principle is presented in Tab. 12 in Appendix A.6, where results are reported for the 5-way 1-shot scenario and in Sect. 6.3, where structured curricula further improve performance and generalization.

To further investigate the generalization capabilities of GEOM, we visualize the relative validation accuracy curve in Fig. 18 in Appendix A.9. Consistently with the findings of Kirsch et al. (2022) and Vettoruzzo et al. (2025) three learning phases can be recognized during the model learning. In the memorization phase the model simply memorizes the training data and the cross-domain accuracy does not improve, the learning-to-learn phase, where the model learns to solve new, unseen tasks, and the final generalization phase where the model generalize out of distribution. These finding supports our hypothesis that GEOM leverage generalization over memorization for cross-domain predictions.

GEOM-IN is included primarily as a reference point, due to the widespread use of ImageNet-1k in the vision community. When comparing GEOM to GEOM-IN, GEOM achieves superior or comparable performance in datasets with minimal class overlap between Meta-Album and ImageNet-1k. In domains with significant class overlap, such as Large Animals and Human Actions, GEOM-IN benefits from the knowledge acquired during training, relying on memorization rather than true generalization. However, in domains like Remote Sensing, where a notable overlap with ImageNet-1k exists but is accompanied by a significant distribution shift (e.g., images acquired through a GPS system vs. a normal camera), GEOM-IN struggles to adapt to these differences and to match GEOM's performance. This suggests that memorization alone may not be sufficient when concepts are represented through significantly different modalities or contexts. Another domain where GEOM-IN prevails over GEOM is Manufacturing. This behavior can be attributed to the reliance of its datasets on low-level features for classification, which are better captured by the large-scale ImageNet-1k (1 281 167 images) compared to the smaller Meta-Album Mini collection (163 200 images). This assumption

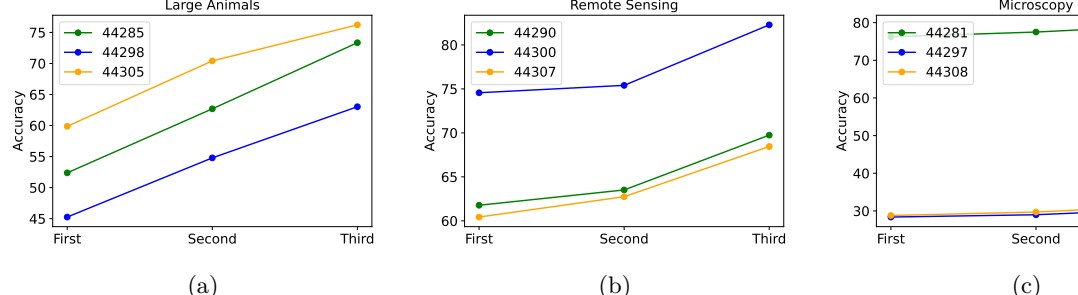

Figure 5: Comparison of GEOM training only on datasets from the first release (*First*, 9 datasets), on datasets from the first and second releases (*Second*, 18 datasets), and on datasets from all three releases (*Third*, 27 datasets) of Meta-Album Mini. The training is performed following the LOO setting described in Sect. 5, and the performance is evaluated on the datasets from the left-out domain (represented with blue, orange, and green colors). Results are reported only for three exemplary scenarios, while the complete set of results can be found in Fig. 17 (Appendix A.6). In particular, (a) and (b) show increased generalization as more out-of-domain datasets are added to the training pipeline, while (c) shows a modest performance improvement due to its reliance on low-level features.

is further corroborated by results obtained with the Extended size of Meta-Album (1 384 616 images), where GEOM performance in the Manufacturing domain improves significantly. As shown in Tab. 9 in Appendix A.6, accuracy increases by 26.1%, 9.4%, and 10.9% for the three datasets in the Manufacturing domain. To further confirm that the results of GEOM vs GEOM-IN, in particular that the performance of GEOM-IN are not influenced by the frozen feature extractor pre-trained on ImageNet-1k, we replace ResNet50 with CLIP (Radford et al., 2021). Tab. 11 in Appendix A.6 evidences comparable relative performance between GEOM and GEOM-IN when a different feature extractor is used. Finally, we evaluate the performance of GEOM vs. GEOM-M when the test tasks are created following the task creation of GEOM-M (Tab. 13): despite GEOM has never experienced tasks that contain classes from mixed domains, the advantage of GEOM-M is still negligible and, surprisingly, the overall highest result is achieved by GEOM when trained excluding OCR. Since this domain is much larger than all the others, it may introduce a significant bias in the final performance.

For detailed accuracy results of Fig. 4, please refer to Tab. 10 in Appendix A.6.

## 5.2 Impact of number of datasets

To investigate whether the generalization ability of the model improves progressively with the number of datasets used during training, we evaluate three distinct scenarios: training exclusively on datasets from the first release, on datasets from the first and second releases, and on datasets from all three releases of Meta-Album. These configurations allow us to examine the relationship between generalization and knowledge accumulation, drawing parallels with the progressive learning process observed in humans (Sheybani et al., 2024b). We refer to these three scenarios as *First*, *Second*, and *Third*, highlighting the usage of all datasets available up to a certain release. In line with the LOO setting described in Sect. 5, training is conducted on datasets spanning nine domains, with evaluations performed cross-domain on the left-out domain. As illustrated in Fig. 5, and in Fig. 17 in Appendix A.6, incorporating additional datasets consistently enhances generalization across all domains. This improvement can be attributed to the increased variability of training tasks, which has been shown to promote robust learning (Chan et al., 2022; Singh et al., 2023; Wang et al., 2024a; Raparthy et al., 2024; Raventós et al., 2023; Panwar et al., 2024). However, such improvement varies across domains. For instance, in Microscopy, Manufacturing, and OCR, the performance gains remain relatively modest compared to other domains. We conjecture that this is due to the reliance of these domains on simple, low-level features, which benefit more from an increased number of images per class, rather than the increased diversity that a higher number of classes introduces. In contrast, domains characterized by greater complexity benefit significantly from the inclusion of additional datasets, as the broader diversity

Table 3: Results using the three sizes of Meta-Album: Micro, Mini, Extended. The training is performed following the setting described in Sect. 5, with all Meta-Album domains, but OCR, included in the training phase. The performance is then evaluated on datasets that do not belong to the Meta-Album benchmark, such as CIFAR-fs (Bertinetto et al., 2019), CUB (Wah et al., 2011), Aircraft (Maji et al., 2013), Meta-iNat (Wertheimer & Hariharan, 2019), EuroSat (Helber et al., 2018), and ISIC (Codella et al., 2018). GEOM-IN is trained using ImageNet-1k. Results show the average across three complete runs of the algorithms.

|                  | CIFAR-fs         | CUB              | Aircraft         | Meta-iNat        | EuroSat          | ISIC             |
|------------------|------------------|------------------|------------------|------------------|------------------|------------------|
| GEOM (Micro)     | $60.47 \pm 4.98$ | $62.17 \pm 2.51$ | $29.26 \pm 0.62$ | $58.38 \pm 6.39$ | $63.70 \pm 1.20$ | $25.69 \pm 1.93$ |
| GEOM (Mini)      | $79.01 \pm 0.95$ | $88.94 \pm 0.70$ | $39.73 \pm 1.32$ | $74.10 \pm 0.12$ | $78.40 \pm 0.84$ | $31.38 \pm 1.33$ |
| GEOM (Extended)  | $76.25 \pm 1.03$ | $90.39 \pm 0.30$ | $40.88 \pm 0.84$ | $75.15 \pm 0.28$ | $79.31 \pm 0.82$ | $31.70 \pm 0.56$ |
| GEOM-IN          | $85.27 \pm 1.08$ | $79.64 \pm 1.01$ | $38.24 \pm 1.20$ | $76.10 \pm 0.32$ | $56.70 \pm 2.32$ | $27.90 \pm 1.41$ |

helps the model generalize to unseen data more effectively. These findings raise an important question of whether this improvement is driven by the increased number of images or by the broader representation of classes, a question explored in detail in the next section. The numerical evidence of these experiments can be found in Tab. 14 in Appendix A.6.

### 5.3 Number of classes vs. number of images

To better understand the factors driving the improved performance of GEOM as more datasets are included during training, we analyze whether the key determinant is an increase in the number of classes or the number of images in the training set. Previous research (Singh et al., 2023; Chan et al., 2022) suggests that increasing the number of classes plays a more significant role in enhancing the generalization capabilities of in-context learners than simply increasing the total number of images. However, these studies are often limited to in-domain settings, and especially restricted to training and test tasks that are both drawn from the same dataset (specifically, Omniglot (Lake et al., 2015)). Our work seeks to validate and extend these claims to a more challenging cross-domain setting. To achieve this, we considered three different versions of Meta-Album with varying sizes: Micro, Mini, and Extended. Since Extended does not include the OCR domain, we remove the three datasets associated with OCR also in Micro and Mini. We then evaluate the model on external datasets outside the Meta-Album benchmark, such as CIFAR-fs (Bertinetto et al., 2019), CUB (Wah et al., 2011), Aircraft (Maji et al., 2013), Meta-iNat (Wertheimer & Hariharan, 2019), EuroSat (Helber et al., 2018), and ISIC (Codella et al., 2018). This allows us to train the model following the same approach described in Sect. 5, but incorporating all datasets from the nine Meta-Album domains, after excluding OCR. The main differences between the three Meta-Album sizes are that Micro and Mini have the same number of images per class, but the number of classes per domain in Mini can be significantly higher than the 20 classes used in Micro. The Extended size, instead, has the same number of classes as Mini when removing the OCR dataset, but the number of images per class may notably increase for some domains. From Tab. 3, we observe that the larger performance improvement occurs when moving from the Micro to the Mini size of Meta-Album, compared to moving from the Mini to the Extended size. These results suggest that the most significant performance improvements arise

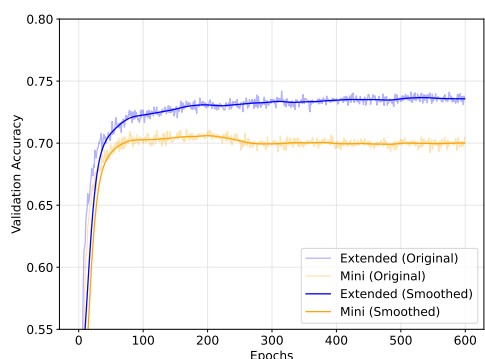

Figure 6: Validation performance of GEOM while trained on the Mini and Extended size of Meta-Album. The Mini size achieves peak performance early but declines due to overfitting, while the Extended size shows steady improvement over longer training periods, indicating the impact of increased image quantities in mitigating overfitting. The validation accuracy at each epoch is calculated on 50 tasks per dataset (1500 tasks in total) and both the original (shaded) and the smoothed (saturated) curves are represented.

from increasing the number of classes, which enriches task vari-
ability and broadens the model's capacity for generalization. On the other hand, the substantial increase in
the number of images in the Extended size does not yield a proportional performance boost, highlighting the
greater importance of class diversity compared to an increase in the number of images per class. This con-
clusion is further supported by the performance comparison between GEOM and GEOM-IN. Despite having
access to a consistently high number of images per class, GEOM-IN does not achieve the same performance
as GEOM (Mini). Even in datasets like CIFAR-fs (Bertinetto et al., 2019) and Meta-iNat (Wertheimer &
Hariharan, 2019), where we expect higher performance for GEOM-IN due to the presence of significant over-
lap with ImageNet-1k classes, GEOM-IN exhibits performance that is only comparable with GEOM (Mini)
and GEOM (Extended). While class diversity emerges as the dominant factor, the dataset size, i.e., the
total number of images, plays a non-negligible role. In the case of Micro, an insufficient number of images
leads to high variance in performance (see Tab. 3). In addition, when comparing the validation accuracy of
Mini and Extended, as in Fig. 6, GEOM on Mini achieves a peak validation accuracy within 200 epochs but
subsequently declines, likely due to overfitting. A possible explanation is that after the model has explored
all possible combinations of the training data, it starts memorizing specific examples rather than learning
generalizable patterns, which may reduce its ability to generalize to unseen classes. Conversely, training on
Extended, which contains approximately five times the number of images in Mini, requires a longer time to
converge but continues improving. These findings lead to two considerations: while longer training times for
a given dataset size may not always enhance performance, a sequential scenario, where datasets and classes
evolve over time, can result in significant performance gains. This is explored further in Sect. 6.

## 6 Sequential learning

In this section, we investigate a more realistic scenario where datasets are presented to the model sequentially
as a stream of tasks rather than being available all at once during training. Following the task definition in
Sect. 3, each dataset $\mathcal{D}_a \in \mathcal{D}$ is divided into $\mathcal{D}_a^{train}$ and $\mathcal{D}_a^{test}$, ensuring no class overlap between the two sets.
During training, each dataset is available for a fixed duration (measured in training epochs), and tasks are
sampled from it proportionally to the allocated time. Once the allocated time elapses, the stream advances to
the next dataset, and previously seen data becomes inaccessible. Importantly, we do not incorporate model
rehearsal techniques (Buzzega et al., 2020; Rebuffi et al., 2017; Gupta et al., 2020), requiring GEOM to rely
solely on its meta-learned knowledge to generalize effectively to new tasks that may involve both previously
seen and novel concepts. To distinguish this scenario from the supervised (offline) setting (Sect. 5), where
all the datasets are available simultaneously during training, we refer to the sequential model as **GEOM-S**
(GEOM-*Sequential*). We define this scenario as "sequential" to highlight the progression of training datasets
ordered with some specific heuristic, e.g., with a domain-based order or with an increasing complexity. This
terminology reflects a key distinction from traditional continual learning approaches. Our method does not
involve training until convergence on each dataset before advancing to the next, and it aims to evaluate the
model on completely new tasks with different classes from those observed during training. This scenario
aligns more closely with the meta-learning literature and the human learning process.

More formally, at time $T$, with $T \leq A$, the model has observed the datasets $\mathcal{D}_1^{train}, \ldots, \mathcal{D}_T^{train}$, possibly
corresponding to different domains. The evaluation is performed by sampling new, unseen tasks $\mathcal{T}_{new} \sim$
$\mathcal{D}_t^{test}, t < T$ from datasets observed earlier in the sequence to assess performance on previously encountered
domains. To better manage computational resources, GEOM-S is evaluated only at the end of the training
stream, after all datasets in the Meta-Album Mini benchmark have been processed sequentially. Additionally,
we investigate the model's ability to retain knowledge by measuring catastrophic forgetting on previously
seen domains (Sect. 6.2).

An important consideration in the sequential paradigm is the order in which datasets are presented. One
straightforward approach is to organize the datasets in domains and present a sequence of domains to the
model. This ensures a gradual shift in concepts, as each domain comprises three related datasets, and it is
evaluated in Sect. 6.1. However, this method does not account for the progressive structuring of information,
which can facilitate more effective learning (Sheybani et al., 2024b). To explore alternative dataset ordering,
we evaluate curriculum-based approaches (Bengio et al., 2009; Soviany et al., 2022). These include a TL-
based curriculum (Faber et al., 2024), which balances similarity and difficulty in the dataset presentation to

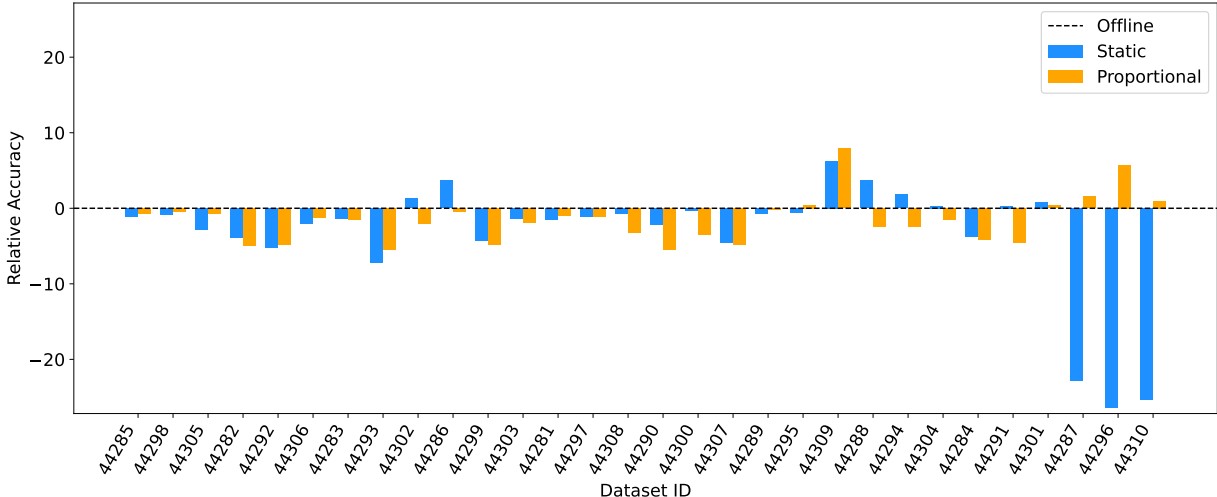

Figure 7: Relative performance of GEOM-S using a *static* and *proportional* approach for assigning training epochs to each dataset compared to the *offline* baseline, where all the datasets are available simultaneously. The relative accuracy is calculated as the difference between the accuracy achieved with the *static* (*proportional*) approach and the *offline* baseline, which is set as the reference point at zero.

create a structured learning path, and an OT-based curriculum (Alvarez-Melis & Fusi, 2020; Chang et al., 2023), where datasets are ordered based on their relevance to previously acquired knowledge. These strategies are detailed in Sect.6.3.1 and Sect.6.3.2, respectively.

### 6.1 Domain-based sequential scenario

To begin, we evaluate GEOM-S in a domain-based sequential scenario, where datasets are ordered according to their respective domains as defined in the Meta-Album benchmark: Large Animals, Small Animals, Plants, Plant Diseases, Microscopy, Remote Sensing, Vehicles, Manufacturing, Human Actions, OCR. Given the difference in dataset sizes across these domains, we evaluate the performance of GEOM-S using two approaches. In the *static* approach each dataset is assigned an equal number of training epochs (20), irrespective of its size, while in the *proportional* approach, the number of training epochs is allocated in proportion to the size of each dataset. Additionally, the results are compared with an *offline* baseline, similar to GEOM, where all the datasets are simultaneously available during training. This baseline, considered as an oracle, represents an idealized scenario where all the available knowledge is present upfront. While less realistic, it helps establish an upper bound for model performance when data accessibility is unconstrained. Importantly, this baseline is trained exclusively on tasks sampled from $\mathcal{D}_a^{train}$ and evaluated on new tasks from $\mathcal{D}_a^{test}$, to have fair results with the streaming scenario. Therefore, unlike the GEOM model introduced in Sect. 5, the offline baseline does not assess cross-domain generalization; instead, it measures the model's ability to "adapt" to new tasks from known domains, as typical in in-domain meta-learning.

Fig. 7 illustrates the performance of GEOM-S using the static and proportional approach relative to the offline baseline, where all the datasets are available simultaneously. The relative accuracy is computed as the difference between the accuracy achieved with each approach and the accuracy of the offline baseline, which is set as the reference point at zero. More quantitative results can also be found in Tab. 17 in Appendix A.7. As expected, the proportional approach results in an overall better performance compared to static, particularly for the final three datasets, in the OCR domain. These datasets are significantly larger in terms of both images and classes, and the static approach allocates an insufficient number of epochs to achieve even partial convergence. In contrast, the proportional approach addresses this limitation by assigning a more appropriate number of training epochs based on dataset size. Despite its advantages, the proportional approach presents challenges in real-world scenarios. It assumes prior knowledge of the size of incoming

datasets to appropriately distribute training time/epochs, which is often unrealistic. Furthermore, when a new dataset is introduced, it is impossible to retroactively adjust the epochs allocated to previous datasets, as no information from them is retained. A more practical alternative might involve training the model until convergence on each dataset, as commonly done in continual learning applications (Wang et al., 2024b). However, determining the convergence point remains a challenging task (Han et al., 2023), and with a large number of datasets, this approach can be prohibitively time-consuming and computationally expensive. Considering these constraints, we adopt the static approach for the remainder of this paper. While it may not achieve optimal performance in all cases, it provides a consistent and practical framework for evaluating GEOM-S in streaming scenarios.

### 6.2 Analysis of forgetting

To evaluate the model's ability to retain previously learned knowledge, we adopt the backward transfer (BWT) metric, which is widely used in the continual learning literature (Wang et al., 2024b; Lopez-Paz & Ranzato, 2017). BWT provides insight into how well the model maintains performance on earlier tasks as new ones are introduced. In this work, we modify the traditional use of BWT to focus on *domain-based forgetting*, rather than merely task-level forgetting. Specifically, we compute BWT as follows:

$$BWT = \frac{1}{A-1} \sum_{a=1}^{A-1} R_{A,a} - R_{a,a}, \tag{3}$$

where $A$ is the total number of datasets (30 in Meta-Album) and $R_{a,b}$ (with $b < a$) is the average classification accuracy of the model on tasks sampled from $\mathcal{D}_b^{test}$ after training on $\mathcal{D}_a^{train}$. While the BWT is commonly used to measure forgetting in traditional continual learning setups, where tasks typically belong to the same domain, in our case, the dataset $\mathcal{D}_b$ belongs to domains that are different from the domain of $\mathcal{D}_a$. This distinction allows us to evaluate domain-based forgetting, which is the focus of our analysis. To calculate the BWT, we follow the same domain-based sequential order described in the previous section. After training on all datasets from a particular domain in the sequence, we save the model checkpoint and evaluate its performance on test tasks sampled from datasets belonging to previously encountered domains. The resulting accuracies are then used to calculate the average BWT as in Eq. 3. Unlike in typical continual learning settings (Lopez-Paz & Ranzato, 2017), where models are trained until convergence on each dataset, we restrict the training time on each dataset to 20 epochs, following the *static* approach outlined in Sect. 6.1. Moreover, we evaluate the model on entirely new tasks that are distinct from those used for training. In this context, the BWT metric captures the model's ability to leverage previously learned knowledge to generalize to new tasks that represent previously encountered domains. The results, reported in Fig. 8 and in Tab. 4, indicate that early in training, when the model has not yet developed a strong internal representation of the datasets, the model tend to forget, as represented by the negative BWT. However, as training progresses and the model refines its representations, the BWT increases, reflecting improved retention and generalization. This is particularly surprising considering the length of the sequence (30 diverse datasets) and the fact that forgetting is a common challenge in continual learning approaches. Interestingly, the model's performance on previously seen domains even improves as it encounters datasets from new domains, leading to positive BWT values. This supports the findings in Sect. 5.2, which show that an increased number of classes enables the model to generalize more effectively to unseen tasks. This approach aligns well with real-world applications, where new data becomes available over time and seamlessly integrates into the learning process, showcasing the practicality and effectiveness of GEOM-S in diverse, dynamic environments.

### 6.3 Curriculum learning

To emulate how humans build knowledge over time, we propose ordering the datasets based on their level of difficulty. However, the literature lacks a clear consensus on how to effectively quantify dataset difficulty (Soviany et al., 2022; Faber et al., 2024). In this work, we address this gap by utilizing two metrics: a TL-based technique in Sect. 6.3.1 and an OT computation in Sect. 6.3.2. These metrics provide a measure of similarity between datasets, enabling us to establish an order and construct various curricula.

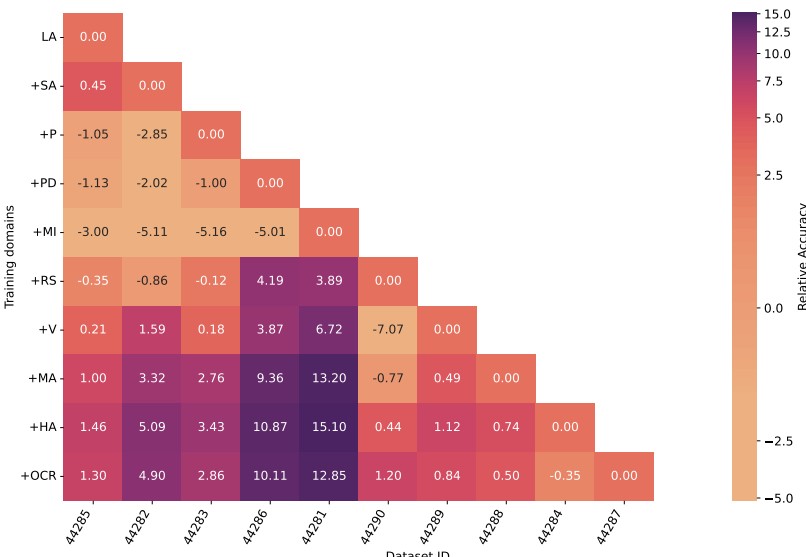

Table 4: Average BWT values computed using a domain-based ordered sequence, as described in Sect. 5. For each domain (denoted in the rows), the model is trained on all datasets from the previous domains, up to that point, and the BWT value is calculated by evaluating the model on test tasks sampled from all previously encountered datasets. The calculation is performed as detailed in Eq. 3 using only datasets from the first release of Meta-Album Mini for simplicity and consistency with the results in Fig. 8.

Figure 8: Heatmap showing the performance difference, used to compute the BWT, on datasets from the first release of Meta-Album Mini (one per domain), training GEOM-S with the static approach and the domain-based streaming scenario described in Sect. 6.1. Each entry $e_{r,c}$ represents the difference in accuracy on tasks sampled from dataset $\mathcal{D}_c$, (column), when the model is trained on all datasets up to domain $r$ (row) versus when the model is trained on all datasets up to the domain that $\mathcal{D}_c$ belongs to. The sequence order of domains is as follows: Large Animals (LA), Small Animals (SA), Plants (P), Plant Diseases (PD), Microscopy (MI), Remote Sensing (RS), Vehicles (V), Manufacturing (MA), Human Actions (HA), OCR. Higher values in the lower part of the heatmap indicate the model's ability to leverage knowledge from previously observed domains to improve performance as more domains are introduced.

|        | BWT   |
|--------|-------|
| LA     | –     |
| + SA   | 0.45  |
| + P    | −1.9  |
| + PD   | −1.15 |
| + MI   | −4.57 |
| + RS   | 1.37  |
| + V    | 0.92  |
| + MA   | 4.19  |
| + HA   | 4.78  |
| + OCR  | 3.80  |

For simplicity and to optimize computational resources, all curricula are built considering the Micro size of Meta-Album, which comprises 31 920 images with a balanced distribution of classes and images per class across all datasets. This is particularly important as unbalanced datasets could skew the computation and affect the results (Mundt et al., 2023; Schouten, 2024). Once the curricula are defined, the dataset indices are replaced with those corresponding to Meta-Album Mini. The full training and evaluation pipeline is then executed using the datasets in Meta-Album Mini to maintain consistency with prior experiments and to avoid the overfitting problem described in Sect. 5.3.

### 6.3.1 Transfer learning-based curriculum

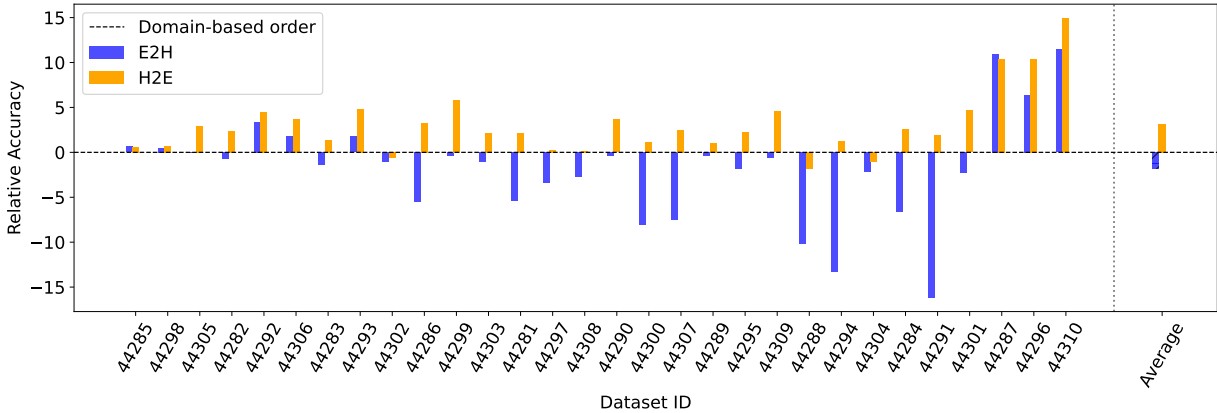

Figure 9: Relative accuracy of the E2H and H2E curricula compared to the domain-based order baseline. The relative accuracy is computed as the difference in performance between each curriculum and the domain-based approach, which is set as the reference point at zero. Datasets in E2H and H2E are ordered according to a TL-based approach and the results are obtained with Meta-Album Mini. The last column reflects the average relative accuracy across all datasets.

As one of the two approaches proposed for constructing curricula, we apply a TL-based strategy to evaluate the dataset difficulty. This method is grounded in the hypothesis that datasets where a model achieves high performance after fine-tuning are inherently less challenging, compared to others with lower performance. By ranking datasets based on their difficulty using this approach, we establish a curriculum that can influence training order and model performance. Specifically, we use the same pre-trained feature extractor employed in GEOM-S, a ResNet-50 (He et al., 2016) model pre-trained on ImageNet-1k (Deng et al., 2009), and we fine-tune a simple projection head with ReLU non-linearity and batch normalization to classify the 20 classes of each dataset. We optimize the cross-entropy loss with Adam optimizer for 100 epochs, starting from a learning rate of $10^{-4}$ and smoothly reducing it with a cosine annealing scheduler. We then evaluate the performance of the fine-tuned model on the test split of each dataset and use this value as a metric to rank datasets. Applying this TL-based approach resulted in the following dataset order:

- TL-based order: [44304, 44299, 44288, 44305, 44283, 44284, 44285, 44298, 44300, 44286, 44291, 44282, 44301, 44294, 44281, 44307, 44290, 44295, 44306, 44293, 44292, 44289, 44303, 44287, 44309, 44297, 44302, 44310, 44296, 44308]

where datasets are ordered from easiest (highest accuracy) to most difficult (lowest accuracy). For our experiments, we evaluate the following baselines:

- *Easy-to-Hard* (E2H): a curriculum learning baseline where datasets are presented from the easiest to the most difficult (increasing difficulty, from dataset ID 44304 to 44308).

- *Hard-to-Easy* (H2E): a curriculum learning baseline where datasets are presented from the most difficult to the easiest (decreasing difficulty, from dataset ID 44308 to 44304). It is sometimes referred to as anti-curriculum (Soviany et al., 2022) in the literature.

- *Domain-based*: the dataset order as presented in Meta-Album, where datasets are grouped into domains, as explained in Sect. 6.1.

The results, illustrated in Fig. 9 and, more extensively, in Tab. 18 in Appendix A.7 confirm that ordering the datasets based on their level of difficulty can improve model performance in the sequential setting. This approach provides a more realistic alternative than simply using a random dataset order, or simply grouping datasets into domains. Interestingly, the best performance is achieved with the H2E configuration, as demonstrated by the average performance gain in the last column of Fig. 9. While this might seem counterintuitive (Sheybani et al., 2024a; Bengio et al., 2009), the H2E configuration may benefit the model by exposing it to challenging datasets early in training. This early exposure allows the model to explore the parameter space more extensively, reducing the risk of overfitting to simpler datasets and fostering greater generalization (Soviany et al., 2022). This behavior is further illustrated in Fig. 10, which shows the learning trend for the E2H and H2E scenarios.

In the E2H setting, the model initially achieves high accuracy on the easiest datasets, but its performance deteriorates as more challenging datasets are introduced. This fact raises some interesting considerations. Firstly, building a sequence that only takes into account the distribution shift from the pre-acquired knowledge of the feature extractor may hamper the model's ability to generalize to harder datasets. Secondly, this highlights the importance of the first phase of training, as observing only simpler datasets at the beginning of the training time could saturate the knowledge of the model and make it less flexible to adapt to new, harder datasets later. Lastly, progressively increasing the difficulty of a dataset at time $T$, without accounting for the knowledge acquired up to that point, may require longer training times when moving to harder

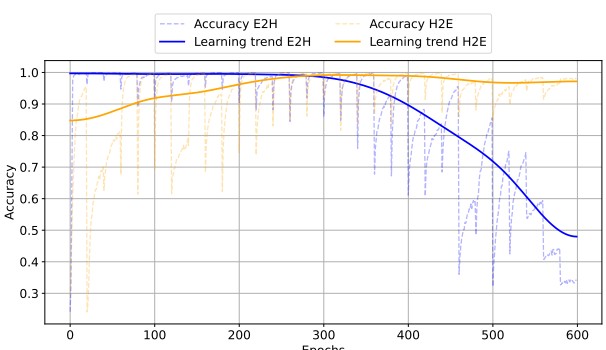

Figure 10: Comparison of learning trends for E2H and H2E TL-based curricula with GEOM-S.

datasets. However, allocating sufficient training epochs for more challenging ones remains a significant challenge, due to a lack of precise metrics for quantifying dataset complexity and the inherent difficulties in estimating the time required for convergence, as discussed in Sect. 6.1. Finally, Appendix A.3 demonstrates the effectiveness of the H2E strategy over E2H when the feature extractor is jointly trained with the rest of the model. This ensures that no pre-acquired knowledge influences the curricula.

### 6.3.2 Optimal transport curriculum

While the TL-based approach provides an intuitive measure of dataset difficulty relative to a pre-trained model, it does not account for difficulty among datasets, and how the knowledge acquired from the previously seen dataset might influence the current. This limitation motivates the use of an OT-based approach (Chang et al., 2023), which quantifies dataset similarity by computing the minimal cost required to transform one probability distribution into another (Peyré et al., 2019). However, applying OT to different datasets presents challenges, as their label sets are often disjoint and unrelated. To overcome this issue, the Optimal Transport Dataset Distance (OTDD) metric in (Alvarez-Melis & Fusi, 2020) proposes to represent a label-induced distribution $\alpha_y$ as a Gaussian $\mathcal{N}(\hat{\mu}_y, \hat{\sum}_y)$ and compute the distance between datasets as follows:

$$d_{OT}(D_A, D_B) = min_{\pi \in \Pi(\alpha, \beta)} \int_{Z \times Z} d_Z(z, z')^p \phi(z, z'). \tag{4}$$

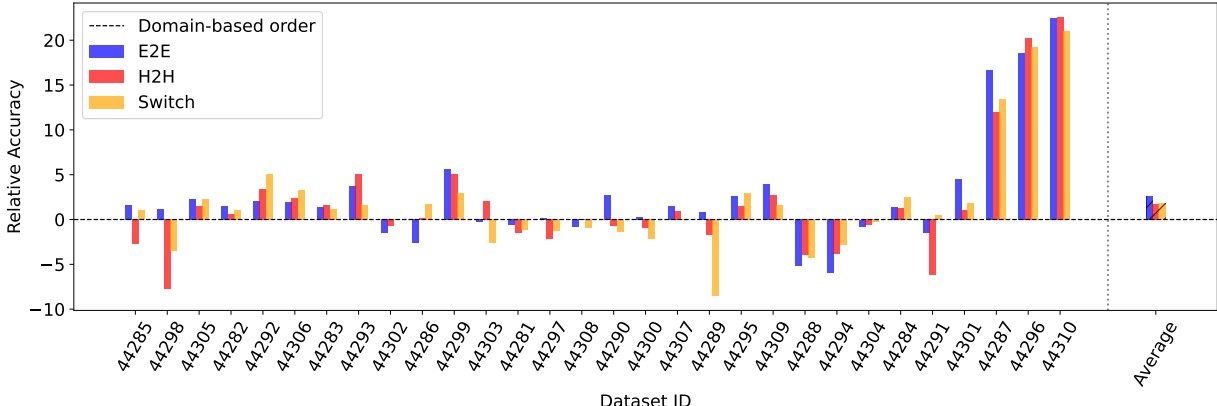

Figure 11: Relative accuracy of the E2E, H2H, and Switch curricula compared to the domain-based order baseline. The relative accuracy is computed as the difference in performance between each curriculum and the domain-based approach, which is set as the reference point at zero. The datasets are ordered based on OTDD (Alvarez-Melis & Fusi, 2020) and the results are obtained with Meta-Album Mini. The last column reflects the average relative accuracy across all datasets.

where $z \triangleq (x, y)$ represents a pair of feature-label and $\mathcal{Z} \triangleq \mathcal{X} \times \mathcal{Y}$. Therefore, we can define

$$d_Z(z, z') = d_Z((x, y), (x', y')) \triangleq (d_X(x, x')^p + W_p^p(\alpha_y, \alpha_{y'}))^{1/p}.$$

as the p-Warssertein distance between feature-label pairs. Representing $\alpha_y$ as a Gaussian is possible after embedding the data with a non-linear mapping (e.g., a neural network) (Seddik et al., 2020). In our experiments, we embed the datasets using a ResNet-50 architecture pre-trained on ImageNet-1k, and we compute the OOTD distance in this embedding space. The similarities between the datasets are visualized in Fig. 14 and in Fig. 15 (Appendix A.7). Notably, datasets from Microscopy, Remote Sensing, and Plant Diseases are the most dissimilar from all others, appearing at the top of the similarity figure. This observation aligns with expectations, as these datasets belong to domains that are significantly different from the rest. Their images are acquired using specialized devices, such as microscopes or GPS systems, and have distinct resolutions and characteristics.

Due to the high computational cost of computing OTDD for large datasets (Alvarez-Melis & Fusi, 2020), we build the curricula using the Micro size of Meta-Album, although we train and evaluate the model using the corresponding datasets in Meta-Album Mini, as previously described. The first step in constructing an OT-based curriculum is identifying a starting dataset. Intuitively, the dataset most similar to ImageNet-1k should be the easiest for our model, as the feature extractor in GEOM-S is pre-trained on ImageNet-1k. However, directly identifying this dataset using OTDD is impractical due to the imbalance between ImageNet-1k and Meta-Album datasets and the wide domain coverage of ImageNet-1k compared to the specific domains in Meta-Album. Instead, we set the first dataset in the TL-based curriculum (dataset ID 44304) as the starting point for all OT-based curricula. From this starting point, we construct three distinct curricula[1]:

- *Easy-to-Easy* (E2E): a curriculum learning baseline where each dataset is the easiest (most similar) with respect to the previous one.

- *Hard-to-Hasy* (H2H): a curriculum learning baseline where each dataset is the most difficult (most dissimilar) with respect to the previous one.

- *Switch*: a curriculum learning baseline where the order is decided by switching from the easiest to the most difficult dataset, iteratively.

---

[1]The detailed order of dataset IDs can be found in Appendix A.2.

It is worth noting that these dataset orders are inherently different from those derived using the TL-based method in Sect. 6.3.1. Unlike the TL-based approach, which calculates similarity relative to ImageNet-1k, OTDD measures pairwise distances between Meta-Album datasets directly. Additionally, always beginning with the dataset closest to ImageNet-1k could potentially replicate the shortcomings observed in the E2H curriculum from Sect. 6.3.1. For this reason, the results of OT-based and TL-based curricula should be viewed as complementary rather than directly comparable.

For consistency and clarity with the results in Sect. 6.3.1, we report the relative accuracy of each curriculum against the domain-based order inherent in Meta-Album. The results, shown in Fig. 11, reaffirm that employing a curriculum strategy yields superior performance compared to simply grouping datasets by domain. Furthermore, it appears that the best-performing curriculum across all datasets is E2E. This aligns with our expectations, as gradual changes in the observed data encourage the model to accumulate knowledge over time, avoid forgetting, and build upon prior learning incrementally. Such an approach mirrors the natural learning processes, which are characterized by steady progress through increasingly challenging tasks that foster both retention of knowledge and generalization.

## 7 Unsupervised training

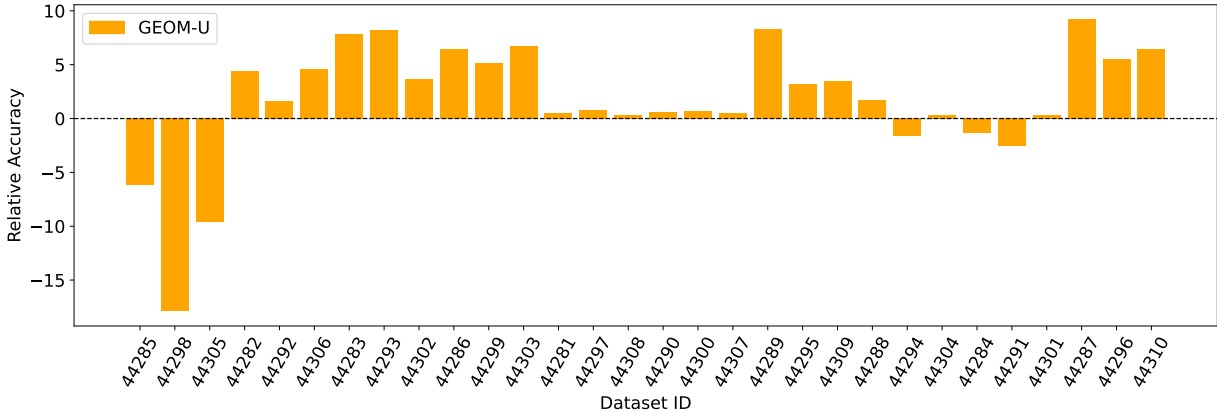

Figure 12: Relative accuracy of GEOM-U compared to CAMeLU (Vettoruzzo et al., 2025) in the unsupervised scenario computed as the performance difference between the two approaches, where CAMeLU is set as the reference point at zero. GEOM-U is trained on an unsupervised version of Meta-Album following the task creation mechanism of CAMeLU and using the LOO approach described in Sect. 5. CAMeLU is trained on ImageNet-1k (Deng et al., 2009), after removing the labels. The evaluation is performed on few-shot tasks sampled from the Meta-Album datasets from the left-out domain.

In many real-world scenarios, collecting a large amount of labeled data to train a model is challenging and impractical. Instead, it is more common to encounter smaller datasets collected from various environments or domains, often without labels. Motivated by this real-world setting, we extend our analysis to the unsupervised scenario, investigating whether training on a collection of small-scale, unlabeled datasets can improve the performance over unsupervised training on a large-scale dataset. We adopt the same rationale proposed in CAMeLU (Vettoruzzo et al., 2025), which generates training tasks from unlabeled data and uses these tasks to train an in-context learner similar to GEOM. During evaluation, we assume the availability of standard few-shot tasks, where the context is fully labeled. We refer to this variant of GEOM as **GEOM-U** (GEOM-*Unsupervised*). The main difference between GEOM-U and CAMeLU is the training data. While GEOM-U is trained with tasks sampled from the Meta-Album datasets across diverse domains, CAMeLU is trained on ImageNet-1k (Deng et al., 2009), a large-scale benchmark that represents a wide data distribution.

To construct tasks, we follow the process outlined in CAMeLU (Vettoruzzo et al., 2025). Let $\mathcal{T}_i$ be the task we want to construct. As detailed in Sect. 3, it consists of $K \times N$ context examples and $Q$ query images. The context samples are generated by randomly sampling $N$ images from an unlabeled training

dataset $\mathcal{D}_a^{train} = \{x_j\}$. Each sampled image is augmented $K$ times with distinct augmentation functions, and all augmented versions of a sample $x_n$ are assigned the same pseudo-label $n \in \{1, \ldots, N\}$. Queries are created using a two-step process. For each query, a random augmentation is applied to an image $x_n$, yielding $\tilde{x}_{n,j}$, and a strategy inspired by *mixup* (Zhang et al., 2018) is used to generate the query image as $x_q = \lambda z_j + (1 - \lambda)\tilde{x}_{n,j}$, where $\lambda \sim Beta(\alpha, \beta)$ and $z_j$ is a random example from $\mathcal{D}_a^{train} = \{x_j\}$. The same label $n$ as the context sample $x_n$ used for the generation is then assigned to the resulting query $x_q$. Additional details can be found in the original CAMeLU paper (Vettoruzzo et al., 2025).

We compare the performance of GEOM-U against CAMeLU, using the architectures described in Sect. 3.3 and the LOO configuration in Sect. 5, where datasets from an entire domain are excluded during training to prevent the leakage of information during evaluation. The results, shown in Fig. 12 and detailed in Tab. 20 (Appendix A.8), indicate that training an in-context learner on diverse small-scale datasets outperforms training on a single large-scale dataset like ImageNet-1k, even in the unsupervised scenario. This performance improvement likely stems from the diversity introduced by the smaller datasets across different domains. The resulting variability in tasks encourages the model to learn domain-invariant features, rather than simply associating images and classes. Additionally, since GEOM-U is trained on small-scale datasets, there is a high chance that multiple images from the same class appear within a single task. Without explicit class labels, the model is forced to treat these instances as distinct entities, rather than grouping them together, increasing task complexity. This, in turn, encourages the development of a more flexible and robust learner capable of handling diverse and unseen data. The only cases where GEOM-U underperforms CAMeLU are in the Large Animals domain. Due to significant overlap with ImageNet-1k (see Fig. 2), this domain suffers from data leakage, giving CAMeLU a significant advantage.

## 8 Conclusions and future work

This work explored the generalization capabilities of ICL within a meta-learning framework, by shifting from reliance on vast, unstructured datasets to a more focused, human-inspired approach using multiple smaller, domain-specific datasets. We demonstrated significant improvements in the ability of ICL models to generalize across tasks. This paradigm not only fosters broader generalization but also enhances interpretability, modularity, and adaptability. The smaller datasets allow for greater control over training dynamics, enabling targeted adjustments to the learning process and facilitating the integration of new data. By presenting datasets sequentially, we observed that in-context learners accumulate knowledge over time, improving their performance without erasing prior learning, a phenomenon akin to lifelong learning in humans. Curriculum strategies based on dataset difficulty proved particularly effective, highlighting the importance of structured exposure to tasks rather than random ordering in fostering adaptive learning. Since real-world data is often noisy, mislabeled, or entirely unlabeled, we also evaluated the model's robustness to label noise (Appendix A.5) and found that it maintains strong performance despite such imperfections. Additionally, our experiments with unsupervised meta-learning demonstrated that the model can generalize effectively even when trained on pseudo-labeled data derived from augmentations. This approach opens avenues for deploying in-context learners in resource-constrained or data-scarce environments, further bridging the gap between artificial systems and natural learning processes. These findings altogether highlight the crucial role of data diversity, task design and learning sequence in unlocking robust generalization across different domains.

Despite the promising outcomes of this study, some open questions remain. For instance, determining the minimum number of classes or the optimal class-to-sample ratio required for effective learning could refine dataset design. Addressing dataset imbalance is another key challenge, especially in streaming scenarios where data availability may vary. Integrating methods that dynamically weigh datasets during training could mitigate these challenges and further enhance performance. Exploring adaptive curriculum strategies that align task difficulty with the model's learning progress may offer a more dynamic and effective training paradigm. Another compelling direction for future research is extending our approach to causal transformers. GEOM is currently based on a meta-learning formulation that assumes permutation invariance within tasks, which naturally aligns with the use of non-causal transformers. Investigating how GEOM training translates to a causal architecture, and whether similar generalization benefits can be retained, represents an important step toward broader applicability and alignment with recent trends in the field.

In conclusion, this work proposes a paradigm inspired by human processes of learning for training in-context learners, emphasizing the importance of structured, diverse, and incremental learning processes. By bridging the gap between artificial and natural learning paradigms, we take a meaningful step toward developing AI systems capable of more efficient, robust, and generalizable learning.

## 9    Broader impact statement

This work advances ICL by leveraging meta-learning and structured, domain-specific datasets for training, enhancing generalization, adaptability, and modularity. Although GEOM relies on Meta-Album, a collection designed to ensure balance across diverse domains, it does not prevent the potential misuse of other datasets during training. In Sect. 5, we highlight the advantage of using small datasets, as they are easier to update and replace. However, this approach may introduce strong distribution biases and unintended side effects during inference (Menon et al., 2020). Furthermore, despite their ease of maintenance, small datasets may suffer from labeling inaccuracies and fail to fully capture the diversity of the training distribution. These could lead to some categories to be underrepresented. Additionally, reliance on pseudo-labeling and augmentation techniques in unsupervised training introduces potential vulnerabilities. While our experiments in Appendix A.5 demonstrate that the model is robust to label noise, adversarial attacks remain a concern and warrant further investigation. Future work should focus on strengthening robustness against such threats while ensuring ethical and responsible deployment of in-context learners.

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

# A  Appendix

## A.1  Default notation

**General terms**

| | |
|---|---|
| ICL | In-context learning |
| LLM | Large language model |
| LOO | Leave-one-out evaluation |
| BWT | Backward transfer |
| TL | Transfer learning |
| OT | Optimal transport |
| OTDD | Optimal transport dataset distance metric (Alvarez-Melis & Fusi, 2020) |

**Curriculum learning strategies**

| | |
|---|---|
| Domain-based | Sequence of datasets ordered as in Meta-Album |
| E2H | Easy-to-hard curriculum |
| H2E | Hard-to-easy curriculum |
| E2E | Easy-to-easy curriculum |
| H2H | Hard-to-hard curriculum |

**GEOM training variants**

| | |
|---|---|
| GEOM-IN | GEOM trained on ImageNet-1k |
| GEOM-M | GEOM trained on a fully merged version of Meta-Album |
| GEOM-S | GEOM trained sequentially |
| GEOM-U | GEOM trained in an unsupervised manner |

**Dataset and task**

| | |
|---|---|
| $\mathcal{D}$ | The set of available datasets $\mathcal{D} = \{\mathcal{D}_a \mid a = 1, \ldots, A\}$ |
| $\mathcal{D}^{LOO}$ | The set of datasets used for evaluation in the LOO scenario |
| $\mathcal{D}_a^{train}$ | A dataset split used during training |
| $\mathcal{D}_a^{test}$ | A dataset split used during evaluation |
| $\mathcal{T}_i$ | A task sampled for training the model |
| $\mathcal{T}_{new}$ | A new task sampled for evaluation |
| $S_i$ | A sequence generated from the task $\mathcal{T}_i$ |
| $N$-way $K$-shot | Few-shot classification with $K$ examples for each of the $N$ classes |
| $Q$ | Number of queries per task |
| $x_j$ | An image, or sample |
| $y_j$ | A label associated to sample $x_j$ |

**Model components**

| | |
|---|---|
| $f_\psi$ | The image encoder (i.e., feature extractor) |
| $g_\phi$ | The label encoder |
| $M_\theta$ | The non-causal transformer encoder with linear classification layer |
| $\mathcal{L}$ | The cross-entropy loss |

**Meta-Album benchmark**

| | |
|---|---|
| Meta-Album | A benchmark consisting of 30 datasets spanning ten domains |
| LA | Large Animals domain |
| SA | Small Animals domain |
| P | Plants domain |
| PD | Plant Diseases domain |
| MI | Microscopy domain |
| RS | Remote Sensing domain |
| V | Vehicles domain |
| MA | Manufacturing domain |
| HA | Human Actions domain |
| OCR | OCR domain |
| Meta-Album sizes | The three different sizes of Meta-Album (Micro, Mini, Extended) |
| (Meta-Album) Micro | The size called "Micro" in Meta-Album |
| (Meta-Album) Mini | The size called "Mini" in Meta-Album |
| (Meta-Album) Extended | The size called "Extended" in Meta-Album |
| Meta-Album releases | Batches of 10 datasets from distinct domains progressively added to the benchmark |
| (Meta-Album) *First* | First release of Meta-Album (10 datasets overall) |
| (Meta-Album) *Second* | The combined set of datasets from the first and second Meta-Album releases (20 datasets overall) |
| (Meta-Album) *Third* | The combined set of datasets from the first, the second, and the third Meta-Album releases (30 datasets overall) |

**Sequential**

| | |
|---|---|
| $t$ | A timestamp in $1, \dots, T$ |
| $\mathcal{D}_t^{train}$ | A train dataset sampled at timestamp $t$ |
| $\mathcal{D}_t^{test}$ | A test dataset sampled at timestamp $t$ |
| $R_{a,b}$ | Model accuracy on $D_b^{test}$ after training on $D_a^{train}$ |

**Unsupervised**

| | |
|---|---|
| $x_n$ | An image with pseudo-label $n \in \{1, \ldots, N\}$ |
| $z_j$ | A randomly sampled image from the training dataset |
| $\lambda$ | A hyperparameter sampled from a $Beta(\alpha, \beta)$ distribution |
| $\tilde{x}_{n,j}$ | An augmented version of image $x_j$ with pseudo-label $n$ |
| $x_q$ | A query image |

## A.2 Experimental details

**Datasets.** In our experiments, we use the Meta-Album benchmark (Ullah et al., 2022)[2], which consists of a collection of datasets spanning 10 different domains. Compared to other benchmark collections, such as Meta-Dataset (Triantafillou et al., 2020) or NEVIS (Bornschein et al., 2024), Meta-Album offers a more balanced dataset distribution while ensuring clear domain separation. The original Meta-Album paper (Ullah et al., 2022) defines a total of 40 datasets, but at the time of writing and experimental setup, only three releases are available, reducing the number of accessible datasets to 30. Each Meta-Album dataset consists of RGB images with a fixed resolution of $128 \times 128$ pixels. For our experiments, we upscale theses images to $224 \times 224$ pixels to match the input requirements of a ResNet-50 pre-trained feature extractor.

Meta-Album datasets are organized into **releases** and **sizes**. Each release introduces 10 new datasets, one for each domain. Therefore, when mentioning the *First* release, we indicate the set of 10 datasets that originally composed Meta-Album, while *Second* and *Third* refer to the collection comprising 10 additional datasets, each, that were introduced by each release (20 and 30 overall, respectively). The datasets also vary in size, with three available configurations: Micro, Mini, and Extended. Micro ensures a balanced distribution, where each dataset consists of 20 classes (with the exception of dataset IDs 44313 and 44312 which have 19 classes), with 40 images per class. Therefore, the total number of images for the 30 datasets that compose the *Third* release of Micro is $31\,920$. Instead, Mini is the ideal size for few-shot learning scenarios as it contains a balanced number of images per class (40), while allowing for a greater number of classes, reaching up to 706 classes per dataset. This increases task diversity, leading to a total number of $163\,200$ images in the *Third* release. Extended is the largest configuration, containing $1\,384\,616$ images, although it contains fewer classes than Mini, as the OCR domain is not included. Table 5 summarizes these details, while a comparison between the number of classes and images between Mini and Extended for each dataset of the *Third* release is provided in Tab. 6.

The dataset splits used in our experiments depend on the specific learning scenario. When evaluating the generalization on unseen domains, as in Sect. 5 and Sect. 7, training and test datasets do not overlap, thus the entire dataset can be used either for training or evaluation purposes. In streaming scenarios (Sect. 6.1) we allocate 80% of dataset classes for training the model and the remaining 20% for the evaluation phase. If a dataset is too small, i.e., the 20% split results in fewer than five classes, we increase the evaluation set size to ensure at least one example per class, allowing us to create a 5-way classification task.

Table 5: Statistics of the Meta-Album collection for Micro, Mini, and Extended sizes, based on the three available releases. The dataset details are obtained using Python's pip package `openml==0.14.2`.

| Size | #domains | #datasets | #images | min/max #classes | min/max #images per class |
|---|---|---|---|---|---|
| Micro | 10 | 30 | $31\,920$ | 19 / 20 | 40 / 40 |
| Mini | 10 | 30 | $163\,200$ | 19 / 706 | 40 / 40 |
| Extended | 9 | 27 | $1\,384\,616$ | 19 / 315 | 1 / $187\,384$ |

---

[2]Meta-Album datasets are downloaded using the `openml==0.14.2` version of the OpenML library (Bischl et al., 2021) via the Python pip package.

Table 6: Dataset information for Mini and Extended splits. For every dataset ID, the overall number of images and the number of classes used for training/evaluation are defined.

(a) Size Mini of Meta-Album.

| Dataset | Images | Train | Evaluation |
|---------|--------|-------|------------|
| 44285 | 12600 | 252 | 63 |
| 44298 | 4800 | 96 | 24 |
| 44305 | 2000 | 40 | 10 |
| 44282 | 3440 | 69 | 17 |
| 44292 | 4080 | 82 | 20 |
| 44306 | 4160 | 84 | 20 |
| 44283 | 4080 | 82 | 20 |
| 44293 | 1000 | 20 | 5 |
| 44302 | 1000 | 20 | 5 |
| 44286 | 1520 | 31 | 7 |
| 44299 | 1000 | 20 | 5 |
| 44303 | 1080 | 22 | 5 |
| 44281 | 1320 | 27 | 6 |
| 44297 | 760 | 14 | 5 |
| 44308 | 840 | 16 | 5 |
| 44290 | 1800 | 36 | 9 |
| 44300 | 1800 | 36 | 9 |
| 44307 | 1520 | 31 | 7 |
| 44289 | 7840 | 157 | 39 |
| 44295 | 840 | 16 | 5 |
| 44309 | 1040 | 21 | 5 |
| 44288 | 2560 | 52 | 12 |
| 44294 | 1880 | 38 | 9 |
| 44304 | 10000 | 200 | 50 |
| 44284 | 2920 | 59 | 14 |
| 44291 | 1560 | 32 | 7 |
| 44301 | 1160 | 24 | 5 |
| 44287 | 28240 | 565 | 141 |
| 44296 | 28240 | 565 | 141 |
| 44310 | 28120 | 563 | 140 |
| Total | 163 200 | 3270 | 810 |

(b) Size Extended of Meta-Album

| Dataset | Images | Train | Evaluation |
|---------|--------|-------|------------|
| 44320 | 49053 | 252 | 63 |
| 44331 | 20480 | 96 | 24 |
| 44338 | 37317 | 40 | 10 |
| 44317 | 473237 | 77 | 19 |
| 44326 | 75222 | 82 | 20 |
| 44340 | 170491 | 94 | 23 |
| 44318 | 8189 | 82 | 20 |
| 44327 | 120688 | 20 | 5 |
| 44335 | 15122 | 20 | 5 |
| 44321 | 54305 | 31 | 7 |
| 44332 | 1596 | 21 | 5 |
| 44336 | 2549 | 22 | 5 |
| 44316 | 4060 | 27 | 6 |
| 44330 | 5530 | 14 | 5 |
| 44342 | 15050 | 16 | 5 |
| 44324 | 31500 | 36 | 9 |
| 44333 | 36707 | 36 | 9 |
| 44341 | 43821 | 32 | 8 |
| 44323 | 16185 | 157 | 39 |
| 44329 | 9625 | 16 | 5 |
| 44343 | 138367 | 21 | 5 |
| 44322 | 8675 | 52 | 12 |
| 44328 | 5640 | 38 | 9 |
| 44337 | 25000 | 200 | 50 |
| 44319 | 10416 | 59 | 14 |
| 44325 | 3389 | 32 | 8 |
| 44334 | 2402 | 24 | 5 |
| Total | 1 384 616 | 1597 | 395 |

We also include external datasets for evaluation purposes. We use ImageNet-1k (Deng et al., 2009) both as a baseline and to compute the overlap with class names and concepts between the classes in ImageNet-1k and Meta-Album. As described in Sect. 4, when searching for the exact match, we extract the names of the classes from the label files of each dataset, pre-process them by removing any underscore and apostrophe, and make the whole word lowercase. If a label in ImageNet-1k is defined by multiple names, from the coarsest to the finest, we select only the finest word. However, this analysis might overlook several minor differences, misspelled items, and hyphenated words. For this reason, we take a step further and try to identify related concepts by means of CLIP (Radford et al., 2021) embeddings of the label names. We take the same pre-processed words, exclude those that had already found a match with the previous technique, and embed them with the aforementioned feature extractor. For each dataset, we then compute the cosine similarity between each embedded word in ImageNet-1k and every word that is still unmatched in the current dataset and we keep the highest score for each word. To set a general threshold that could fit all the datasets, we compute the $90^{th}$ percentile of the similarity distribution for each dataset, in order to only keep matches that have high similarity. Then, we select the median value among all the datasets' percentiles and we define a threshold set at 0.83.

We also consider different datasets for evaluation purposes, as described in Section 3.3. For each dataset, we only use the test split generated following the splits proposed in the previous literature. In particular, we considered CIFAR-fs which consists of 20 classes for testing (Bertinetto et al., 2019); CUB (Wah et al., 2011), which consists of 30 classes in the test set; Aircraft (Triantafillou et al., 2020) with only 15 classes in the test split; Meta-iNat (Wertheimer & Hariharan, 2019) consists of 227 classes reserved for testing. For EuroSat (Helber et al., 2018) and ISIC (Codella et al., 2018), which were not initially meant for meta-learning, we use all their classes in test, which are 10 and 7, respectively.

**Training details.**  We build each training episode as an $N$-way $K$-shot classification task, where $N$ and $K$ are fixed to 5. Following the same model architecture as in Vettoruzzo et al. (2025), we use a ResNet-50 (He et al., 2016) feature extractor $f_\psi$ pre-trained on ImageNet-1k and a class encoder $g_\phi$ consisting of a single learnable layer that maps the $N$ class labels to a dimensionality of 256. The non-causal transformer consists of 8 encoder layers, each incorporating a multi-head self-attention block with 8 attention heads, an MLP with a reverse bottleneck of 3072 (with GeLU activation function), and an input-output feature size of 2304, which corresponds to the concatenation of feature label (with a size of 2048) and the class label features (with a size of 256). Finally, a single-layer classifier maps the transformer output to the predicted category. The episodic training is performed for $300\,000$ iterations with the Adam optimizer, an initial learning rate set at $10^{-5}$, and a warmup cosine scheduler. When referring to epochs and episodes, we define an epoch as a collection of 500 iterations, after which the trainloader is re-initialized. The total number of epochs is set to 600. For the subsequent evaluation, the best-performing model is saved as the one resulting in the highest validation accuracy across $50\,000$ new tasks, sampled from $\mathcal{D}_a^{test}, a = 1, \cdots, A$. The code is written in Python and the experiments are run on an NVIDIA A100-SXM4 GPU with 40GB of VRAM for faster execution. However, the model can also be run and debugged on consumer hardware, such as an NVIDIA GeForce RTX 3070 Ti Laptop GPU.

When selecting a dataset to sample a task from, our study defines three main approaches. The first, used in the supervised (offline) scenario detailed in Sect. 5 and in the *offline* baseline in Sect. 6.1, select each dataset with a probability $p(\mathcal{D}_a) = \frac{|\mathcal{D}_a|}{\sum_{\mathcal{D}_a \in \mathcal{D}} |\mathcal{D}_a|}$, ensuring larger datasets are sampled more frequently. The other two approaches refer to the streaming scenario described in Sect. 6.1, where datasets are processed sequentially. In the *proportional* approach, the number of training iterations allocated to each dataset depends on the size of the dataset. Given a total number of iterations $I$ (set to $300\,000$ by default), each dataset $\mathcal{D}_a$ receives $I_a = I \cdot \frac{|\mathcal{D}_a|}{\sum_{\mathcal{D}_a \in \mathcal{D}} |\mathcal{D}_a|}$ iterations before advancing to the next dataset. In contrast, the *static* approach assigns each dataset an equal number of iterations $I_a = \frac{I}{A}$, ensuring uniform training time across datasets.

Lastly, for the unsupervised part, we follow what is described in Vettoruzzo et al. (2025). We use the same sampling strategy as in supervised (offline) learning, but we assume no labeled data are available during training. We randomly draw $N$ samples from a dataset $\mathcal{D}_a$ and augment images to reconstruct the same $N$-way $K$-shot problem. Each support image is augmented $K$ times, with an augmentation function $\mathcal{A}_k$

sampled from a predefined set of transformations $\mathcal{A}$. Queries go through a two-step augmentation process to enhance diversity and increase the task complexity: firstly, $K$ queries are generated from the same image $x_j$ via another set of augmentations $\mathcal{A}_j$ and then mixed with an external sample $z_j$ drawn from the same dataset $\mathcal{D}_a$ with the following method: $x_q = \lambda z_j + (1 - \lambda)\tilde{x}_{n,j}$, where $\lambda \sim Beta(\alpha, \beta)$ with $\alpha = 1, \beta = 1$ and $\lambda \in (0, 0.5)$.

## A.3  TL based curricula

To further prove the effectiveness of our curriculum strategy and demonstrate that the learning trend shown in Sect. 6.3.1, where H2E generally achieves higher accuracy values than E2H, is not influenced by the frozen weights of ResNet50 pre-trained on ImageNet-1k, we use the very same architecture but we jointly train our image feature extractor from scratch. Although the results shown in Tab. 7 are lower than the original, which would require a thorough revision of the architecture and/or the training time, the learning trend shown in Fig. 13 still evidences that H2E generally achieves better performance than E2H.

Table 7: Accuracy results of GEOM-S using different TL-based curricula and a feature extractor trained from scratch: *easy-to-hard* (E2H), *hard-to-easy* (H2E), and domain-based order. The same number of epochs (20) is assigned to each dataset, using the *static* approach in Sect. 6.1. The bold font highlights the best-performing approach for each dataset. Results show the average across three complete runs of the algorithms.

| | Large Animals | | | Small Animals | | | | Plants | | | Plant Diseases | |
| | 44285 | 44298 | 44305 | 44282 | 44292 | 44306 | 44283 | 44293 | 44302 | 44286 | 44299 | 44303 |
|---|---|---|---|---|---|---|---|---|---|---|---|---|
| E2H | $26.49 \pm 3.08$ | $21.84 \pm 0.91$ | $23.96 \pm 1.56$ | $22.42 \pm 3.36$ | $22.18 \pm 1.21$ | $22.16 \pm 1.68$ | $33.60 \pm 5.32$ | $23.42 \pm 2.22$ | $22.52 \pm 1.33$ | $25.48 \pm 2.22$ | $26.15 \pm 11.25$ | $23.66 \pm 1.75$ |
| H2E | $\mathbf{49.32 \pm 4.91}$ | $\mathbf{29.79 \pm 1.12}$ | $\mathbf{38.77 \pm 3.26}$ | $28.14 \pm 10.17$ | $\mathbf{28.42 \pm 1.77}$ | $\mathbf{29.28 \pm 2.58}$ | $\mathbf{57.39 \pm 11.11}$ | $\mathbf{33.33 \pm 5.79}$ | $\mathbf{26.98 \pm 0.63}$ | $40.99 \pm 7.52$ | $33.26 \pm 18.56$ | $28.51 \pm 4.33$ |
| Domain-based | $39.59 \pm 2.44$ | $25.83 \pm 1.06$ | $33.30 \pm 2.59$ | $\mathbf{32.50 \pm 3.76}$ | $25.94 \pm 1.85$ | $26.70 \pm 0.98$ | $43.95 \pm 5.33$ | $30.92 \pm 6.62$ | $25.32 \pm 2.75$ | $\mathbf{44.27 \pm 5.63}$ | $\mathbf{39.60 \pm 12.69}$ | $\mathbf{29.51 \pm 5.47}$ |

| | Microscopy | | | Remote Sensing | | | Vehicles | | |
| | 44281 | 44297 | 44308 | 44290 | 44300 | 44307 | 44289 | 44295 | 44309 |
|---|---|---|---|---|---|---|---|---|---|
| E2H | $23.24 \pm 5.58$ | $19.90 \pm 0.42$ | $\mathbf{25.98 \pm 2.14}$ | $32.19 \pm 8.72$ | $44.97 \pm 13.70$ | $32.00 \pm 6.04$ | $20.71 \pm 1.33$ | $22.11 \pm 3.77$ | $20.55 \pm 1.53$ |
| H2E | $\mathbf{32.89 \pm 2.58}$ | $\mathbf{24.39 \pm 3.12}$ | $20.06 \pm 0.10$ | $\mathbf{46.65 \pm 4.88}$ | $\mathbf{75.63 \pm 5.58}$ | $\mathbf{47.71 \pm 2.78}$ | $24.63 \pm 2.56$ | $\mathbf{25.05 \pm 1.91}$ | $\mathbf{24.32 \pm 0.92}$ |
| Domain-based | $29.54 \pm 3.78$ | $21.11 \pm 1.53$ | $22.11 \pm 2.74$ | $39.16 \pm 2.90$ | $62.43 \pm 1.79$ | $42.87 \pm 3.49$ | $\mathbf{25.11 \pm 1.30}$ | $24.98 \pm 1.39$ | $24.27 \pm 1.47$ |

| | Manufacturing | | | Human Actions | | | OCR | | |
| | 44288 | 44294 | 44304 | 44284 | 44291 | 44301 | 44287 | 44296 | 44310 |
|---|---|---|---|---|---|---|---|---|---|
| E2H | $41.16 \pm 10.09$ | $24.45 \pm 1.20$ | $35.51 \pm 10.26$ | $27.85 \pm 1.09$ | $24.37 \pm 1.07$ | $25.56 \pm 1.49$ | $20.39 \pm 0.39$ | $20.62 \pm 0.24$ | $20.27 \pm 0.47$ |
| H2E | $\mathbf{74.38 \pm 5.67}$ | $\mathbf{39.26 \pm 2.77}$ | $\mathbf{87.25 \pm 2.45}$ | $\mathbf{49.01 \pm 5.67}$ | $\mathbf{32.20 \pm 2.84}$ | $\mathbf{44.26 \pm 6.96}$ | $19.68 \pm 0.05$ | $20.46 \pm 0.13$ | $\mathbf{21.12 \pm 0.10}$ |
| Domain-based | $61.90 \pm 4.76$ | $35.06 \pm 1.84$ | $66.22 \pm 8.91$ | $41.41 \pm 0.35$ | $30.00 \pm 2.17$ | $38.37 \pm 3.86$ | $\mathbf{20.46 \pm 0.08}$ | $\mathbf{20.88 \pm 0.23}$ | $20.90 \pm 0.23$ |

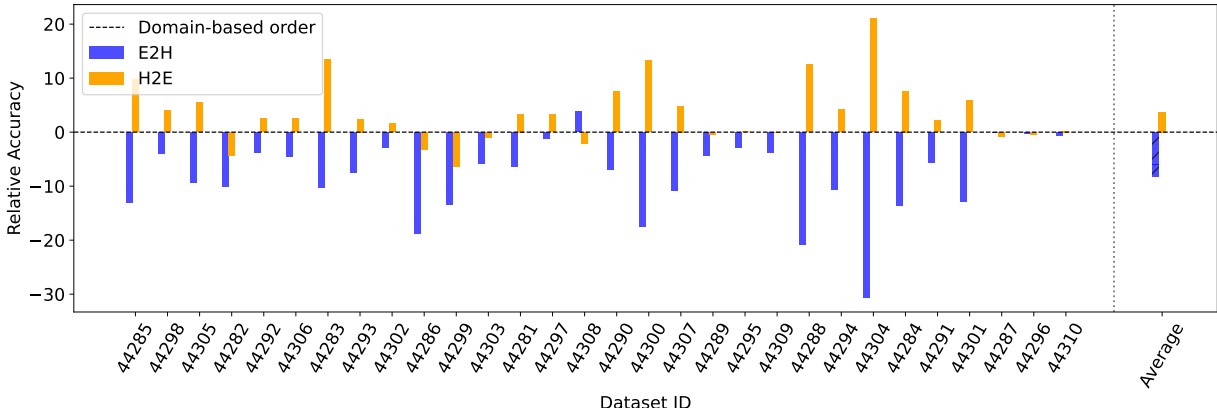

Figure 13: Relative validation accuracy of different TL curricula when the feature extractor is trained from scratch. The trend is equivalent to the one shown in Sect. 6.3.1.

## A.4 Optimal transport curricula

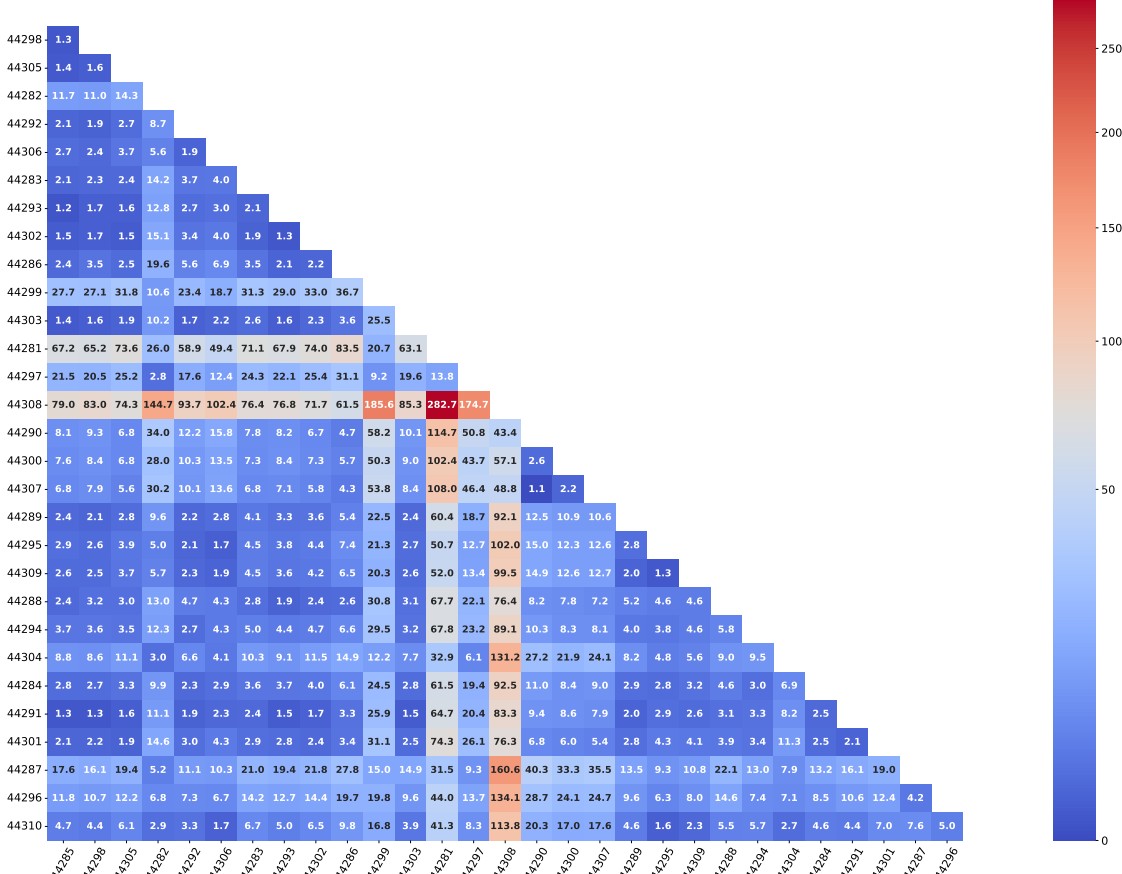

Figure 14: Heatmap representing the dataset similarity for all datasets in Meta-Album Mini computed with OTDD (Alvarez-Melis & Fusi, 2020). The lower the number the closer/more similar are the datasets.

Using OTDD (Alvarez-Melis & Fusi, 2020), we construct three curricula for our experiments based on the dataset distance in Fig. 14:

- *Easy-to-Easy* (E2E): [44304, 44310, 44295, 44309, 44306, 44292, 44303, 44285, 44293, 44302, 44305, 44298, 44291, 44289, 44301, 44284, 44294, 44283, 44288, 44286, 44307, 44290, 44300, 44296, 44287, 44282, 44297, 44299, 44281, 44308];

- *Hard-to-Hard* (H2H): [44304, 44308, 44281, 44290, 44299, 44307, 44297, 44300, 44287, 44286, 44296, 44288, 44282, 44302, 44310, 44301, 44306, 44294, 44283, 44309, 44305, 44295, 44293, 44284, 44289, 44303, 44292, 44285, 44298, 44291];

- *Switch* (Switch): [44304, 44308, 44290, 44281, 44297, 44307, 44300, 44299, 44282, 44286, 44293, 44287, 44296, 44288, 44302, 44310, 44295, 44283, 44285, 44294, 44292, 44301, 44305, 44306, 44309, 44284, 44291, 44289, 44298, 44303].

In addition to reporting the distance values, Fig. 15 visualizes the dataset similarity relationships. The x-axis represents the starting dataset, while the y-axis orders all other datasets from most similar (bottom) to most dissimilar (top). Colors indicate the domain to which each dataset belongs. As previously mentioned, distances are computed using the Micro size of the datasets rather than Mini. However, since the model is trained and evaluated on Mini, we report only the Mini dataset IDs for simplicity. The corresponding dataset IDs for both the Micro and Mini size of Meta-Album are listed in Tab. 8.

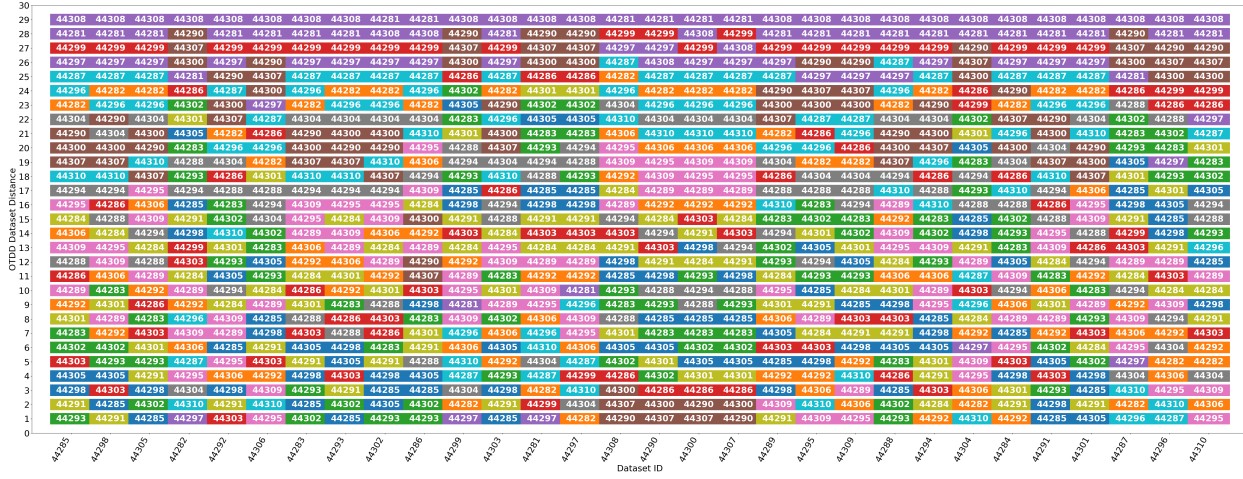

Figure 15: Dataset similarity for all datasets in Meta-Album Mini computed with OTDD (Alvarez-Melis & Fusi, 2020). A column is assigned to each dataset and it shows the dataset IDs ordered from the easiest/similar (bottom) to the most difficult/dissimilar (top) dataset. Datasets with the same colors are associated with the same domain: blue for Large Animals, orange for Small Animals, green for Plants, red for Plant Diseases, purple for Microscopy, brown for Remote Sensing, pink for Vehicles, gray for Manufacturing, yellow for Human Actions, light blue for OCR.

Table 8: Dataset IDs for Micro and Mini sizes of Meta-Album.

| Domain | Micro dataset IDs | | | Mini dataset IDs | | |
|---|---|---|---|---|---|---|
| Large Animals | 44241 | 44313 | 44275 | 44285 | 44298 | 44305 |
| Small Animals | 44238 | 44248 | 44276 | 44282 | 44292 | 44306 |
| Plants | 44239 | 44249 | 44272 | 44283 | 44293 | 44302 |
| Plant Diseases | 44242 | 44314 | 44273 | 44286 | 44299 | 44303 |
| Microscopy | 44237 | 44312 | 44278 | 44281 | 44297 | 44308 |
| Remote Sensing | 44246 | 44315 | 44277 | 44290 | 44300 | 44307 |
| Vehicles | 44245 | 44251 | 44279 | 44289 | 44295 | 44309 |
| Manufacturing | 44244 | 44250 | 44274 | 44288 | 44294 | 44304 |
| Human Actions | 44240 | 44247 | 44271 | 44284 | 44291 | 44301 |
| OCR | 44243 | 44252 | 44280 | 44287 | 44296 | 44310 |

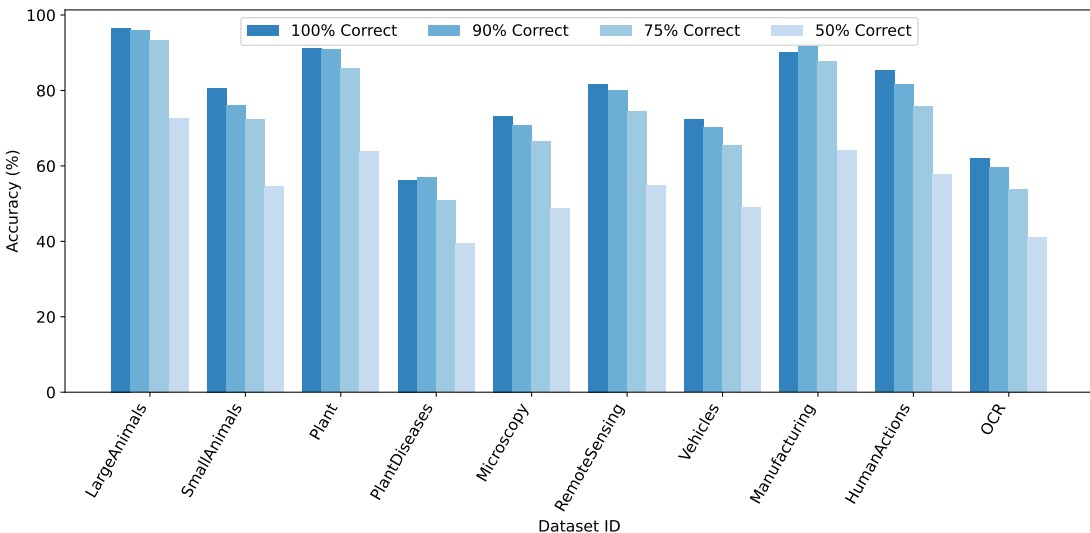

Figure 16: Model robustness to input-label mapping perturbations by varying the proportion of correctly labeled examples in the demonstrations 100-90-75-50% (corresponding to 0-2-6-12 mislabeled examples) at test time. Only the datasets in the first release of Meta-Album Mini are shown for simplicity.

## A.5    Robustness to label noise

A key challenge in evaluating GEOM is understanding its reliance on input-label mappings in the demonstrations to perform a task. In real-world scenarios, mislabeling errors or label noise during pre-processing, as well as challenges in assigning correct labels to certain samples, can lead to incorrect input-label mappings. To simulate this, we introduce perturbations in the input-label mapping for a subset of examples, varying the proportion of correctly labeled instances in the test task context. The results, illustrated in Fig. 16, reveal that the model remains robust to label perturbation even when only 75% of the labels in the task context are correct. This aligns with the findings in Min et al. (2022b), suggesting that meta-training with an explicit in-context learning objective encourages the model to rely less on the input-label mapping and instead leverage other aspects of the demonstrations to make predictions. The complete results are reported in Tab. 15.

Additionally, we examine the effects of applying label perturbations exclusively during the training phase. The results indicate that the model effectively exploits the task context for test time predictions rather than relying on memorized input-label mappings from training. Indeed, if the model were memorizing erroneous mappings, this would result in significant performance degradation during testing, which is not observed in Tab. 16. Interestingly, introducing minor label perturbations (e.g., 10% of the demonstrations) during training acts as a form of regularization (Guo et al., 2021), improving the model's ability to generalize across domains, even with more challenging tasks.

## A.6 Additional results - Offline learning

Table 9: Comparison between GEOM trained on Meta-Album Mini and on Meta-Album Extended. The training is performed following the LOO setting described in Sect. 5, and the performance is evaluated on the datasets from the left-out domain. The dataset IDs differ between the Mini and Extended sizes, and they are reported here as they appear in the Meta-Album website (Ullah et al., 2022). OCR is not part of the Extended size of Meta-Album. Results show the average across three complete runs of the algorithms.

| | Large Animals | | | Small Animals | | | Plants | | | Plant Diseases | | |
|---|---|---|---|---|---|---|---|---|---|---|---|---|
| | 44285 | 44298 | 44305 | 44282 | 44292 | 44306 | 44283 | 44293 | 44302 | 44286 | 44299 | 44303 |
| Mini | 73.34 ± 1.34 | 63.03 ± 3.03 | 76.22 ± 1.62 | 78.05 ± 0.75 | 52.34 ± 0.75 | 55.72 ± 0.35 | 78.38 ± 1.22 | 51.14 ± 0.74 | 37.92 ± 0.54 | 78.35 ± 1.06 | 87.75 ± 0.76 | 58.02 ± 0.76 |
| | 44320 | 44331 | 44338 | 44317 | 44326 | 44340 | 44318 | 44327 | 44335 | 44321 | 44332 | 44336 |
| Extended | 73.15 ± 1.86 | 59.44 ± 2.38 | 67.61 ± 3.19 | 76.24 ± 1.78 | 51.95 ± 1.12 | 54.67 ± 1.79 | 78.35 ± 0.19 | 51.98 ± 1.27 | 38.57 ± 0.25 | 78.01 ± 0.47 | 85.49 ± 0.89 | 58.6 ± 0.6 |

| | Microscopy | | | Remote Sensing | | | Vehicles | | |
|---|---|---|---|---|---|---|---|---|---|
| | 44281 | 44297 | 44308 | 44290 | 44300 | 44307 | 44289 | 44295 | 44309 |
| Mini | 79.41 ± 0.55 | 30.64 ± 0.56 | 31.53 ± 0.27 | 69.74 ± 0.62 | 82.28 ± 1.33 | 68.45 ± 1.56 | 42.39 ± 1.29 | 57.11 ± 0.52 | 36.78 ± 0.94 |
| | 4316 | 44330 | 44342 | 44324 | 44333 | 44341 | 44323 | 44329 | 44343 |
| Extended | 77.30 ± 0.59 | 31.98 ± 0.34 | 32.01 ± 0.92 | 68.50 ± 0.40 | 85.58 ± 0.43 | 67.58 ± 0.49 | 43.97 ± 1.98 | 47.66 ± 0.39 | 36.71 ± 0.98 |

| | Manufacturing | | | Human Actions | | | OCR | | |
|---|---|---|---|---|---|---|---|---|---|
| | 44288 | 44294 | 44304 | 44284 | 44291 | 44301 | 44287 | 44296 | 44310 |
| Mini | 73.05 ± 0.99 | 56.34 ± 0.72 | 87.36 ± 0.69 | 72.66 ± 1.00 | 55.94 ± 1.61 | 53.50 ± 2.97 | 30.47 ± 0.31 | 26.68 ± 0.47 | 39.16 ± 0.22 |
| | 44322 | 44328 | 44337 | 44319 | 44325 | 44334 | - | - | - |
| Extended | 92.11 ± 0.60 | 61.62 ± 0.37 | 96.91 ± 0.18 | 74.00 ± 2.36 | 55.33 ± 2.57 | 55.06 ± 4.15 | - | - | - |

Table 10: Performance comparison among GEOM, GEOM-M, and GEOM-IN across all Meta-Album (Mini) datasets. The training is performed following the LOO setting described in Sect. 5 (for GEOM and GEOM-M) and on ImageNet-1k (Deng et al., 2009) for GEOM-IN. The performance is then evaluated on the Meta-Album datasets in the left-out domain. The bold font highlights the best-performing approach for each dataset. Results show the average across three complete runs of the algorithms.

| | Large Animals | | | Small Animals | | | Plants | | | Plant Diseases | | |
|---|---|---|---|---|---|---|---|---|---|---|---|---|
| | 44285 | 44298 | 44305 | 44282 | 44292 | 44306 | 44283 | 44293 | 44302 | 44286 | 44299 | 44303 |
| GEOM | 73.34 ± 1.34 | 63.03 ± 3.03 | 76.22 ± 1.62 | 78.05 ± 0.75 | 52.34 ± 0.75 | 55.72 ± 0.35 | **78.38 ± 1.22** | **51.14 ± 0.74** | **37.92 ± 0.54** | **78.35 ± 1.06** | **87.75 ± 0.76** | **58.02 ± 0.76** |
| GEOM-M | 71.77 ± 0.41 | 63.97 ± 0.36 | 68.38 ± 0.30 | **78.37 ± 0.56** | 51.25 ± 0.86 | 54.09 ± 1.50 | 76.57 ± 1.51 | 47.16 ± 2.00 | 36.35 ± 0.26 | 77.16 ± 0.88 | 86.65 ± 1.13 | 57.71 ± 0.29 |
| GEOM-IN | **90.33 ± 0.44** | **98.49 ± 0.10** | **95.88 ± 0.02** | 74.29 ± 0.71 | **55.14 ± 0.43** | **62.98 ± 0.81** | 75.14 ± 1.35 | 48.25 ± 1.50 | 37.54 ± 1.52 | 67.53 ± 2.59 | 80.11 ± 4.00 | 47.46 ± 0.64 |

| | Microscopy | | | Remote Sensing | | | Vehicles | | |
|---|---|---|---|---|---|---|---|---|---|
| | 44281 | 44297 | 44308 | 44290 | 44300 | 44307 | 44289 | 44295 | 44309 |
| GEOM | **79.41 ± 0.55** | **30.64 ± 0.56** | **31.53 ± 0.27** | **69.74 ± 0.62** | **82.28 ± 1.33** | **68.45 ± 1.56** | 42.39 ± 1.29 | **57.11 ± 0.52** | 36.78 ± 0.94 |
| GEOM-M | 78.67 ± 0.66 | 30.64 ± 0.92 | 30.36 ± 0.55 | 67.96 ± 0.66 | 81.58 ± 0.43 | 66.11 ± 1.59 | 45.47 ± 1.20 | 57.44 ± 0.59 | 35.19 ± 0.08 |
| GEOM-IN | 71.78 ± 1.25 | 30.81 ± 0.64 | 30.82 ± 0.63 | 68.37 ± 1.01 | 79.33 ± 2.84 | 67.42 ± 2.47 | **57.04 ± 0.61** | 52.83 ± 1.80 | **46.08 ± 0.60** |

| | Manufacturing | | | Human Actions | | | OCR | | |
|---|---|---|---|---|---|---|---|---|---|
| | 44288 | 44294 | 44304 | 44284 | 44291 | 44301 | 44287 | 44296 | 44310 |
| GEOM | 73.05 ± 0.99 | 56.34 ± 0.72 | 87.36 ± 0.69 | 72.66 ± 1.00 | 55.94 ± 1.61 | 53.50 ± 2.97 | 30.47 ± 0.31 | 26.68 ± 0.47 | 39.16 ± 0.22 |
| GEOM-M | 71.32 ± 0.18 | 58.33 ± 0.31 | 85.51 ± 0.26 | 74.24 ± 0.18 | 55.76 ± 0.32 | 55.72 ± 0.54 | 31.14 ± 0.36 | 26.77 ± 0.26 | 39.92 ± 0.26 |
| GEOM-IN | **85.52 ± 1.41** | **66.66 ± 0.80** | **94.93 ± 0.80** | **88.91 ± 0.63** | **82.46 ± 0.76** | **67.44 ± 0.40** | **31.86 ± 0.81** | **29.17 ± 0.06** | **41.63 ± 1.10** |

Table 11: Performance comparison among GEOM and GEOM-IN across all Meta-Album (Mini) datasets. The pre-trained weights are inherited from CLIP Radford et al. (2021) and the `timm`[3] library. The training is performed following the LOO setting described in Sect. 5 and on ImageNet-1k (Deng et al., 2009) for GEOM-IN. The performance is then evaluated on the Meta-Album datasets in the left-out domain. The bold font highlights the best-performing approach for each dataset. Results show the average across three complete runs of the algorithms.

| | Large Animals | | | Small Animals | | | | Plants | | | Plant Diseases | | |
| | 44285 | 44298 | 44305 | 44282 | 44292 | 44306 | 44283 | 44293 | 44302 | 44286 | 44299 | 44303 |
|---|---|---|---|---|---|---|---|---|---|---|---|---|
| GEOM | $84.78 \pm 1.58$ | $53.20 \pm 5.18$ | $86.79 \pm 1.89$ | $\mathbf{69.82 \pm 0.47}$ | $46.53 \pm 0.98$ | $69.13 \pm 0.87$ | $91.26 \pm 0.55$ | $\mathbf{68.46 \pm 1.18}$ | $\mathbf{29.60 \pm 1.45}$ | $\mathbf{45.34 \pm 6.01}$ | $\mathbf{70.57 \pm 3.86}$ | $\mathbf{49.95 \pm 1.41}$ |
| GEOM-IN | $\mathbf{96.25 \pm 0.09}$ | $\mathbf{91.37 \pm 0.81}$ | $\mathbf{98.71 \pm 0.30}$ | $64.77 \pm 1.46$ | $\mathbf{52.56 \pm 1.12}$ | $\mathbf{77.32 \pm 0.41}$ | $\mathbf{93.52 \pm 0.43}$ | $65.01 \pm 4.78$ | $27.33 \pm 4.56$ | $34.49 \pm 7.53$ | $60.25 \pm 8.61$ | $47.72 \pm 5.49$ |

| | Microscopy | | | Remote Sensing | | | Vehicles | | |
| | 44281 | 44297 | 44308 | 44290 | 44300 | 44307 | 44289 | 44295 | 44309 |
|---|---|---|---|---|---|---|---|---|---|
| GEOM | $\mathbf{46.01 \pm 5.53}$ | $\mathbf{24.17 \pm 1.89}$ | $\mathbf{23.29 \pm 3.28}$ | $77.36 \pm 4.71$ | $84.38 \pm 0.98$ | $64.27 \pm 2.91$ | $74.25 \pm 3.89$ | $51.59 \pm 5.46$ | $45.30 \pm 1.58$ |
| GEOM-IN | $29.25 \pm 7.24$ | $22.86 \pm 2.97$ | $20.57 \pm 2.87$ | $\mathbf{89.67 \pm 2.57}$ | $\mathbf{86.31 \pm 2.09}$ | $\mathbf{80.25 \pm 0.39}$ | $\mathbf{88.63 \pm 1.54}$ | $\mathbf{52.89 \pm 7.80}$ | $\mathbf{69.71 \pm 2.77}$ |

| | Manufacturing | | | Human Actions | | | OCR | | |
| | 44288 | 44294 | 44304 | 44284 | 44291 | 44301 | 44287 | 44296 | 44310 |
|---|---|---|---|---|---|---|---|---|---|
| GEOM | $\mathbf{84.87 \pm 2.32}$ | $\mathbf{75.26 \pm 3.05}$ | $91.11 \pm 0.98$ | $61.09 \pm 3.99$ | $44.17 \pm 2.56$ | $40.16 \pm 12.94$ | $28.84 \pm 0.97$ | $30.83 \pm 0.53$ | $36.96 \pm 0.38$ |
| GEOM-IN | $82.87 \pm 6.41$ | $72.36 \pm 2.15$ | $\mathbf{94.92 \pm 0.33}$ | $\mathbf{97.41 \pm 0.48}$ | $\mathbf{88.23 \pm 0.94}$ | $\mathbf{73.44 \pm 4.87}$ | $\mathbf{33.58 \pm 1.57}$ | $\mathbf{35.82 \pm 3.25}$ | $\mathbf{40.03 \pm 2.11}$ |

Table 12: Performance comparison between GEOM and GEOM-M across all Meta-Album (Mini) datasets in the 5-ways 1-shot setting ($N = 5, K = 1$). The training is performed following the LOO setting described in Sect. 5. The performance is then evaluated on the Meta-Album datasets in the left-out domain. The bold font highlights the best-performing approach for each dataset. Results show the average across three complete runs of the algorithms.

| | Large Animals | | | Small Animals | | | | Plants | | | Plant Diseases | | |
| | 44285 | 44298 | 44305 | 44282 | 44292 | 44306 | 44283 | 44293 | 44302 | 44286 | 44299 | 44303 |
|---|---|---|---|---|---|---|---|---|---|---|---|---|
| GEOM | $\mathbf{55.69 \pm 1.59}$ | $\mathbf{48.01 \pm 1.90}$ | $\mathbf{58.13 \pm 1.52}$ | $62.87 \pm 0.83$ | $\mathbf{37.45 \pm 0.63}$ | $\mathbf{40.47 \pm 0.48}$ | $\mathbf{59.39 \pm 1.11}$ | $\mathbf{36.58 \pm 0.34}$ | $\mathbf{28.91 \pm 0.50}$ | $\mathbf{62.14 \pm 0.30}$ | $\mathbf{76.88 \pm 0.79}$ | $\mathbf{42.56 \pm 0.89}$ |
| GEOM-M | $54.55 \pm 0.54$ | $47.53 \pm 0.73$ | $51.21 \pm 0.29$ | $\mathbf{63.61 \pm 1.02}$ | $35.39 \pm 0.90$ | $37.77 \pm 1.06$ | $57.02 \pm 1.55$ | $33.44 \pm 0.68$ | $27.79 \pm 0.30$ | $59.79 \pm 1.38$ | $75.53 \pm 0.31$ | $40.63 \pm 0.38$ |

| | Microscopy | | | Remote Sensing | | | Vehicles | | |
| | 44281 | 44297 | 44308 | 44290 | 44300 | 44307 | 44289 | 44295 | 44309 |
|---|---|---|---|---|---|---|---|---|---|
| GEOM | $\mathbf{66.44 \pm 1.35}$ | $\mathbf{25.43 \pm 0.80}$ | $\mathbf{25.73 \pm 0.06}$ | $\mathbf{49.79 \pm 1.40}$ | $\mathbf{64.66 \pm 1.30}$ | $\mathbf{49.80 \pm 1.29}$ | $32.71 \pm 0.58$ | $\mathbf{41.35 \pm 0.63}$ | $\mathbf{28.17 \pm 1.02}$ |
| GEOM-M | $65.80 \pm 0.87$ | $25.16 \pm 0.39$ | $25.21 \pm 0.34$ | $48.41 \pm 1.30$ | $63.33 \pm 0.66$ | $47.33 \pm 1.51$ | $\mathbf{34.13 \pm 0.69}$ | $39.92 \pm 0.80$ | $27.66 \pm 0.82$ |

| | Manufacturing | | | Human Actions | | | OCR | | |
| | 44288 | 44294 | 44304 | 44284 | 44291 | 44301 | 44287 | 44296 | 44310 |
|---|---|---|---|---|---|---|---|---|---|
| GEOM | $\mathbf{64.31 \pm 0.78}$ | $39.29 \pm 0.97$ | $\mathbf{78.79 \pm 1.33}$ | $53.45 \pm 0.91$ | $\mathbf{39.68 \pm 1.16}$ | $\mathbf{38.11 \pm 1.19}$ | $24.63 \pm 0.02$ | $23.07 \pm 0.34$ | $\mathbf{29.56 \pm 0.37}$ |
| GEOM-M | $63.77 \pm 1.01$ | $\mathbf{40.31 \pm 0.66}$ | $77.59 \pm 0.20$ | $\mathbf{54.90 \pm 0.22}$ | $38.88 \pm 0.78$ | $37.94 \pm 0.33$ | $\mathbf{24.79 \pm 0.13}$ | $\mathbf{23.11 \pm 0.14}$ | $29.29 \pm 0.11$ |

---

[3]https://timm.fast.ai/

Table 13: Performance of GEOM vs GEOM-M when models are tested on GEOM-M-like tasks, i.e., a task can include classes from different datasets and domain. The training is performed following the LOO setting described in Sect. 5. The name of the domain is the one excluded during the training and test phases. The only exception is Oracle, which have access to all the training classes of all the 30 datasets in Meta-Album. Results show the average across three complete runs of the algorithms.

|  | Large Animals | Small Animals | Plants | Plant Diseases | Microscopy |
|---|---|---|---|---|---|
| GEOM | $91.33 \pm 0.15$ | $92.86 \pm 0.30$ | $93.35 \pm 0.20$ | $93.26 \pm 0.19$ | $93.43 \pm 0.19$ |
| GEOM-M | $\mathbf{92.86 \pm 0.09}$ | $\mathbf{93.94 \pm 0.25}$ | $\mathbf{93.87 \pm 0.04}$ | $\mathbf{94.29 \pm 0.09}$ | $\mathbf{94.29 \pm 0.13}$ |
| Oracle | $98.59 \pm 0.09$ | $98.59 \pm 0.09$ | $98.59 \pm 0.09$ | $98.59 \pm 0.09$ | $98.59 \pm 0.09$ |

|  | Remote Sensing | Vehicles | Manufacturing | Human Actions | OCR |
|---|---|---|---|---|---|
| GEOM | $93.07 \pm 0.38$ | $92.82 \pm 0.43$ | $92.01 \pm 0.34$ | $93.27 \pm 0.24$ | $\mathbf{98.94 \pm 0.09}$ |
| GEOM-M | $\mathbf{94.11 \pm 0.15}$ | $\mathbf{93.84 \pm 0.26}$ | $\mathbf{93.47 \pm 0.15}$ | $\mathbf{94.17 \pm 0.14}$ | $98.66 \pm 0.06$ |
| Oracle | $98.59 \pm 0.09$ | $98.59 \pm 0.09$ | $98.59 \pm 0.09$ | $98.59 \pm 0.09$ | $98.59 \pm 0.09$ |

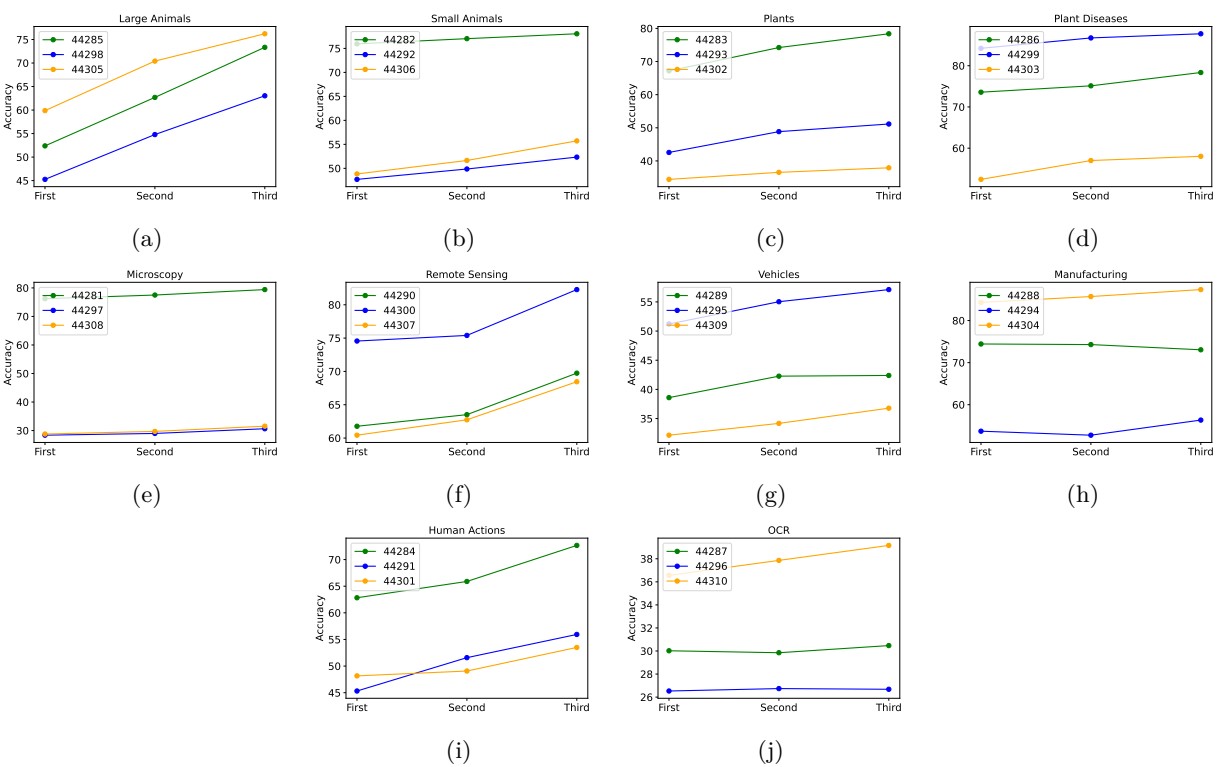

Figure 17: Comparison of GEOM training only on datasets from the first release (*First*, 9 datasets), on datasets from the first and second releases (*Second*, 18 datasets), and on datasets from all three releases (*Third*, 27 datasets) of Meta-Album Mini. The training is performed following the LOO setting described in Sect. 5, and the performance is evaluated on the datasets from the left-out domain (represented with blue, orange, and green colors).

Table 14: Comparison of GEOM training only on datasets from the first release (*First*, 9 datasets), on datasets from the first and second releases (*Second*, 18 datasets), and on datasets from all three releases (*Third*, 27 datasets) of Meta-Album Mini. The training is performed following the LOO setting described in Sect. 5, and the performance is evaluated on the datasets from the left-out domain. Results show the average across three complete runs of the algorithms.

| | Large Animals | | | Small Animals | | | Plants | | | Plant Diseases | | |
| | 44285 | 44298 | 44305 | 44282 | 44292 | 44306 | 44283 | 44293 | 44302 | 44286 | 44299 | 44303 |
|---|---|---|---|---|---|---|---|---|---|---|---|---|
| *First* | $52.38 \pm 1.97$ | $45.26 \pm 2.18$ | $59.88 \pm 2.18$ | $75.94 \pm 0.22$ | $47.70 \pm 0.72$ | $48.83 \pm 2.99$ | $67.18 \pm 1.43$ | $42.56 \pm 0.71$ | $34.43 \pm 2.06$ | $73.58 \pm 0.89$ | $84.22 \pm 1.13$ | $52.43 \pm 0.61$ |
| *Second* | $62.69 \pm 0.46$ | $54.79 \pm 1.96$ | $70.41 \pm 0.57$ | $77.03 \pm 0.80$ | $49.86 \pm 0.75$ | $51.64 \pm 0.38$ | $74.22 \pm 0.98$ | $48.84 \pm 0.85$ | $36.55 \pm 1.27$ | $75.12 \pm 0.78$ | $86.74 \pm 0.45$ | $57.00 \pm 0.13$ |
| *Third* | $73.34 \pm 1.34$ | $63.03 \pm 3.03$ | $76.22 \pm 1.62$ | $78.05 \pm 0.75$ | $52.34 \pm 0.75$ | $55.72 \pm 0.35$ | $78.38 \pm 1.22$ | $51.14 \pm 0.74$ | $37.92 \pm 0.54$ | $78.35 \pm 1.06$ | $87.75 \pm 0.76$ | $58.02 \pm 0.76$ |

| | Microscopy | | | Remote Sensing | | | Vehicles | | |
| | 44281 | 44297 | 44308 | 44290 | 44300 | 44307 | 44289 | 44295 | 44309 |
|---|---|---|---|---|---|---|---|---|---|
| *First* | $76.23 \pm 0.54$ | $28.38 \pm 0.53$ | $28.79 \pm 0.74$ | $61.77 \pm 0.71$ | $74.56 \pm 1.10$ | $60.43 \pm 1.17$ | $38.59 \pm 1.46$ | $51.21 \pm 0.32$ | $32.15 \pm 0.08$ |
| *Second* | $77.50 \pm 1.66$ | $28.99 \pm 0.32$ | $29.71 \pm 0.36$ | $63.51 \pm 0.39$ | $75.40 \pm 0.89$ | $62.74 \pm 0.78$ | $42.27 \pm 1.04$ | $55.03 \pm 0.58$ | $34.17 \pm 1.12$ |
| *Third* | $79.41 \pm 0.55$ | $30.64 \pm 0.56$ | $31.53 \pm 0.27$ | $69.74 \pm 0.62$ | $82.28 \pm 1.33$ | $68.45 \pm 1.56$ | $42.39 \pm 1.29$ | $57.11 \pm 0.52$ | $36.78 \pm 0.94$ |

| | Manufacturing | | | Human Actions | | | OCR | | |
| | 44288 | 44294 | 44304 | 44284 | 44291 | 44301 | 44287 | 44296 | 44310 |
|---|---|---|---|---|---|---|---|---|---|
| *First* | $74.43 \pm 2.10$ | $53.70 \pm 0.81$ | $84.30 \pm 0.71$ | $62.83 \pm 3.06$ | $45.32 \pm 4.32$ | $48.16 \pm 3.27$ | $30.02 \pm 1.25$ | $26.53 \pm 0.92$ | $36.56 \pm 1.02$ |
| *Second* | $74.29 \pm 2.21$ | $52.75 \pm 0.30$ | $85.71 \pm 1.66$ | $65.89 \pm 1.35$ | $51.58 \pm 3.31$ | $49.07 \pm 2.58$ | $29.85 \pm 0.54$ | $26.74 \pm 0.69$ | $37.86 \pm 0.87$ |
| *Third* | $73.05 \pm 0.99$ | $56.34 \pm 0.72$ | $87.36 \pm 0.69$ | $72.66 \pm 1.00$ | $55.94 \pm 1.61$ | $53.50 \pm 2.97$ | $30.47 \pm 0.31$ | $26.68 \pm 0.47$ | $39.16 \pm 0.22$ |

Table 15: Model robustness to input-label mapping perturbations by varying the proportion of correctly labeled examples in the demonstrations 100-90-75-50% (corresponding to 0-2-6-12 mislabeled examples) at test time. The model is trained on all the Meta-Album datasets and the evaluation is performed on the test set of each dataset. Results show the average across three complete runs of the algorithms.

| | Large Animals | | | Small Animals | | | Plants | | | Plant Diseases | | |
| | 44285 | 44298 | 44305 | 44282 | 44292 | 44306 | 44283 | 44293 | 44302 | 44286 | 44299 | 44303 |
|---|---|---|---|---|---|---|---|---|---|---|---|---|
| 100% correct | $96.53 \pm 0.09$ | $95.54 \pm 0.31$ | $95.50 \pm 0.59$ | $80.70 \pm 0.32$ | $56.99 \pm 0.40$ | $75.04 \pm 0.77$ | $91.18 \pm 0.54$ | $59.47 \pm 1.06$ | $35.37 \pm 2.06$ | $56.26 \pm 2.18$ | $86.69 \pm 2.42$ | $49.49 \pm 1.68$ |
| 90% correct | $95.92 \pm 0.22$ | $95.07 \pm 0.39$ | $94.77 \pm 1.66$ | $76.17 \pm 0.24$ | $53.83 \pm 0.16$ | $73.77 \pm 1.15$ | $90.88 \pm 0.15$ | $55.51 \pm 0.38$ | $32.26 \pm 1.70$ | $57.05 \pm 3.33$ | $83.30 \pm 2.47$ | $50.14 \pm 0.50$ |
| 75% correct | $93.37 \pm 0.31$ | $91.79 \pm 0.35$ | $90.12 \pm 2.01$ | $72.42 \pm 0.44$ | $47.72 \pm 0.79$ | $67.87 \pm 1.30$ | $85.90 \pm 0.23$ | $51.53 \pm 1.70$ | $31.02 \pm 0.19$ | $50.80 \pm 2.95$ | $76.49 \pm 1.80$ | $46.16 \pm 0.66$ |
| 50% correct | $72.56 \pm 1.31$ | $70.44 \pm 2.29$ | $67.14 \pm 0.79$ | $54.67 \pm 1.67$ | $35.84 \pm 1.13$ | $51.16 \pm 0.26$ | $63.96 \pm 1.08$ | $42.46 \pm 3.35$ | $27.80 \pm 0.08$ | $39.60 \pm 2.38$ | $63.82 \pm 4.21$ | $37.75 \pm 0.61$ |

| | Microscopy | | | Remote Sensing | | | Vehicles | | |
| | 44281 | 44297 | 44308 | 44290 | 44300 | 44307 | 44289 | 44295 | 44309 |
|---|---|---|---|---|---|---|---|---|---|
| 100% correct | $73.23 \pm 0.59$ | $32.43 \pm 2.78$ | $30.74 \pm 0.78$ | $81.74 \pm 0.66$ | $94.29 \pm 0.34$ | $75.17 \pm 2.15$ | $72.23 \pm 0.41$ | $71.21 \pm 1.81$ | $51.53 \pm 2.36$ |
| 90% correct | $70.88 \pm 1.77$ | $31.21 \pm 2.50$ | $29.88 \pm 0.58$ | $80.17 \pm 0.73$ | $93.66 \pm 0.29$ | $72.87 \pm 1.78$ | $70.19 \pm 0.06$ | $70.13 \pm 0.84$ | $57.84 \pm 1.99$ |
| 75% correct | $66.42 \pm 1.69$ | $28.53 \pm 1.12$ | $27.90 \pm 2.00$ | $74.37 \pm 0.37$ | $89.18 \pm 0.48$ | $68.36 \pm 1.86$ | $65.37 \pm 0.27$ | $60.93 \pm 1.35$ | $53.42 \pm 0.75$ |
| 50% correct | $48.80 \pm 1.00$ | $26.62 \pm 1.40$ | $24.66 \pm 2.34$ | $54.76 \pm 0.13$ | $68.65 \pm 1.11$ | $50.84 \pm 0.73$ | $49.12 \pm 0.61$ | $46.57 \pm 6.51$ | $39.76 \pm 1.54$ |

| | Manufacturing | | | Human Actions | | | OCR | | |
| | 44288 | 44294 | 44304 | 44284 | 44291 | 44301 | 44287 | 44296 | 44310 |
|---|---|---|---|---|---|---|---|---|---|
| 100% correct | $90.04 \pm 2.56$ | $72.89 \pm 1.05$ | $98.34 \pm 0.15$ | $85.26 \pm 2.55$ | $68.86 \pm 1.55$ | $57.33 \pm 2.59$ | $61.98 \pm 0.25$ | $57.03 \pm 0.27$ | $72.60 \pm 0.42$ |
| 90% correct | $91.58 \pm 2.97$ | $69.38 \pm 1.25$ | $98.00 \pm 0.16$ | $81.54 \pm 1.80$ | $67.93 \pm 1.26$ | $57.01 \pm 2.53$ | $59.59 \pm 0.16$ | $54.53 \pm 0.19$ | $70.42 \pm 0.58$ |
| 75% correct | $87.71 \pm 2.59$ | $63.97 \pm 2.28$ | $96.62 \pm 0.27$ | $75.68 \pm 2.40$ | $60.69 \pm 2.09$ | $53.24 \pm 0.99$ | $53.85 \pm 0.22$ | $49.01 \pm 0.79$ | $64.54 \pm 0.34$ |
| 50% correct | $64.10 \pm 0.85$ | $46.28 \pm 0.87$ | $77.06 \pm 0.05$ | $57.74 \pm 1.04$ | $44.26 \pm 0.57$ | $37.18 \pm 1.74$ | $41.11 \pm 0.06$ | $37.43 \pm 0.66$ | $48.29 \pm 0.32$ |

Table 16: Model robustness to input-label mapping perturbations by varying the proportion of correctly labeled examples in the demonstrations 100-90-75-50% (corresponding to 0-2-6-12 mislabeled examples) at training time. The model is trained on all the Meta-Album datasets and the evaluation is performed on the test set of each dataset. Results show the average across three complete runs of the algorithms.

| | Large Animals | | | Small Animals | | | | Plants | | | Plant Diseases | |
|---|---|---|---|---|---|---|---|---|---|---|---|---|
| | 44285 | 44298 | 44305 | 44282 | 44292 | 44306 | 44283 | 44293 | 44302 | 44286 | 44299 | 44303 |
| 100% correct | $96.53 \pm 0.09$ | $95.54 \pm 0.31$ | $95.50 \pm 0.59$ | $80.70 \pm 0.32$ | $56.99 \pm 0.40$ | $75.04 \pm 0.77$ | $91.18 \pm 0.54$ | $59.47 \pm 1.06$ | $35.37 \pm 2.06$ | $56.26 \pm 2.18$ | $86.69 \pm 2.42$ | $49.49 \pm 1.68$ |
| 90% correct | $96.42 \pm 0.24$ | $95.41 \pm 0.28$ | $94.73 \pm 0.85$ | $78.26 \pm 0.18$ | $55.51 \pm 1.06$ | $75.36 \pm 0.99$ | $91.24 \pm 0.32$ | $60.48 \pm 1.55$ | $34.08 \pm 0.89$ | $59.12 \pm 2.28$ | $86.72 \pm 2.59$ | $51.22 \pm 0.89$ |
| 75% correct | $96.34 \pm 0.08$ | $95.20 \pm 0.39$ | $95.01 \pm 1.35$ | $77.65 \pm 0.34$ | $55.52 \pm 0.78$ | $75.26 \pm 1.62$ | $90.89 \pm 0.51$ | $58.77 \pm 2.82$ | $33.56 \pm 2.70$ | $59.66 \pm 1.54$ | $83.74 \pm 1.61$ | $51.40 \pm 2.15$ |
| 50% correct | $94.65 \pm 0.29$ | $89.82 \pm 1.93$ | $91.19 \pm 1.33$ | $73.87 \pm 0.13$ | $50.23 \pm 1.22$ | $71.09 \pm 0.62$ | $87.43 \pm 0.29$ | $46.66 \pm 3.63$ | $30.14 \pm 3.51$ | $49.44 \pm 2.73$ | $78.35 \pm 6.52$ | $45.38 \pm 2.62$ |

| | Microscopy | | | Remote Sensing | | | Vehicles | | |
|---|---|---|---|---|---|---|---|---|---|
| | 44281 | 44297 | 44308 | 44290 | 44300 | 44307 | 44289 | 44295 | 44309 |
| 100% correct | $73.23 \pm 0.59$ | $32.43 \pm 2.78$ | $30.74 \pm 0.78$ | $81.74 \pm 0.66$ | $94.29 \pm 0.34$ | $75.17 \pm 2.15$ | $72.23 \pm 0.41$ | $71.21 \pm 1.81$ | $51.53 \pm 2.36$ |
| 90% correct | $74.43 \pm 1.67$ | $32.55 \pm 2.92$ | $28.50 \pm 1.59$ | $82.48 \pm 0.62$ | $94.44 \pm 0.40$ | $75.94 \pm 1.66$ | $72.01 \pm 0.15$ | $69.79 \pm 4.42$ | $58.94 \pm 1.73$ |
| 75% correct | $71.79 \pm 1.10$ | $32.37 \pm 2.25$ | $27.80 \pm 2.32$ | $81.75 \pm 0.70$ | $95.04 \pm 0.65$ | $74.10 \pm 2.24$ | $71.77 \pm 0.58$ | $72.24 \pm 0.94$ | $55.20 \pm 2.98$ |
| 50% correct | $63.74 \pm 6.08$ | $29.70 \pm 3.45$ | $26.46 \pm 2.05$ | $75.49 \pm 2.23$ | $92.99 \pm 0.46$ | $68.88 \pm 1.48$ | $67.46 \pm 1.06$ | $54.17 \pm 4.67$ | $41.64 \pm 1.46$ |

| | Manufacturing | | | Human Actions | | | OCR | | |
|---|---|---|---|---|---|---|---|---|---|
| | 44288 | 44294 | 44304 | 44284 | 44291 | 44301 | 44287 | 44296 | 44310 |
| 100% correct | $90.04 \pm 2.56$ | $72.89 \pm 1.05$ | $98.34 \pm 0.15$ | $85.26 \pm 2.55$ | $68.86 \pm 1.55$ | $57.33 \pm 2.59$ | $61.98 \pm 0.25$ | $57.03 \pm 0.27$ | $72.60 \pm 0.42$ |
| 90% correct | $92.94 \pm 0.74$ | $72.29 \pm 1.94$ | $98.55 \pm 0.18$ | $82.85 \pm 2.40$ | $70.77 \pm 0.81$ | $60.04 \pm 1.32$ | $61.76 \pm 0.66$ | $56.05 \pm 0.18$ | $71.83 \pm 0.38$ |
| 75% correct | $92.63 \pm 1.81$ | $72.85 \pm 1.45$ | $98.45 \pm 0.22$ | $83.48 \pm 1.47$ | $70.76 \pm 1.80$ | $61.88 \pm 1.13$ | $60.03 \pm 0.25$ | $54.32 \pm 0.16$ | $70.43 \pm 0.34$ |
| 50% correct | $92.14 \pm 1.06$ | $67.55 \pm 5.38$ | $97.64 \pm 0.23$ | $78.02 \pm 1.19$ | $56.03 \pm 4.78$ | $47.20 \pm 4.70$ | $53.31 \pm 0.67$ | $47.47 \pm 0.47$ | $64.34 \pm 0.84$ |

## A.7 Additional results - Sequential learning

Table 17: Comparative results of GEOM-S assigning to each dataset the same number of epochs (*static*) or a proportion dependent on the size of each dataset (*proportional*). The *offline* baseline can be seen as an oracle baseline as all datasets are available simultaneously during training (non-sequential approach). In this setting, the training split of each dataset is used for sampling training tasks, while the performance is evaluated on the test split, as described in Sect. 6. Results show the average across three complete runs of the algorithms.

| | Large Animals | | | Small Animals | | | | Plants | | | Plant Diseases | |
|---|---|---|---|---|---|---|---|---|---|---|---|---|
| | 44285 | 44298 | 44305 | 44282 | 44292 | 44306 | 44283 | 44293 | 44302 | 44286 | 44299 | 44303 |
| Static | $95.34 \pm 0.04$ | $94.73 \pm 0.54$ | $92.60 \pm 1.94$ | $76.83 \pm 0.77$ | $51.75 \pm 0.62$ | $72.97 \pm 0.79$ | $89.80 \pm 0.85$ | $52.23 \pm 2.44$ | $36.64 \pm 1.32$ | $59.92 \pm 1.14$ | $82.34 \pm 0.72$ | $48.04 \pm 2.18$ |
| Proportional | $95.77 \pm 0.33$ | $95.06 \pm 0.25$ | $94.71 \pm 1.22$ | $75.75 \pm 0.46$ | $52.10 \pm 0.48$ | $73.74 \pm 0.48$ | $89.66 \pm 0.76$ | $54.02 \pm 2.81$ | $33.35 \pm 1.30$ | $55.81 \pm 0.77$ | $81.90 \pm 1.55$ | $47.57 \pm 1.11$ |
| Offline | $96.53 \pm 0.09$ | $95.54 \pm 0.31$ | $95.50 \pm 0.59$ | $80.70 \pm 0.32$ | $56.99 \pm 0.40$ | $75.04 \pm 0.77$ | $91.18 \pm 0.54$ | $59.47 \pm 1.06$ | $35.37 \pm 2.06$ | $56.26 \pm 2.18$ | $86.69 \pm 2.42$ | $49.49 \pm 1.68$ |

| | Microscopy | | | Remote Sensing | | | Vehicles | | |
|---|---|---|---|---|---|---|---|---|---|
| | 44281 | 44297 | 44308 | 44290 | 44300 | 44307 | 44289 | 44295 | 44309 |
| Static | $71.66 \pm 1.39$ | $31.29 \pm 0.84$ | $30.05 \pm 1.79$ | $79.55 \pm 0.86$ | $94.00 \pm 1.24$ | $70.65 \pm 0.74$ | $71.46 \pm 0.11$ | $70.65 \pm 1.91$ | $57.81 \pm 1.59$ |
| Proportional | $72.27 \pm 1.28$ | $31.33 \pm 0.26$ | $27.56 \pm 1.32$ | $76.30 \pm 1.26$ | $90.74 \pm 1.57$ | $70.40 \pm 1.11$ | $71.98 \pm 0.32$ | $71.68 \pm 0.74$ | $59.44 \pm 0.98$ |
| Offline | $73.23 \pm 0.59$ | $32.43 \pm 2.78$ | $30.74 \pm 0.78$ | $81.74 \pm 0.66$ | $94.29 \pm 0.34$ | $75.17 \pm 2.15$ | $72.23 \pm 0.41$ | $71.21 \pm 1.81$ | $51.53 \pm 2.36$ |

| | Manufacturing | | | Human Actions | | | OCR | | |
|---|---|---|---|---|---|---|---|---|---|
| | 44288 | 44294 | 44304 | 44284 | 44291 | 44301 | 44287 | 44296 | 44310 |
| Static | $93.69 \pm 0.34$ | $74.72 \pm 1.38$ | $98.61 \pm 0.07$ | $81.51 \pm 1.30$ | $69.14 \pm 1.42$ | $58.14 \pm 5.26$ | $39.13 \pm 0.61$ | $30.65 \pm 0.14$ | $47.23 \pm 1.03$ |
| Proportional | $87.56 \pm 0.52$ | $70.45 \pm 1.14$ | $96.87 \pm 0.59$ | $81.11 \pm 1.93$ | $64.34 \pm 1.24$ | $57.71 \pm 3.05$ | $63.52 \pm 0.25$ | $62.75 \pm 0.30$ | $73.57 \pm 0.23$ |
| Offline | $90.04 \pm 2.56$ | $72.89 \pm 1.05$ | $98.34 \pm 0.15$ | $85.26 \pm 2.55$ | $68.86 \pm 1.55$ | $57.33 \pm 2.59$ | $61.98 \pm 0.25$ | $57.03 \pm 0.27$ | $72.60 \pm 0.42$ |

Table 18: Accuracy results of GEOM-S using different TL-based curricula: *easy-to-hard* (E2H), *hard-to-easy* (H2E), and domain-based order. The same number of epochs (20) is assigned to each dataset, using the *static* approach in Sect. 6.1. The bold font highlights the best-performing approach for each dataset. Results show the average across three complete runs of the algorithms.

| | Large Animals | | | Small Animals | | | | Plants | | | Plant Diseases | |
| | 44285 | 44298 | 44305 | 44282 | 44292 | 44306 | 44283 | 44293 | 44302 | 44286 | 44299 | 44303 |
|---|---|---|---|---|---|---|---|---|---|---|---|---|
| E2H | **96.00 ± 0.27** | 95.24 ± 0.21 | 92.63 ± 0.85 | 76.15 ± 0.38 | 55.14 ± 0.29 | 74.78 ± 0.46 | 88.38 ± 0.86 | 53.98 ± 2.85 | 35.54 ± 0.76 | 54.38 ± 1.18 | 82.00 ± 3.22 | 46.97 ± 1.42 |
| H2E | 95.97 ± 0.31 | **95.47 ± 0.41** | **95.55 ± 0.80** | **79.18 ± 0.54** | **56.28 ± 0.61** | **76.66 ± 0.48** | **91.21 ± 0.51** | **57.03 ± 1.32** | 35.98 ± 3.08 | **63.19 ± 1.55** | **88.17 ± 1.36** | **50.16 ± 1.49** |
| Domain-based | 95.34 ± 0.48 | 94.73 ± 0.54 | 92.60 ± 1.94 | 76.83 ± 0.77 | 51.75 ± 0.62 | 72.97 ± 0.79 | 89.80 ± 0.85 | 52.23 ± 2.44 | **36.64 ± 1.32** | 59.92 ± 1.14 | 82.34 ± 0.72 | 48.04 ± 2.18 |

| | Microscopy | | | Remote Sensing | | | Vehicles | | |
| | 44281 | 44297 | 44308 | 44290 | 44300 | 44307 | 44289 | 44295 | 44309 |
|---|---|---|---|---|---|---|---|---|---|
| E2H | 66.29 ± 4.38 | 27.93 ± 1.12 | 27.31 ± 2.24 | 79.14 ± 0.99 | 85.89 ± 1.00 | 63.12 ± 1.07 | 71.10 ± 0.49 | 68.78 ± 3.12 | 57.23 ± 5.69 |
| H2E | **73.80 ± 2.71** | **31.52 ± 1.47** | **30.17 ± 0.43** | **83.22 ± 1.30** | **95.13 ± 0.72** | **73.12 ± 1.18** | **72.53 ± 0.60** | **72.88 ± 3.38** | **62.37 ± 1.26** |
| Domain-based | 71.66 ± 1.39 | 31.29 ± 0.84 | 30.05 ± 1.79 | 79.55 ± 0.86 | 94.00 ± 1.24 | 70.65 ± 0.74 | 71.46 ± 0.11 | 70.65 ± 1.91 | 57.81 ± 1.59 |

| | Manufacturing | | | Human Actions | | | OCR | | |
| | 44288 | 44294 | 44304 | 44284 | 44291 | 44301 | 44287 | 44296 | 44310 |
|---|---|---|---|---|---|---|---|---|---|
| E2H | 83.46 ± 4.97 | 61.43 ± 2.69 | 96.49 ± 0.92 | 74.94 ± 0.67 | 52.97 ± 4.69 | 55.87 ± 4.51 | **50.10 ± 0.24** | 37.05 ± 0.50 | 58.67 ± 0.61 |
| H2E | 91.85 ± 1.58 | **75.94 ± 0.49** | 97.50 ± 0.03 | **84.11 ± 2.10** | **71.10 ± 1.24** | **62.81 ± 1.76** | 49.51 ± 0.59 | **41.04 ± 0.62** | **62.17 ± 0.60** |
| Domain-based | **93.69 ± 0.34** | 74.72 ± 1.38 | **98.61 ± 0.07** | 81.51 ± 1.30 | 69.14 ± 1.42 | 58.14 ± 5.26 | 39.13 ± 0.61 | 30.65 ± 0.14 | 47.23 ± 1.03 |

Table 19: Accuracy results of GEOM-S using different OT-based curricula: *easy-to-easy* (E2E), *hard-to-hard* (H2H), *Switch*, and the domain-based order. The same number of epochs (20) is assigned to each dataset, using the *static* approach in Sect. 6.1. The bold font highlights the best-performing approach for each dataset. Results show the average across three complete runs of the algorithms.

| | Large Animals | | | Small Animals | | | | Plants | | | Plant Diseases | |
| | 44285 | 44298 | 44305 | 44282 | 44292 | 44306 | 44283 | 44293 | 44302 | 44286 | 44299 | 44303 |
|---|---|---|---|---|---|---|---|---|---|---|---|---|
| E2E | **96.94 ± 0.22** | **95.90 ± 0.07** | **94.88 ± 0.16** | **78.37 ± 0.36** | 53.80 ± 0.59 | 74.88 ± 0.60 | 91.18 ± 0.71 | 55.93 ± 1.92 | 35.16 ± 0.90 | 57.27 ± 3.11 | **88.00 ± 2.58** | 47.74 ± 0.54 |
| H2H | 92.66 ± 0.37 | 87.00 ± 0.56 | 94.05 ± 1.31 | 77.44 ± 1.06 | 55.16 ± 0.18 | 75.41 ± 0.75 | **91.44 ± 0.54** | **57.25 ± 3.34** | 35.89 ± 0.65 | 60.12 ± 3.71 | 87.36 ± 1.79 | **50.14 ± 1.71** |
| Switch | 96.45 ± 0.12 | 91.19 ± 3.71 | 94.85 ± 1.45 | 77.85 ± 0.69 | **56.84 ± 0.58** | **76.28 ± 0.52** | 91.02 ± 0.34 | 53.90 ± 2.34 | 36.55 ± 2.30 | **61.70 ± 2.82** | 85.31 ± 1.36 | 45.39 ± 1.35 |
| Domain-based | 95.34 ± 0.48 | 94.73 ± 0.54 | 92.60 ± 1.94 | 76.83 ± 0.77 | 51.75 ± 0.62 | 72.97 ± 0.79 | 89.80 ± 0.85 | 52.23 ± 2.44 | **36.64 ± 1.32** | 59.92 ± 1.14 | 82.34 ± 0.72 | 48.04 ± 2.18 |

| | Microscopy | | | Remote Sensing | | | Vehicles | | |
| | 44281 | 44297 | 44308 | 44290 | 44300 | 44307 | 44289 | 44295 | 44309 |
|---|---|---|---|---|---|---|---|---|---|
| E2E | **71.02 ± 1.28** | **31.47 ± 2.47** | 29.19 ± 2.16 | **82.30 ± 0.96** | **94.30 ± 0.54** | **72.13 ± 0.80** | **72.30 ± 0.84** | 73.31 ± 2.09 | **61.78 ± 1.33** |
| H2H | 70.21 ± 2.44 | 29.09 ± 0.06 | **30.05 ± 1.57** | 78.85 ± 1.47 | 93.04 ± 0.36 | 71.55 ± 1.09 | 69.76 ± 1.06 | 72.21 ± 1.43 | 60.51 ± 3.47 |
| Switch | 70.49 ± 2.31 | 30.06 ± 0.63 | 29.12 ± 1.29 | 78.21 ± 1.88 | 91.81 ± 1.46 | 70.73 ± 2.32 | 62.88 ± 0.33 | **73.57 ± 0.67** | 59.42 ± 0.57 |
| Domain-based | 71.66 ± 1.39 | 31.29 ± 0.84 | 30.05 ± 1.79 | 79.55 ± 0.86 | 94.00 ± 1.24 | 70.65 ± 0.74 | 71.46 ± 0.11 | 70.65 ± 1.91 | 57.81 ± 1.59 |

| | Manufacturing | | | Human Actions | | | OCR | | |
| | 44288 | 44294 | 44304 | 44284 | 44291 | 44301 | 44287 | 44296 | 44310 |
|---|---|---|---|---|---|---|---|---|---|
| E2E | 88.54 ± 1.91 | 68.71 ± 0.77 | 97.83 ± 0.95 | 82.89 ± 0.31 | 67.62 ± 0.76 | **62.70 ± 2.10** | **55.76 ± 0.33** | 49.26 ± 0.56 | 69.69 ± 0.76 |
| H2H | **89.77 ± 3.40** | 70.86 ± 0.48 | 98.03 ± 0.22 | 82.82 ± 1.23 | 62.92 ± 2.38 | 59.21 ± 2.35 | 51.17 ± 0.26 | **50.93 ± 0.36** | **69.82 ± 0.68** |
| Switch | 89.36 ± 0.08 | **71.85 ± 0.95** | **98.34 ± 0.10** | **84.01 ± 1.60** | **69.62 ± 1.88** | 59.98 ± 0.88 | 52.55 ± 0.61 | 49.90 ± 0.67 | 68.24 ± 0.72 |
| Domain-based | **93.69 ± 0.34** | 74.72 ± 1.38 | **98.61 ± 0.07** | 81.51 ± 1.30 | 69.14 ± 1.42 | 58.14 ± 5.26 | 39.13 ± 0.61 | 30.65 ± 0.14 | 47.23 ± 1.03 |

## A.8 Additional results - Unsupervised learning

Table 20: Comparison between GEOM-U and CAMeLU (Vettoruzzo et al., 2025). GEOM-U is trained with the LOO approach described in Sect. 5 on Meta-Album Mini removing the class labels during training, while CAMeLU is trained on ImageNet-1k. The bold font highlights the best-performing approach for each dataset. Results show the average across three complete runs of the algorithms.

| | Large Animals | | | | Small Animals | | | Plants | | | | Plant Diseases | |
| | 44285 | 44298 | 44305 | 44282 | 44292 | 44306 | 44283 | 44293 | 44302 | 44286 | 44299 | 44303 |
|---|---|---|---|---|---|---|---|---|---|---|---|---|
| GEOM-U | $84.49 \pm 0.53$ | $78.43 \pm 1.30$ | $83.43 \pm 1.06$ | $\mathbf{84.70 \pm 0.12}$ | $\mathbf{58.51 \pm 0.38}$ | $\mathbf{66.71 \pm 0.50}$ | $\mathbf{90.10 \pm 0.14}$ | $\mathbf{60.34 \pm 0.51}$ | $\mathbf{45.04 \pm 0.29}$ | $\mathbf{87.47 \pm 0.63}$ | $\mathbf{92.74 \pm 0.29}$ | $\mathbf{62.31 \pm 0.62}$ |
| CAMeLU | $\mathbf{90.69 \pm 0.19}$ | $\mathbf{96.34 \pm 0.16}$ | $\mathbf{93.03 \pm 0.29}$ | $80.28 \pm 0.34$ | $56.93 \pm 0.27$ | $62.09 \pm 0.88$ | $82.25 \pm 0.28$ | $52.13 \pm 0.49$ | $41.34 \pm 0.91$ | $81.01 \pm 0.20$ | $87.56 \pm 1.53$ | $55.54 \pm 0.41$ |

| | Microscopy | | | Remote Sensing | | | Vehicles | | |
| | 44281 | 44297 | 44308 | 44290 | 44300 | 44307 | 44289 | 44295 | 44309 |
|---|---|---|---|---|---|---|---|---|---|
| GEOM-U | $\mathbf{81.97 \pm 0.41}$ | $\mathbf{34.40 \pm 0.56}$ | $\mathbf{34.30 \pm 0.58}$ | $\mathbf{80.22 \pm 0.64}$ | $\mathbf{92.31 \pm 0.24}$ | $\mathbf{78.49 \pm 0.74}$ | $\mathbf{61.58 \pm 0.59}$ | $\mathbf{57.32 \pm 0.38}$ | $\mathbf{46.55 \pm 0.57}$ |
| CAMeLU | $81.45 \pm 0.09$ | $33.65 \pm 0.26$ | $34.01 \pm 0.47$ | $79.60 \pm 0.36$ | $91.57 \pm 0.10$ | $78.02 \pm 0.37$ | $53.31 \pm 0.23$ | $54.10 \pm 0.47$ | $43.11 \pm 1.11$ |

| | Manufacturing | | | Human Actions | | | OCR | | |
| | 44288 | 44294 | 44304 | 44284 | 44291 | 44301 | 44287 | 44296 | 44310 |
|---|---|---|---|---|---|---|---|---|---|
| GEOM-U | $\mathbf{97.49 \pm 0.08}$ | $74.97 \pm 1.08$ | $\mathbf{99.32 \pm 0.05}$ | $89.20 \pm 0.38$ | $77.27 \pm 0.50$ | $\mathbf{72.57 \pm 0.21}$ | $\mathbf{38.27 \pm 0.10}$ | $\mathbf{32.60 \pm 0.30}$ | $\mathbf{45.74 \pm 0.18}$ |
| CAMeLU | $95.81 \pm 0.45$ | $\mathbf{76.62 \pm 0.53}$ | $98.99 \pm 0.19$ | $\mathbf{90.52 \pm 0.22}$ | $\mathbf{79.82 \pm 0.25}$ | $72.20 \pm 0.61$ | $29.06 \pm 0.39$ | $27.04 \pm 0.36$ | $39.27 \pm 0.44$ |

## A.9 Miscellanea

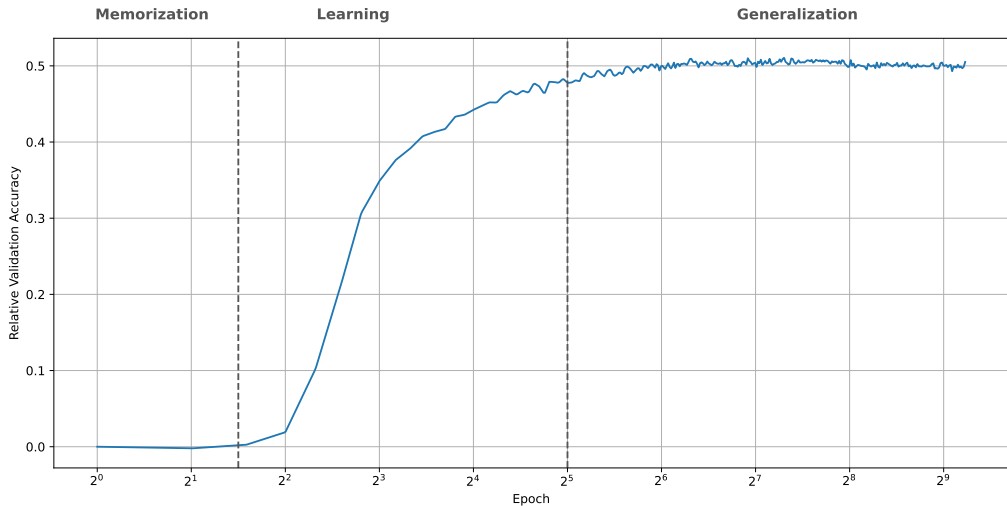

Figure 18: Visualization of GEOM learning trend represented as the relative validation accuracy, computed as the accuracy relative to its initial value, over 600 epochs. For every epoch, 50 tasks are sampled from the test split of each datasets (resulting in 1500 tasks/epoch) and the mean validation accuracy is computed over all tasks. The curve shows three different phases in the learning trend: memorization, learning, and generalization. These findings are consistend with the observations in Kirsch et al. (2022) and Vettoruzzo et al. (2025)

