# OpenReview forum: "Generalization over Memorization in In-Context Learning"
_TMLR — Rejected by TMLR_

### Review · Reviewer_7FoW · 2025-04-08

**Summary Of Contributions:**

This paper systematically investigates meta-learning for in-context learning (ICL) using the Meta-Album and ImageNet datasets. It demonstrates that dataset construction, the number of classes, and the number of images in meta-learning impact ICL performance. Additionally, it explores the influence of data ordering in sequential learning and compares various curriculum learning pipelines designed to promote generalization-over-memorization.

**Audience:**

Yes

**Broader Impact Concerns:**

No specific Broader Impact Concerns

**Claims And Evidence:**

Yes

**Requested Changes:**

The paper advances meaningful insights into meta-learning for in-context learning (ICL). Its contributions would be strengthened through further clarification of its core claims, particularly through more controlled comparisons between the use of multiple smaller, domain-specific datasets versus vast, uncurated language corpora.The comparison between Meta-Album and ImageNet seems not sufficiently support this claim and, more importantly, seems not enough to provide clear guidelines for dataset design, especially given that some experiments in the paper also rely on ImageNet pretraining. Additionally, deeper exploration of the interplay between memorization and generalization would enhance the paper’s theoretical grounding and practical implications. While generalization versus memorization is a compelling topic, the paper discusses it very little. Does the LOO and Sequential Evaluation mechanism inherently imply generalization over memorization?

Minor Questions:
1.	Are all GEOM-S models in Section 6 pretrained on ImageNet prior to sequential learning? If not, could the observed conclusion (e.g., H2E > E2H) being different with or w/o pretraining?

2.	The experiments consistently use an N=K=5 (N-way K-shot) configuration. Why was this setting chosen? Would varying N and K influence the results?

3.	Could the authors comment on the applicability of their conclusions to causal transformers, which may be of broader interest to the community?

Some references that might be related to the topics including burstiness, ICL, and generalization:

Santoro A, Bartunov S, Botvinick M, Wierstra D, Lillicrap T. Meta-learning with memory-augmented neural networks
Kirsch L, Harrison J, Sohl-Dickstein J, Metz L. General-purpose in-context learning by meta-learning transformers
Raparthy SC, Hambro E, Kirk R, Henaff M, Raileanu R. Generalization to new sequential decision making tasks with in-context learning
Zhao Y, Qu Y, Staniszewski K, Tworkowski S, Liu W, Miłoś P, Wu Y, Minervini P. Analysing the impact of sequence composition on language model pre-training.
Wang F, Lin C, Cao Y, Kang Y. Benchmarking General-Purpose In-Context Learning

**Strengths And Weaknesses:**

I found several discoveries in the paper to be interesting and valuable to the community. For instance, in Section 6.2, the analysis reveals positive BWT values, indicating that the model’s performance on previously seen domains improves as it encounters new datasets. In Section 6.3.1, the comparison between H2E and E2H pipelines highlights how curricula organization affects ICL performance.
A main claim of the paper is that training on multiple smaller, domain-specific datasets fosters generalization. However, this claim requires further clarification. The concept of "multiple smaller, domain-specific datasets" appears to contrast with "vast, uncurated language corpora," but these approaches share notable similarities. A clearer distinction between these methodologies is needed to strengthen the argument.
Section 5.1 demonstrates the superiority of GEOM-M over GEOM, which appears closely tied to the concept of "burstiness," a feature widely recognized as critical for incentivizing ICL. Section 5.2 emphasizes the importance of class diversity over intra-class variability, a point that has been noted in prior works. Adding relevant references here would strengthen the discussion.
Nonetheless, I find limited evidence supporting the claim that "multiple smaller, domain-specific datasets" outperform "vast, uncurated language corpora." The role of burstiness depends on sequence construction, and increasing class diversity or intra-class variability may contradict this assertion. Furthermore, GEOM-IN often surpasses GEOM and GEOM-M in performance, and comparisons between GEOM-S and offline training suggest that accessing all data simultaneously may outperform curriculum or sequential learning. In Section 7, where Meta-Album outperforms ImageNet in unsupervised training, this outcome does not directly validate the superiority of smaller, domain-specific datasets, although it validates the superiority of Meta Album to ImageNet to some extent.

---

> ### Author Response · Authors · 2025-04-20
> **Response to reviewer 7FoW (part 1)**
>
> We thank the reviewer for the time spent reading and analysing our work, and for joining a constructive discussion.
>
> ### About large-scale dataset and organized collection
> We agree that the phrasing of the claim could benefit from clarification, which we have now added in the Introduction, in Section 4 and in Section 5.1. Our primary comparison was meant to compare a collection that preserved boundaries (GEOM) and one that did not (GEOM-M), in a way that emulates a large-scale vision dataset. Indeed, they share the same number of classes and images, in order to ensure a fair comparison. We eventually included a well-known large-scale dataset, i.e., ImageNet-1k, as a baseline, due to its wide use in the vision literature. Experimental results in Section 5.1 demonstrates that keeping the datasets separated (as in GEOM) results in good performance while offering other advantages, such as modularity, easier maintenance, better adaptability, and ethical manageability, as stated in Section 5 and in the Conclusions. Indeed, our intention is not to suggest that multiple small, domain-specific datasets *universally outperform* large, uncurated corpora, but rather that they can match or even exceed their performance in some cases while offering additional practical advantages. In fact, all the other sections in the paper support this claim: for instance, it allows to add more datasets as they become available (Section 5.2), it allows to train the model sequentially (Section 6), and to build a curriculum (Section 6.3). All of this is not possible by considering a single, large datasets.
>
> ### About the concept of "burstiness"
> We appreciate the comment of the reviewer and we have added a brief discussion on the concept of “burstiness” in Section 5.1. Specifically, in GEOM, “burstiness” arises from the structure of the training data, which is composed of domain-specific image groups (or clusters) rather than being uniformly sampled. This clustered organization leads to tasks that exhibit bursty distributions. In contrast, GEOM-M constructs training tasks by uniformly sampling from the merged Meta-Album datasets, resulting in “less-bursty” tasks. The burstiness in GEOM is beneficial as it promotes in-context learning [1,2,3,4] and supports strong generalization performance.
>
> ### Adding references class diversity vs inter-class variability
> We thank the reviewer for the recommendations and we have now added the suggested references in the Related Work section [5,6,7] and in Section 5.2 [3, 4].
>
> ### Requested changes: GEOM and ImageNet-1k pre-training
> To overcome the limitations of the image feature extractor based on ImageNet-1k features, we conducted experiments to prove the effectiveness of our methodology: we added a GEOM vs GEOM-IN comparison with CLIP feature extractor (Tab.11 in Appendix A.6) and a comparison of the TL-based curricula with a clean ResNet50 feature extractor jointly trained with the transformer encoder.
> Finally, future work may evaluate our model on other benchmarks. Our decision to focus on Meta-Album was due to its well-curated nature, domain diversity, and dataset splits, which allow us to simulate a range of different scenarios, by keeping the computational resources controlled.
>
> ### Requested changes: Generalization vs memorization
> When highlighting the generalization abilities of GEOM, we cover different aspects: (1) in the LOO case the model is exposed to a test set of domains that were never seen in the training phase and, indeed, if the model is evaluated on classes that belong to new domains during the first iterations, the performance improvements are close to zero, because it is still memorizing examples that it has encountered. However, the learning trend suddenly and abruptly fosters the learning of how to solve N-way K-shot tasks and finally generalizes to unseen classes and domains. This behaviour is shown in Fig. 18 in the appendix. (2) The study of forgetting in the Sequential case is another proof of how the model still solves new tasks (same domains that it has already seen, but unknown classes) even when that particular domain is located far in the training time. Moreover, the performance even shows an increase in the BWT values, highlighting the fact that increasing further knowledge fosters the generalization to unseen examples. It might be objected that the model retains the memory, however, the number of datasets is very high (30) and, in any case, the test classes will still differ from the training classes. These are the reasons why we claim that the model can generalize rather than memorize. We also extended the concept of generalization to the unsupervised case, where the model has to learn from images that are unlabelled to generalize to the same few-shot inference scenario. We also refer to [10] as the concept of memorization vs generalization.

---

> ### Author Response · Authors · 2025-04-20
> **Response to reviewer 7FoW (part 2 + list of references)**
>
> ### Requested changes: about pre-training and curricula
> Only the feature extractor is frozen on the features learned from ImageNet-1k, as witnessed in prior work (GPICL [6], CAML [11], CAMeLU [12]). The transformer is always learned from scratch. However, it was an interesting point to evaluate the effectiveness of our curricula by introducing this modification to our model. Therefore, to accomplish the reviewer’s interest, we ran the same experiment, with the very same architecture, yet we let the weights of ResNet50 be learned together with the transformer. Although with an expected decrease in the absolute value of the validation accuracy, the relative performance between E2H and H2E, compared with the baseline, remains aligned with the original findings. These are shown in Fig. 13 in the Appendix.
>
> ### About the choice of N=5, K=5
> We chose the 5-way 5-shot setting primarily as the standard and most adopted practice in the meta-learning and few-shot learning literature [13, 14, 15, 16],. To demonstrate the generality of our approach, we have now included additional results for the 5-way 1-shot scenario (N=5, K=1) in Table 10 and Section 5.1. These results confirm that our method extends naturally to other N-way K-shot configurations.
>
> ### About: causal transformers
> Our current approach, GEOM, is grounded in a meta-learning formulation that assumes permutation invariance within tasks, i.e., there is no inherent ordering among the context samples. This assumption aligns with the use of a non-causal transformer, as also mentioned in related work [11, 12], which similarly exploits a non-causal transformer for few-shot classification [17].
> At the time of writing, there are at least a couple of interesting papers [18, 6] that extend this approach on images to causal transformers. Although none of them has been accepted yet, indicating that there might still be room for improvement in this direction, they certainly provide a promising prospect and we plan to investigate it as future work. We have added further details in the Conclusions section of our paper.
>
> ### References
> [1] Stephanie Chan, et al. 2022. Data distributional properties drive emergent in-context learning in transformers. In NeurIPS.
>
>
> [2] Singh, Aaditya, et al. The transient nature of emergent in-context learning in transformers. Advances in Neural Information Processing Systems 36 (2023): 27801-27819.
>
>
> [3] Zhao, Yu, et al. "Analysing the impact of sequence composition on language model pre-training." arXiv preprint arXiv:2402.13991 (2024)
>
>
> [4] Chan, Bryan, et al. "Toward Understanding In-context vs. In-weight Learning." The Thirteenth International Conference on Learning Representations.
>
>
> [5] Santoro, Adam, et al. "Meta-learning with memory-augmented neural networks." International conference on machine learning. PMLR, 2016.
>
>
> [6] Kirsch, Louis, et al. "General-purpose in-context learning by meta-learning transformers." arXiv preprint arXiv:2212.04458 (2022).
>
>
> [7] Wang, Fan, et al. "Benchmarking General-Purpose In-Context Learning." arXiv preprint arXiv:2405.17234 (2024).
>
> [8] Raparthy, Sharath Chandra, et al. "Generalization to New Sequential Decision Making Tasks with In-Context Learning." International Conference on Machine Learning. PMLR, 2024.
>
>
> [9] Zhao Y, et al. “Analysing the impact of sequence composition on language model pre-training.”
>
>
> [10] Radford, A., et al. (2019). Language models are unsupervised multitask learners. OpenAI blog, 1(8), 9.
>
>
> [11] Fifty, Christopher, et al. "Context-Aware Meta-Learning." The Twelfth International Conference on Learning Representations.
>
>
> [12] Vettoruzzo, Anna, et al. "Unsupervised Meta-Learning via In-Context Learning."  The Twelfth International Conference on Learning Representations. 2025.
>
>
> [13] Vinyals, O., et al. (2016). Matching networks for one shot learning. Advances in neural information processing systems, 29.
>
>
> [14] Chelsea Finn, et al. (2017). Model-Agnostic Meta-Learning for Fast Adaptation of Deep Networks. In Proceedings of the 34th International Conference on Machine Learning (pp. 1126–1135). PMLR.
>
>
> [15] Jake Snell, et al. (2017). Prototypical Networks for Few-shot Learning. CoRR, abs/1703.05175.
>
>
> [16] Sung, F., et al. (2018). Learning to compare: Relation network for few-shot learning. In Proceedings of the IEEE conference on computer vision and pattern recognition (pp. 1199–1208).
>
>
> [17] Hollmann, F. (2025). Accurate predictions on small data with a tabular foundation model. Nature, 637(8045), 319-326.
>
>
> [18] Bratulić, Jelena, et al. "What Matters for In-Context Learning: A Balancing Act of Look-up and In-Weight Learning." arXiv preprint arXiv:2501.06256 (2025).

---

### Review · Reviewer_fgjE · 2025-04-10

**Summary Of Contributions:**

This paper explores training strategies for in-context learners. Notably, it aims to understand how the organization of the training dataset can influence the generalization ability of in-context learners (ICL). Therefore, it introduces a framework (GEOM) for training ICL with Meta-Learning methodologies. Most importantly, the authors want to show that the generalization ability of ICL can be influenced by the way the training dataset is given to the model and they demonstrate through experiments that a small number of well-organized datasets can lead to better generalization abilities than a single large unorganized training dataset. Such experiments are conducted on offline, sequential, and unsupervised scenarios.

**Audience:**

Yes

**Broader Impact Concerns:**

No concern

**Claims And Evidence:**

No

**Requested Changes:**

# Critical
- I want to see experiments with GEOM-M, and similarly ImageNet-1k with a Merged and Sequential case.
- I want confirmation that the fact that GEOM outperforms GEOM-M is not just a problem of convergence. I admit that this is so far my opinion and I did not find elements in the current version of the paper that contradict this opinion.
- I want to see experiments where the model is trained from scratch
- Experiments where the feature extractor is not pre-trained on ImageNet

**Strengths And Weaknesses:**

Before giving my review I would like to point out that my area of expertise lies more in Continual Learning rather than meta-learning. Therefore, if some of my remarks are a consequence of misunderstanding on my side, I am open to discussion and would gladly admit any mistake that might be a consequence of my ignorance of the meta-learning domain.

# Strenghts
- This paper is well-written, easy to read.
- The experiments are for the most part comprehensive and it appears that a lot of care has been given to the experimental section.
- The introduced setup is valuable.
- The conclusions, if true, are interesting. **However, I am currently not convinced by the experiments**; see weaknesses.

# Weaknesses
While I am convinced that a lot of care has been put into this paper, I have identified a lot of weaknesses and limitations which I believe to be critical.

## 1. Feature extractor trained on IN-1k does not make sense.

The main weakness comes from the fact that the feature extractor is trained on ImageNet-1k. Therefore, when training GEOM, this model has access to the knowledge from Imagenet-1k obtained during pre-training, **and** the knowledge from Meta-album. In contrast, GEOM-IN, has accessed to *only* the knowledge contained in ImageNet-1k. This is completely ignored when considering the performances of GEOM, when the authors simply conclude that this must be because of the organization of the dataset. On top of this, the authors ensured that Meta-Album datasets that were similar to Imagenet-1k were removed. Therefore, it is guaranteed that the evaluation datasets (which I believe are drawn from Meta-album) might be dissimilar to ImageNet-1k, while Figure 13 in the Appendix shows that similarities among datasets from Meta Album do exist. My point is that on top of the pre-training issue, the evaluation datasets might also be more similar to training datasets in the case of GEOM rather than GEOM-IN.

## 2. Insufficient experiments to back up current claims

The authors make broad claims considering that a small organized set of datasets should be more valuable for generalization than a large unorganized dataset. This is the main takeaway of the paper; which I unfortunately have a lot of issues agreeing with. Such a claim contradicts the current state of research, as correctly pointed out in the paper. In that sense, a lot of experiments are required to prove such a point, and for me the only experiments that confirm this claim are the ones comparing GEOM and GEOM-M. However, this has been conducted in only one setup (Meta Album mini). Other experiments, as I have pointed out above are not giving insights. In addition, the results observed between GEOM and GEOM-M are poorly justified technically. The authors assume that such results are "expected" by pointing out similarities to human learning, but I find this argument not very convincing. In my opinion, this can be a consequence of GEOM-M showing much more difficult tasks, and I am wondering if the model has been training until convergence for each task. I have trouble understanding how many epochs have been used for each task in these setups. Similarly, I would like to see such results on the larger dataset (Extended), and why not do the same procedure with ImageNet? Or other classification datasets. Such a wide claim must be substantially backed up.

## 3. Overclaim / Misconception regarding human analogy

As discussed above, I am not convinced by the human analogy argument. As Yann Lecun's lecture discusses, **humans are actually introduced to extremely large amounts of data, in an unsupervised manner.** Way more than LLM. See [Yann LeCun's lecture](https://www.youtube.com/watch?v=MiqLoAZFRSE) at 18:25 minutes. I apologize I could not find such information in the related paper. I am inclined to agree that after acquiring a world model, the analogy makes sense, but this comes **after** training on a vast, unstructured amount of data. In any case, let us say that this analogy makes sense, it is **not a justification of the paper's finding by itself**. It can be a fun explanation, but it is **not a demonstration**. Especially, in section 6.3.1, the authors have counter-intuitive results and have no problem justifying it by the fact that this is actually different from human learning.

## 4. Overclaim regarding relation between number of classes and performances

In section 5.3, the paper explains that "the most significant performance improvements arise from increasing the number of classes, which enriches task variability and broadens the model's capacity for generalization". In that case, why is GEOM-M performing worse than GEOM? If my understanding is correct, each task should have a way larger amount of classes than GEOM's tasks.

## 5. Unfair comparison with the static approach

In section 6.1, this paper claims that assuming prior knowledge of the datasets is unrealistic, and therefore the static strategy (using the same number of epochs for each dataset) is adopted. However, it seems to me that being able to compute a dataset similarity matrix as well as ordering them in a curriculum manner should be way more unlikely than knowing the number of epochs to use beforehand. In any case, as long as you loop over the dataset you know its size so you can adapt the number of epochs consequently. The problem is that I am convinced that the number of epochs being small plays a very important factor in the conclusions of the paper. Training a few epochs will give an advantage to easier datasets, hence the conclusion that large-scale datasets will lead to poorer generalization.

## 6. Unrealistic ordering in unsupervised and ignored data leakage issue

In section 7, it is assumed that the dataset is unlabeled, however, it can still be ordered. This again seems more unrealistic than the number of epochs being adaptative. The authors additionally consider data leakage from Imagenet-1k, but GEOM-U benefits from a pre-trained model on ImageNet-1k so its overall training dataset is way larger than CAMeLU and suffers from the same data leakage.

## Update after reading other reviews: excessive references to Vettoruzzo et al.

After reading other reviewers' reviews, I equally want to point out that the number of references to Vettoruzzo et al. is inadequate and should be reconsidered. 5 papers from Vettoruzzo et al. are cited while the second most cited author has 2 cited papers.

---

> ### Author Response · Authors · 2025-04-20
> **Response to reviewer fgjE (part 1)**
>
> We thank the reviewer for the time spent reading and analysing our work, and for joining a constructive discussion. We also would like to mention that throughout our rebuttal, as well as already done in our manuscript, we will refer to the definition of tasks seen in Section 3.2, which is borrowed from the meta-learning literature [1][2], rather than continual learning. To also target a broader audience, in these first lines of rebuttal, we would like to provide a tangible example of what a task is for meta-learning. A task can be regarded as a way to organize a batch of the dataloader, which always consists of N classes and K examples per class, namely support set, and Q queries, referred to as query set, to be classified to perform backpropagation (the Q queries are assumed to belong to the N classes as well). The true labels are eventually permuted, therefore, independently of the actual class that is loaded in the batch, the labels will be in the range [0, N-1]. In meta-learning, a task does not correspond to a dataset that is trained until convergence, rather it could be imagined as a batch of the whole dataloader with the aforementioned peculiarities.
>
> ### About pre-training on ImageNet-1k
> To overcome the limitations of the image feature extractor based on ImageNet-1k features, and referring to the fact that GEOM-IN may have less external knowledge (i.e., other than ImageNet-1k) than GEOM, we conducted experiments to prove the effectiveness of our methodology: we added a GEOM vs GEOM-IN comparison with CLIP feature extractor (Tab.11 in Appendix A.6). The results are mostly aligned with those in Fig.4 (and Tab.10), indicating that the real difference for GEOM-IN occurs whenever it has access, in the form of data leakage, to the test domains during the training phase.
>
> ### About similarities among Meta-Album datasets and "removing" datasets
> We apologize for the misunderstanding, however, we did not remove any Meta-Album dataset that overlapped with ImageNet-1k. Fig.2 and Fig.4 simply highlight the fact that some overlap does exist, and this justifies the higher performance of GEOM-IN over GEOM for the datasets that include class overlap. In addition, similarities in Fig.14 [it was Fig.13, as the reviewer mentioned, before the addition of a new image in the Appendix] are computed as distance from embeddings that are extracted from a pre-trained ResNet50, the same used in the model: this means that our feature extractor might find similar two datasets that even belong to the different domains because it locates them in a certain region of its hyperspace. In essence, since it is pre-trained and frozen, it may not be able to extract valuable features from certain classes/datasets/domains. This, however, was used as the starting point for our analysis.
>
> ### About task difficulty, convergence, GEOM vs GEOM-M
> Our intention in the paper is not to suggest that multiple small, domain-specific datasets *universally outperform* large, uncurated corpora, but rather that they *can match or even exceed* their performance in some cases, while offering additional practical advantages, such as modularity, easier maintenance, better adaptability, and ethical manageability, as stated in Section 5 and in the Conclusions, and it aligns with emerging findings in the LLM literature [3,4].
> The key evidence for this comes from the comparison between GEOM and GEOM-M (Section 5.1), and between GEOM-U and CAMeLU (Section 7). However, all the other sections in the paper support this claim, by demonstrating how keeping the dataset separated, instead of merging them into a single large pool, enables desirable capabilities that are otherwise difficult to implement. For instance, it allows to add more datasets as they become available (Section 5.2), it allows to train the model sequentially (Section 6), and to build a curriculum (Section 6.3). All of this is not possible by considering a single, large datasets. To reflect this perspective more clearly, we have added explicit clarifications in the Introduction.
> Regarding task difficulty, we would like to mention that it is not necessarily true that GEOM-M faces harder tasks. In fact, tasks in GEOM-M involves coarse-grained classification (e.g., classifying dog vs. airplane vs. microscopy image vs. bacteria vs. plant), which are often easier than the fine-grained classification tasks in GEOM (e.g., classifying different bird species, or classify different human protein). We have now clarified this point in Section 5.1.
> Concerning the generality of our findings: while Meta-Album Mini was the primary benchmark due to its domain diversity, dataset splits and (almost) balanced nature, we report results with both Meta-Album Micro and Extended (Tables 3 and 8), and we compare it with GEOM-IN which is trained on ImageNet-1k. Our approach is in principle applicable to any setting where well-defined task boundaries can be established, and we see this as a natural direction for future work.

---

> ### Author Response · Authors · 2025-04-20
> **Response to reviewer fgjE (part 2)**
>
> ### About human-learning analogies
>
> We thank the reviewer for engaging the discussion about human learning and we would like to make clear that we recognize the complications that may arise when trying to explore the human brain and the wide range of the obscure mechanisms that drive its behaviour and learning paths. We decided to mention human learning as the inspiration that drove this work and its link with humans came from some evidence borrowed from the neuroscience field. First of all, it has been long proved that the developmental trajectories in children are very slow [5], and they usually come together with failures in the recognition of known objects in scenes that are visually crowded [6]. Under this assumption, it is true that children are immersed in cluttered environments, but it is also true that this clutter does not directly drive the learning. Indeed, analyses of head-mounted camera footage show that infants' visual experiences follow a right-skewed distribution, with repeated exposure to a small number of individual faces [9] and objects [10]. Children first develop visual invariances that support the recognition of a limited number of objects under varying viewing conditions. The same applies in the learning to read, where the brain first learns to recognize and differentiate lines, curves, and strokes, before being able to generalize to single characters and entire words [11]. Moreover, this initial exposure to a limited set of objects is fundamental for later visual development; individuals who lacked early visual experience show a permanent deficit in configural face processing as adults [12]. The other analogy with human learning derives from the fact that, after this first experience up to the second year of life, children start experiencing a burst development in object recognition, up to the point that a single instance is enough to generalize to new samples [13]: this is somehow similar in the learning trend of the transformer encoder, as also shown in Fig.18. In humans, this sharp transition is tightly bound with the development of language abilities [7][8][14]. To make this evidence more clear, we strengthened this discussion in the introduction while removing or improving conclusions that may have sounded rushed.
>
> In light of the above consideration, we also clarified the curriculum results in Sect.6.3.1. These results do not undermine the analogy with human learning, but rather highlight a limitation of current machine learning models, compared to humans. By considering the learning trend of a meta-learned transformer encoder, as shown in Fig.18, transition between the first and the second phase of learning (identified as the memorization and learning in Fig.18) is relevant for the following ability of the model to generalize. Starting with very simple datasets (where simple, in this case, refers to the similarity to a pre-trained configuration, but gives no information about the similarity among datasets, as this is studied in Sect.6.3.2) may saturate the knowledge of the model when it reaches the generalization phase (Fig.18) and can no longer increase.
>
> ### About number of tasks GEOM vs GEOM-M
>
> The purpose of section 5.3 is to analyze whether the model performance is affected by the number of classes or the number of images in the dataset. As clarified in the definition of tasks above, each task always consists of 5 (N) classes in meta-learning and NK+Q examples. GEOM and GEOM-M share the same classes and images, the difference is that the latter breaks the boundaries among datasets. This means that, for each episode to sample the N-way K-shot task from, GEOM-M would have access to all the Meta-Album classes, while GEOM can only sample the N-way K-shot task from one dataset at each episode. However, the structure is shared and the error is backpropagated after each task.

---

> ### Author Response · Authors · 2025-04-20
> **Response to reviewer fgjE (part 3)**
>
> ### About the prior knowledge on dataset features
>
> We agree with the reviewer that some prior knowledge or assumptions have to be made both in the case of defining in advance the number of epochs or computing the similarity matrix. We would like to point out that the metric that we have used to compute that is based on OTDD [15], which requires the datasets to have similar sizes. This means that we can decide the number of samples to extract the information from and apply it to all of our datasets. Furthermore, we start from a pre-trained, frozen, ResNet50 feature extractor that has learned on ImageNet-1k: of course, this is intended to provide a starting point from some pre-acquired knowledge to define one or more curricula. We acknowledge the reviewer’s concerns regarding the number of epochs assigned to each dataset, as this choice was the result of a thorough and carefully considered evaluation process. For this reason, in Fig.7 we show that there is not so much difference in the evaluation accuracy when we compare a model trained with a static (fixed number of epochs) or proportional approach. The only relevant difference is in the OCR domain (last three datasets) because its datasets are way larger than all the others. This is a limitation of Meta-Album, whereas the ideal case would have similar sizes for all the datasets (please, notice that the size, in any case, is never the very same). Additional studies and future work may address these questions when this is applied to an unbalanced collection.
>
> ### About unsupervised
> We think there is a misunderstanding, as we are not ordering the datasets in the unsupervised scenario. In Section 7 we aim to demonstrate that GEOM can be extended to the unsupervised (offline) setting, where no order is required. The training is performed following the LOO scenario described in Section 5, with the addition that tasks need to be automatically constructed from an unlabeled dataset. This is different from the results in Section 5, which instead represents the supervised (offline) scenario, and also from Section 6, which investigates the (supervised) sequential scenario.
> We want to clarify that both in CAMeLU and GEOM-U only the feature extractor is pre-trained on ImageNet-1k while the transformer encoder is learned during training. The only difference between GEOM-U and CAMeLU is the structure and nature of the training dataset. While the transformer encoder in GEOM-U is trained on tasks constructed from 9 domains in Meta-Album (LOO setting), where each task is sampled from a single dataset, CAMeLU is trained with tasks generated from ImageNet-1k. Both approaches are then evaluated on tasks sampled from the left-out domain in Meta-Album. Therefore, if the evaluation dataset (such as the ones in LA) is very similar to ImageNet, there is some data leakage in the transformer parameters, which is the only learnable part of our model. The feature extractor is only used to obtain meaningful features, and it can be easily replaced with a different model pre-trained on a dataset different from ImageNet-1k, as demonstrated in Tab.11 in Appendix A.6.
>
> ### Critical: GEOM-M and ImageNet-1k sequential and merged
> There is a problem here: the use of Meta-Album does allow us to build sequential orders and curricula. GEOM-M and ImageNet-1k are large scale dataset to consider as a single pool, so defining an order from a *whole* is non-trivial and this is one of the practical reason to use and analyse Meta-Album with GEOM.
>
> ### Critical: problem of convergence
> We are wondering if the reviewer refers to the convergence, especially on tasks, from the continual-learning point of view, where a task is, at the end of the day, a dataset seen at a certain iteration. We adopt the terminology borrowed from the meta-learning literature, where a task is defined, concretely, as an episode consisting of K examples for each of the N classes in a batch. In light of this hypothesis, GEOM and GEOM-M contain the same number of classes and samples per class, and the number of tasks that we sample is the same (500 tasks for each epoch). To further clarify the convergence, we add Fig.18 to the manuscript, which shows the relative validation accuracy on unseen classes over the epochs (after each training epoch, the model is validated). For each epoch, the validation accuracy is computed as the average among 50 tasks for each of the 30 datasets available (1500 tasks).

---

> ### Author Response · Authors · 2025-04-20
> **Response to reviewer fgjE (part 4)**
>
> ### Critical: concern about pre-trained feature extractor
>
> We understand the concern about the use of a pre-trained feature extractor. Our model design was firstly inspired by CAML [16], and previous work had also proven how the use of known projected features sped up the convergence (GPICL [18]). The role of the transformer in the learning to generalize, instead, was visually described in GPICL [18] and CAMeLU [17]. The use of a clean feature extractor would naturally increase the training time, and may require adaptations to the transformer size, other than the fact that it would not be appropriate, for a rebuttal, to retrain everything from scratch. Nevertheless, another reviewer raised an interesting point about the outcome of the model in the curriculum case, with a non-pre-trained feature extractor. For this reason, we decided to run the experiment with the very same architecture and let the weights of ResNet50 be learned together with the transformer. Although with an expected decrease in the absolute value of the validation accuracy, the relative performance between E2H and H2E, compared with the baseline, remains aligned with the original findings. These are shown in Fig.13 and Tab.7 in the Appendix. We hope that this experiment provides an additional solid confirmation of our approach in the study of curricula.
>
> ### Critical: feature extractor not pre-trained on ImageNet-1k
> If our understanding is correct, the doubt expressed by the reviewer was related to the fact that GEOM-IN has access to the pre-trained, frozen knowledge of ImageNet-1k via the feature extractor and the same applies to the training set. Instead, GEOM still has access to the pre-trained knowledge of the same feature extractor, but benefits from another dataset (Meta-Album) for the training phase. In light of this assumption, we would like to first recall that Meta-Album (Mini) is a much smaller collection than ImageNet-1k and, additionally, we train GEOM by intentionally excluding each test domain: this alone would put us in a harder position. That being said, to accomplish the reviewer’s request and prove that the GEOM-IN can be trained over a dataset that differs from the pre-acquired knowledge of the feature extractor, we ran new experiments that compare GEOM vs GEOM-IN (Tab.11 in Appendix A.6) by replacing ResNet50 pre-trained on ImageNet with CLIP [19]. Results are mostly aligned with Tab.10.
>
> ### About excessive referencing
> We have now refined the bibliography, thanks for pointing this out

---

> ### Author Response · Authors · 2025-04-20
> **List of references for fgjE**
>
> [1] Vettoruzzo, Anna, et al. "Advances and challenges in meta-learning: A technical review." IEEE transactions on pattern analysis and machine intelligence 46.7 (2024): 4763-4779.
>
>
> [2] Hospedales, Timothy, et al. "Meta-learning in neural networks: A survey." IEEE transactions on pattern analysis and machine intelligence 44.9 (2021): 5149-5169.
>
> [3] Sang Michael Xie, Hieu Pham, Xuanyi Dong, Nan Du, Hanxiao Liu, Yifeng Lu, Percy S Liang, Quoc V Le, Tengyu Ma, and Adams Wei Yu. Doremi: Optimizing data mixtures speeds up language model pretraining. Advances in Neural Information Processing Systems, 36, 2024.
>
>
> [4] Aakanksha Chowdhery, Sharan Narang, Jacob Devlin, Maarten Bosma, Gaurav Mishra, Adam Roberts, Paul Barham, Hyung Won Chung, Charles Sutton, Sebastian Gehrmann, et al. Palm: Scaling language modeling with pathways. Journal of Machine Learning Research, 24(240):1–113, 2023.
>
>
> [5] MacNamara, J. (1982) Names for Things: A study of Child Language. Cambridge, MA: MIT Press.
>
>
> [6] Farzin, F., Rivera, S. M., and Whitney, D. (2010). Spatial resolution of conscious visual perception in infants. Psychol. Sci. 21, 1502–1509. doi: 10.1177/0956797610382787
>
>
> [7] Rosch, E., Mervis, C. B., Gray, W. D., Johnson, D. M., and Boyes-Braem, P. (1976). Basic objects in natural categories. Cogn. Psychol. 8, 382–439. doi: 10.1016/0010-0285(76)90013-X
>
>
> [8] Samuelson, L. K., and Smith, L. B. (2005). They call it like they see it: spontaneous naming and attention to shape. Dev. Sci. 8, 182–198. doi: 10.1111/j.1467-7687.2005.00405.x
>
>
> [9] Jayaraman, S., Fausey, C. M., and Smith, L. B. (2017). Why are faces denser in the visual experiences of younger than older infants? Dev. Psychol. 53, 38–49. doi: 10.1037/dev0000230
>
>
> [10] Clerkin, E. M., Hart, E., Rehg, J. M., Yu, C., and Smith, L. B. (2017). Real-world visual statistics and infants’ first-learned object names. Philos. Trans. R. Soc. B 372:20160055. doi: 10.1098/rstb.2016.0055
>
>
> [11] Stanislas Dehaene (2009). Reading in the Brain: The New Science of How We Read. New York: Penguin + 388 pp. ISBN: 978-0-14-311805-3
>
>
> [12] Maurer, D., Mondloch, C. J., and Lewis, T. L. (2007). Sleeper effects. Dev. Sci. 10, 40–47. doi: 10.1111/j.1467-7687.2007.00562.x
>
>
> [13] Smith, L. B. (2003). Learning to recognize objects. Psychol. Sci. 14, 244–250. doi: 10.1111/1467-9280.03439
>
>
> [14] Smith, L. B., Jones, S. S., Landau, B., Gershkoff-Stowe, L., and Samuelson, L. (2002). Object name learning provides on-the-job training for attention. Psychol. Sci. 13, 13–19. doi: 10.1111/1467-9280.00403
>
>
> [15] David Alvarez-Melis and Nicolo Fusi. Geometric dataset distances via optimal transport. Advances in Neural Information Processing Systems, 33:21428–21439, 2020.
>
>
> [16] Fifty, Christopher, et al. "Context-Aware Meta-Learning." The Twelfth International Conference on Learning Representations.
>
>
> [17] Anna Vettoruzzo, Lorenzo Braccaioli, Joaquin Vanschoren, and Marlena Nowaczyk. Unsupervised meta-learning via in-context learning. In International Conference on Learning Representations, 2025.
>
>
> [18] Kirsch, Louis, et al. "General-purpose in-context learning by meta-learning transformers." arXiv preprint arXiv:2212.04458 (2022).
>
>
> [19] Radford, A., Kim, J., Hallacy, C., Ramesh, A., Goh, G., Agarwal, S., Sastry, G., Askell, A., Mishkin, P., Clark, J., & others (2021). Learning transferable visual models from natural language supervision. In International conference on machine learning (pp. 8748–8763).

---

> > ### Comment · Reviewer_fgjE · 2025-05-14
> >
> > First of all, I would like to sincerly thank the authors for clarifying misunderstandings from my side and for the inclusion of additional experiments which I believe complement the manuscript. It seems to me that the majority of my concerns were addressed or were mostly a consequence of a my lack of expertise in the area of Meta Learning.
> >
> > That being said, I remain skeptical about this paper. Following the point raised by reviewer TSog, regarding the experimental setting: E. Issues with the experimental settings and the way conclusions are drawn from them. I understand now that GEOM is introduced to harder tasks than GEOM-M. However, I have to say this paragraph from reviewer TSog makes sense to me:
> >
> > > Also, it is expectable that GEOM is better than GEOM-M when tested on Meta-Album. Consider two datasets, one with images of birds, another with images of textures (based on Figure 1). Learning to distinguish images of birds from those of textures (as in GEOM-M) is arguably much easier than distinguishing images of birds from those of other birds (GEOM). Given that the latter is the type of tasks with which the system is ultimately evaluated at test time, we’d expect systems trained in the latter setting to be better as it’s consistent. So I do not find this experimental setting well configured to support the authors’ main claim about the benefit of “multiple smaller, domain-specific datasets”.
> >
> > And, unfortunately, I do not see much argument regarding this point. Despite it being obvious that the authors have invested a lot of time and effort in this work, and that the updated version contains a lot of improvements, the claims made in this work seem to be overstatements given the current experimental setup.

---

> > > ### Author Response · Authors · 2025-05-14
> > > **Follow-up to Reviewer fgjE**
> > >
> > > We sincerely thank the reviewer for their constructive feedback, for acknowledging the clarifications and additional experiments, and for engaging the discussion. We would like to respectfully clarify a few points that we believe address this concern more thoroughly.
> > >
> > > The reviewer and reviewer TSog suggest that GEOM should naturally outperform GEOM-M (which mixes all domains into a single large dataset), because GEOM involves training on more fine-grained distinctions (e.g., birds vs. birds), which is allegedly more aligned with the test-time tasks. However, this assumption does not fully hold in the cross-domain setting we consider.
> > >
> > > In our experimental setup, meta-test tasks in GEOM come from **novel domains** and datasets that were entirely unseen during meta-training. For example, a model may be trained on various natural image datasets (like birds, textures, or cars) and then evaluated on completely unrelated domains like protein images from human cells. Therefore, when the target domain is entirely unknown (GEOM), there is no strong evidence to assume that training on fine-grained distinctions within domains (e.g., birds vs. birds) should generalize better than training on mixed-domain tasks (e.g., birds vs. textures, as in GEOM-M). **This point undermines the assumption that the tasks in GEOM are more consistent** with the evaluation tasks. In fact, the variety and diversity in GEOM-M (which constructs training tasks by sampling across merged domains) is traditionally believed to be more beneficial in this case, as it is supposed to encourage the meta-learner to acquire generalizable inductive biases that are less domain-specific and more adaptable to novel, unrelated tasks from a new, unseen domain.
> > >
> > > Thus, while it is true that distinguishing birds from other birds may be more challenging than distinguishing birds from textures, this increased difficulty does not necessarily imply better generalization to entirely different tasks, such as classifying subcellular protein patterns — particularly when these target tasks belong to fundamentally different visual and semantic domains. In fact, one could argue that this domain shift constitutes the greater challenge. Our design, which preserves domain boundaries during training, does offer a real advantage when transferring knowledge to unseen domains, thereby reinforcing our claim.
> > >
> > > The comment about "consistency" between training and evaluation tasks is valid in **same-domain** meta-learning scenarios. However, in cross-domain meta-learning, the main challenge is generalization to test tasks from new domains different from the training domains. Therefore, the reviewer's notion that "the type of tasks at test time are more similar to GEOM's training tasks" is not fully accurate. At test time, we evaluate the model on entirely new domains with no class overlap. Thus, "consistency" here cannot be equated with "domain similarity". Instead, what matters is the model's ability to generalize across domain shifts — and our results demonstrate that GEOM often outperforms or matches GEOM-M in this regard.
> > >
> > > As explained in the paper, one of GEOM's strengths lies in its real-world practicality: small, domain-specific datasets are easier to curate, update, and deploy independently. This design facilitates curriculum learning, sequential updates, and interpretability — all of which are far more challenging under the merged-task setting of GEOM-M. Furthermore, in practical deployment scenarios, new tasks often arise from completely novel domains with little or no overlap with the training data. Our training strategy with GEOM is tailored to this reality.
> > >
> > > In summary, we appreciate the reviewer's perspective but respectfully believe that the experimental design supports our main claim. The use of multiple small, domain-specific datasets in GEOM does not aim to make training tasks harder for the sake of it — rather, it encourages the model to develop robust cross-domain generalization strategies.
> > >
> > > We hope this response clarifies our position and alleviates the concern.

---

> ### Comment · Reviewer_fgjE · 2025-05-15
>
> Thank you for these additional elements. I understand the author's standpoint that these are news domains, and so the concern raised by TSog should not hold.
>
> However, here is where I believe I disagree:
>
> > Therefore, when the target domain is entirely unknown (GEOM), there is no strong evidence to assume that training on fine-grained distinctions within domains (e.g., birds vs. birds) should generalize better than training on mixed-domain tasks (e.g., birds vs. textures, as in GEOM-M). This point undermines the assumption that the tasks in GEOM are more consistent with the evaluation tasks. In fact, the variety and diversity in GEOM-M (which constructs training tasks by sampling across merged domains) is traditionally believed to be more beneficial in this case, as it is supposed to encourage the meta-learner to acquire generalizable inductive biases that are less domain-specific and more adaptable to novel, unrelated tasks from a new, unseen domain.
>
> The domain might be new, but the task may require similar fine-grained analysis. I would argue that learning to distinguish between bird species (birds vs birds) is more similar to (dogs vs dogs) than (cells vs cats).
> Roughly speaking:
>
> |Method|Train|Test|
> |:-|:-|:-|
> |GEOM|fine grained (birds vs birds)|fine grained (cells vs cells)|
> |GEOM-M|Coarse (birds vs cats) |fine grained (cells vs cells)|
>
> So, when the authors make the following claim:
>
> > Therefore, when the target domain is entirely unknown (GEOM), there is no strong evidence to assume that training on fine-grained distinctions within domains (e.g., birds vs. birds) should generalize better than training on mixed-domain tasks (e.g., birds vs. textures, as in GEOM-M).
>
> It seems to me that it is an overclaim to talk about generalization; the evaluation is quite specific.

---

> ### Author Response · Authors · 2025-05-15
> **Reply to Reviewer fgjE**
>
> We thank the reviewer for their attention to our comments. We would like to add that GEOM-M tasks come from a wider task distribution, as they do not only include tasks from different domain and datasets, but they also include tasks *picked* from the same datasets or domain; in other words: $\text{supp} (p(\tau_{geom})) \subset \text{supp}(p(\tau_{geom\text{-}m}))$, with *supp* that indicates the support of the probability distribution. We should expect GEOM-M to be more general than GEOM. However, the increased difficulty in the whole set of training tasks in GEOM leads the model to comparable performance.
>
> To provide additional evidence and push the boundary further, we are evaluating the following condition: we train GEOM-M with all the training classes of all the 30 Meta-Album datasets. Then, we take the worst-performing model in GEOM (OCR) test the performance of these two models on $5000$ GEOM-M-like tasks (i.e., the test tasks are built in the same way as GEOM-IN training tasks) to prove that the model is not biased towards the training task distribution.
> * GEOM       accuracy: 98.94 $\pm$ 0.09,
> * GEOM-M  accuracy: 98.59 $\pm$ 0.09.
>
> (Average computed over three runs and three different seeds, OCR domain is entirely excluded from the test set).
>
> Once again, the results are comparable. With the due time limit, we can easily set and provide results for this experiment as well. We are already setting the experiments up: if the reviewer is willing to know more about this and thinks this experiment is worth to validate our initial assumption, we may refer to the AE to be given the time to conclude the tests.

---

> > ### Author Response · Authors · 2025-05-16
> > **Experiments on GEOM-M-like test tasks**
> >
> > We added the previously described test. Results are available in Tab.13 of the updated manuscript. Despite GEOM has never experienced tasks that contain classes from mixed domains, the advantage of GEOM-M is still negligible (approximately 1% for each domain) and, surprisingly, the overall highest result is achieved by GEOM when trained excluding OCR, even when compared with the Oracle (GEOM-M trained on all the datasets of all the domains, as our latest comment to this thread). Since this domain is much larger than all the others, it may introduce a significant bias in the final performance.

---

> > > ### Comment · Reviewer_fgjE · 2025-05-17
> > > **Thank you for your reply and the additional experiments.**
> > >
> > > Thank you for the additional experiments. This is interesting.
> > > I have updated my review accordingly.

---

### Review · Reviewer_TSog · 2025-04-14

**Summary Of Contributions:**

This work conducts experiments investigating various settings of multi-task in-context learning using the Meta-Album dataset as the main dataset.

**Audience:**

Yes

**Claims And Evidence:**

No

**Requested Changes:**

Suggestions are included inline in the weakness section above.

**Strengths And Weaknesses:**

### Strengths

- Use of Meta-Album for in-context learning experiments seems appropriate

### Weaknesses

There are several major issues.

**A. The title of this work “Generalization over Memorization” (which is also the name of the proposed setting, GEOM) misrepresents the actual contribution of this work.**

With the exception of Sec 5.1, all the discussions and experiments conducted in this work are not about memorization vs generalization. In fact, even 5.1. is not really about “memorization” (in the conventional sense of memorizing training tasks, e.g., [1]) but rather about the impact of the similarity and fuzzy overlaps between training and test data on the test performance. While the experiment of Sec 5.1 itself is an interesting empirical result, it does not address the fundamental question of memorization vs. ability to learn new tasks in-context, when compared to existing work that precisely and rigorously studies the behavior of in-context learners in terms of memorization of training tasks vs. abilities to learn unseen tasks in-context [2, 3]. In particular, Raventós et al. NeurIPS 2023 [1] demonstrate how the task diversity in the pretraining has an influence on the behavior of in-context learners to follow the training task distribution (i.e., memorization-like behavior) or to generalize to novel tasks (generalization behavior); effectively demonstrating when in-context learning could achieve generalization over memorization.
Therefore, I find the title and the method name misleading. I recommend the authors to revise this corresponding narrative.

Also, in light of existing works, the following sentence in the abstract is a bold claim that is not appropriate for the content of this work:

> we propose a **paradigm shift**: training on multiple smaller, domain-specific datasets to improve generalization.


[1] Zhang, Bengio, Hardt, Recht,Vinyals. Understanding deep learning requires rethinking generalization. ICLR 2017

[2] Raventós, Paul, Chen, Ganguli. Pretraining task diversity and the emergence of non-Bayesian in-context learning for regression. NeurIPS 2023

[3] Panwar, Ahuja, Goyal. In-Context Learning through the Bayesian Prism. ICLR 2024


**B. There are overclaims about the relation to human learning**

There are numerous statements claiming that the proposed setting aligns with human learning. However, I find many of these connections to be vague, and in several instances, even inaccurate. For example:

> (Sec 5.2 Impact of number of datasets) These configurations allow us to examine the relationship between generalization and knowledge accumulation, drawing parallels with the progressive learning process observed in humans

Here, three different models are trained on three datasets of different (increasing large) sizes; I do not see anything so special about this setting that captures aspects of human learning.

> (Sec 6.3) By doing so, we could verify the impact of the datasets’ order on the model performance and draw analogies with the human learning process.

Here too, I do not see anything compelling to be highlighted in relation to human learning. Curriculum may be inspired from human learning, but what matters is the actual curriculum; I do not see any aspects of these image classification experiments that reflect human-like gradual learning (or are the authors claiming that children first learn to classify birds before learning to classify flowers?), especially given that the hard-to-easy (H2E) curriculum outperforms human-motivated easy-to-hard one:

> (Sec 6.3.1) the best performance is achieved with the H2E configuration, as demonstrated by the average performance gain in the last column of Fig. 9. While this is counterintuitive compared to the human learning process

Also:

> (Sec 5.1) our results suggest that focusing on one domain at a time enhances cross-domain generalization, akin to human learning, which prioritizes mastering individual tasks before integrating broader knowledge

I do not see any compelling analogy to human learning here either. Compare for example to the exact example used by the authors “[children] start by recognizing individual letters, which, over time, builds the foundation for reading complex sentences”.

In my view, all these claims on connections to human learning are too vague to be acceptable in any scientific paper. My suggestion is to eliminate the emphasis on connections to human learning (I do not see anything that would be lost in terms of contributions of this work by doing so).


**C. Misconception regarding the distinction between meta-learning and in-context learning.**

There is a fatal misconception regarding the distinction between meta-learning and in-context learning. First of all,

> Recent work has sought to draw connections between ICL and meta-learning (Min et al., 2022a; Kirsch et al., 2022; Fifty et al., 2024)


This sentence is inaccurate. The related work section of the GPT3 paper by Brown et al. NeuriPS 2020 already discusses the in-context learning ability of GPT3 as meta-learning by citing the work by Hochreiter et al. ICANN 2001 [4] where in-context learning without weight updates was already achieved.

[4] Hochreiter, Younger, Conwell. Learning to Learn Using Gradient Descent. ICANN 2001

> (Sec 3.1) although both ICL and meta-learning utilize demonstration contexts for task adaptation, they differ fundamentally in their approach

This sentence, and the entire Table 1 is representative of the misconception. In-context learning is a type of meta-learning; they are not meant to be compared on the same level; it is as if one was listing the differences between gradient-based learning algorithms (a class of methods) and Adam (an instantiation).

It should also be noted that while the name, in-context learning, is a rather recent terminology, the concept of in-context learning, i.e., meta-learning based on a sequence learning neural network, has been studies by much earlier works, see e.g., in-context regression by Hochreiter et al. ICANN 2001 [4] and in-context classification by Santoro et al, ICML 2016 [5] (there are many more papers). I also invite the authors to more closely look at recent papers studying in-context learning, including [6, 7, 8, 9], to realize that the distinction proposed in Table 1 is not appropriate.

[6] Garg, Tsipras, Liang, Valiant. What Can Transformers Learn In-Context? A Case Study of Simple Function Classes. NeurIPS 2022.

[7] von Oswald, Niklasson, Randazzo, Sacramento, Mordvintsev, Zhmoginov, Vladymyrov. Transformers learn in-context by gradient descent. ICML 2023

[8] Bai, Chen, Wang, Xiong, Mei. Transformers as Statisticians: Provable In-Context Learning with In-Context Algorithm Selection. NeurIPS 2023.

[9] von Oswald et al. Uncovering mesa-optimization algorithms in Transformers. ICLR 2024 Workshop on Mathematical and Empirical Understanding of Foundation Models.

**D. Issue with the model name**

Related to A, I am not convinced that the name GEOM is appropriate. As far as I can see, the main property that differentiates GEOM from prior in-context learners is their application to a specific dataset, Meta Album. In my view, this does not deserve a new name.

In particular, for GEOM-U:

> The main difference between GEOM-U and CAMeLU is the training data. While GEOM-U is trained with tasks sampled from the Meta-Album datasets across diverse domains, CAMeLU is trained on ImageNet-1k (Deng et al., 2009), a large-scale benchmark that represents a wide data distribution.

So do the authors plan to give a new name to CAMeLU every time it is applied to a different dataset? This does not sound appropriate.

On top of this, as also stated in Issue A, “generation over memorization” is not the central contribution of this work (see [2] and [3]). I therefore recommend eliminating this name GEOM; it’s simply in-context learning applied to Meta Album, under various settings (LOO, sequential, unsupervised).

**E. Issues with the experimental settings and the way conclusions are drawn from them**

**Issues with 5.1**

Most crucially, at the design-level, I cannot agree that the GEOM-M vs. GEOM comparison is the right setting to conclude on the claimed benefit of “training on multiple smaller, domain-specific datasets to foster generalization” (claimed in the introduction). GEOM-M is a very specific way to construct meta-training example sequences for in-context image classifiers by mixing images from various datasets (while GEOM follows the classic way to construct meta-training sequences by preserving the domain boundaries). This is not at all representative of the “vast, uncurated, unstructured” training of LLMs that the authors are criticizing. To me, there is no clear analog of GEOM-M for LLMs; it could be something like taking two separate articles/paragraphs, breaking them into sentences and mixing them, which does not correspond to the reality of LLM training.

Therefore, in my view, one of the core claims of this work “(2) demonstrates that this approach fosters improved generalization compared to training on a single, large-scale dataset”
is not well supported by these experiments. **To me, this issue alone is enough to vote for rejection, following the soundness criterion of TMLR.**

Also, it is expectable that GEOM is better than GEOM-M when tested on Meta-Album. Consider two datasets, one with images of birds, another with images of textures (based on Figure 1). Learning to distinguish images of birds from those of textures (as in GEOM-M) is arguably much easier than distinguishing images of birds from those of other birds (GEOM). Given that the latter is the type of tasks with which the system is ultimately evaluated at test time, we’d expect systems trained in the latter setting to be better as it’s consistent. So I do not find this experimental setting well configured to support the authors’ main claim about the benefit of “multiple smaller, domain-specific datasets”.

> When comparing GEOM to GEOM-IN, GEOM achieves superior or comparable performance in datasets with minimal class overlap between Meta-Album and ImageNet-1k.

This is factually not correct. On Small animals 44292 and 44306, Vehicle 44289, and Manufacturing 44288, GEOM-IN largely outperforms GEOM. While I acknowledge that the authors discuss the case of “Manufacturing”, I do not understand the useful takeaway message from here, apart from the fact that models trained under data somewhat similar to test data tend to perform well on the corresponding data, but that’s not always the case. I cannot see the clear takeaway messages that are generally useful beyond this specific data combination of Meta Album and ImageNet-1k.

> while LLMs rely on vast amounts of unfiltered data to achieve generalization, a child can generalize with fewer, more meaningful examples. This suggests that the key to generalization may depend less on the sheer volume of data and more on its quality and the sequence in which it is presented.

This is an interesting hypothesis, but I do not find any compelling experimental settings and results in this work to validate this.

**Regarding 5.3:**

Regarding the main claim:

> These results suggest that the most significant performance improvements arise from increasing the number of classes, which enriches task variability and broadens the model’s capacity for generalization.

I see three flaws in the current experimental settings to draw this conclusion. First, (1) in the current setting, there is no disentangling between adding new classes and adding more images; as adding new classes also adds more images. Therefore, we do not really know whether more total images or more classes ultimately helped. Secondly, (2) the comparison is done using two different starting points: the impact of more classes is evaluated by moving from Micro to Mini, while the impact of more images is evaluated from Mini to Extended. This is not a fair comparison between these two modifications; given that Micro is a toy setting, it is naturally much easier to improve over the toy Micro setting than over the Mini setting.
Finally, (3) the conclusion is expected to change depending on the actual amounts of classes and images added: adding only one extra class may be less useful than adding 100 more images per class, while adding 10 classes is likely more useful than adding 1 extra image per class. All these experimental settings need to be refined to draw reliable conclusions.

**Regarding Sec 6:**

First of all, this setting relates to continual meta-learning [10, 11].

[10] Yap, Ritter, Barber. Addressing Catastrophic Forgetting in Few-Shot Problems. ICML 2021

[11] Son, Lee, Kim. When Meta-Learning Meets Online and Continual Learning: A Survey. IEEE Transactions on Pattern Analysis and Machine Intelligence (TPAMI) 2024

**Regarding Sec 6.1:**

Here again, we cannot conclude from this experiment whether “proportional” is helpful simply because it provides more epochs than “static” or it is effectively proportional to the data size. In order to verify this, it is crucial to conduct an additional experiment for static but with the max number of epochs used by proportional. This is a crucial question since naively increasing the number of epochs might introduce forgetting.

> While the BWT is commonly used to measure forgetting in traditional continual learning setups, where tasks typically belong to the same domain, in our case, the dataset Db belongs to domains that are different from the domain of Da.

I do not think this statement is accurate. See e.g., the CTRL benchmark of [12].

[12] Veniat, Denoyer, Ranzato. Efficient Continual Learning with Modular Networks and Task-Driven Priors. ICLR 2021

Also very crucially: while I definitely find it interesting to conduct experiments using Meta-Album, I cannot evaluate how good the reported results are, because there is no prior work on this setting. Sec 6.1 results demonstrate that with the exception of the OCR part (the last three sets), the sequential learning performance almost matches that of the offline baseline, and Sec 6.2 effectively shows that there is no forgetting even for “static”.  This is unusual from the common findings in the continual meta-learning benchmarks [10], and it is completely unclear from these experiments why this is the case; is this because of the Meta-Album? Or is there something special to in-context learning? I would suggest running the same experiment on a more classic dataset for which some solid baselines from prior work is known–for example the Triathlon dataset from [10] or CTRL from [12]--and compare to existing baselines from prior work. Otherwise, I cannot evaluate the true value of these results.

**Regarding 6.3.2**

> it appears that the best-performing curriculum across all datasets is E2E. This aligns with our expectations, as gradual changes in the observed data encourage the model to accumulate knowledge over time, avoid forgetting, and build upon prior learning incrementally.

Unlike the TL-based strategy of Sec 6.3.1., OT-based method does not specify the first task to be used. The authors used the easiest task (44304) for both E2E and H2H, but this choice is arbitrary; why not use the hardest task 44308 to initialize H2H? At least this experiment seems necessary to conclude on the superiority of E2E.

Also, in my view, the final comment about “alignment to the expectation” has limited scientific value, since the opposite was found in 6.3.1, making this comment completely arbitrary (when it works it works, when it doesn’t it doesn’t).

**Regarding Sec 7**

While I fully agree that the unsupervised setting is important for low-resource scenarios, I am not convinced that the experiments presented here demonstrate broadly useful takeaways. In particular, it is unclear what scientific question motivates comparing GEOM-U against CAMeLU. While GEOM-U does not use labels for training, it does use Meta-Album datasets for training; which may potentially be more “in-domain” than CAMeLU trained on ImageNet-1k since both systems are tested on Meta Album test sets. So the takeaway isn’t clear to me. It seems more natural to evaluate the limit of unsupervised setting by comparing GEOM-U to GEOM (but even with that, I do not see how this fits to the main idea of this work).


**Overall**, while I appreciate the experimental breadth the authors aimed to cover in this work, I find that each section lacks rigor and depth in their experiments to strongly support the claims. As a result, many of the claims (as I listed above) remain speculative. Considering the TMLR’s main acceptance criteria which are correctness and soundness, I find more solid experimentations to be necessary.


**F. Excessive references to Vettoruzzo et al.**

The manuscript contains a high number of citations to Vettoruzzo et al. (15 times in the main text).  I find this excessive citation inappropriate, given that the authors miss many other important references. I recommend the authors critically assess whether each citation is essential and consider incorporating more diverse sources to provide a more balanced view.

In particular, I find the following citations problematic:

> Meta-learning approaches are explicitly trained to adapt to new tasks by leveraging previously learned knowledge and information extracted from a small set of data (context) (Vettoruzzo et al., 2024b).

This is a generic statement about meta-learning. Referring to seminal works seem important, e.g., Schmidhuber (already cited below) and Finn et al. ICML 2017 [13] at least.

[13] Finn, Abbeel, Levine. Model-Agnostic Meta-Learning for Fast Adaptation of Deep Networks. ICML 2017

While additional citations to surveys are welcome, Vettoruzzo is not the only option here either. Consider also Hospedales et al. TPAMI 2022 [14] and Son et al. TPAMI 2024 [11].

[14] Hospedales, Antoniou, Micaelli, Storkey. Meta-learning in neural networks: A survey. IEEE Transactions on Pattern Analysis and Machine Intelligence. 2022

[11] Son, Lee, Kim. When Meta-Learning Meets Online and Continual Learning: A Survey. IEEE Transactions on Pattern Analysis and Machine Intelligence (TPAMI) 2024

> This capability has been compared with meta-learning, which explicitly trains models to adapt to new tasks by leveraging prior knowledge (Schmidhuber, 1987; Vettoruzzo et al., 2024b).

Same here.

> [Page 4, Regarding sequential learning] … and meta-learning-based approaches (Vettoruzzo et al., 2024c; Gupta et al., 2020; Javed & White, 2019)

Similarly, a highlight on Vettoruzzo et al. overlooks other prior works on meta-learning-based sequential learning, such as Irie et al. ICML 2022 [15], and Lee et al. NeurIPS 2023 [16].

[15] Irie, Schlag, Csordás, Schmidhuber. A Modern Self-Referential Weight Matrix That Learns to Modify Itself. ICML 2022.

[16] Lee, Son, Kim. Recasting Continual Learning as Sequence Modeling. NeurIPS 2023

> (Page 10) This improvement can be attributed to the increased variability of training tasks, which has been shown to promote robust learning (Vettoruzzo et al., 2025; Chan et al., 2022; Singh et al., 2023).

Again, while Vettoruzzo is prominently highlighted, it misses other important related works by Raventós et al. NeurIPS 2023 [2] and Panwar et al. ICLR 2024 [3].

[2] Raventós, Paul, Chen, Ganguli. Pretraining task diversity and the emergence of non-Bayesian in-context learning for regression. NeurIPS 2023

[3] Panwar, Ahuja, Goyal. In-Context Learning through the Bayesian Prism. ICLR 2024

**Other/Minor issues:**

**Confusing presentation:** I found it very confusing that the first version of GEOM presented in Sec. 3.2. is the leaving-one-out (LOO) variation. This corresponds to neither the sequential version, GEOM-S, nor the unsupervised version GEOM-U, which are briefly introduced in the introduction. Relatedly, the emphasis is on sequential learning from the start of the paper, but Sec 5 suddenly introduces the “supervised learning (offline)” setting. I have two comments (1) It is completely unclear to me how this setting fits to the entire narrative of the paper presented until this point. (2) calling this supervised (offline) learning is confusing (given that sequential learning is also supervised learning); I’d suggest to call this case **non-sequential** instead.

**Figure 4**: I highly recommend the authors to increase the resolution; they are currently not visually pleasant as they are pixelated

> However, existing training paradigms for in-context learners rely on vast, unstructured datasets, which are costly and challenging to collect.

I do not fully agree with this. The strength of the current language modeling setting is precisely the fact that the data is easy to collect (abundant on the internet) and it requires no human labeling (therefore free).

**Typo**: Page 18 “Hard-to-Hasy”

---

> ### Author Response · Authors · 2025-04-20
> **Response to reviewer TSog (part 1)**
>
> ### About the paper title
> When highlighting the generalization abilities of transformers, we cover different aspects: as the reviewer noted, (1) in the LOO case the model is exposed to a test set of domains that were never seen in the training phase. The sequential case (2) reflects different settings, as all the datasets are available during training, but test classes remain unknown until the in-context learning phase. Although we did not provide a mathematical and rigorous study about the actual distribution shift between training and test, nor the threshold for which task diversity raises in-context learning abilities, as the excellent work in [1], we provide an empirical study on a broader and real-case scenario where, *in fact*, there is an undeniable distribution shift between the training and test sets, either when we exclude an entire domain, or if we know the domain, but some classes are left unknown until inference time. The paper further extends these experiments to the challenging unsupervised scenario (3), where neither the classes, nor the labels are known.
> From a similar perspective, the reviewer does not consider our use of the term *generalization* conventional. We recognize that we differ from [2], but our case study is characterized by some differences, indeed. First of all, when evaluating performance of a generic classification model, the traditional split involves *shared* classes between the train and test set, but different samples. In this context, the generalization performance is evaluated on how good (provided a desired accuracy metric) a model is to associate new samples to the learned classes (i.e., concepts). Considered through this lens, our evaluation of the generalization abilities fundamentally misaligns with the traditional meaning of generalization, but this is intended to be. This is mainly due to the obvious fact that we do not have overlap between train and test classes (in any of the three aforementioned and studied cases), and, above all, to the fact that we perform label permutation for every episode. As an example, as we operate in a few-shot inference setting, we could cite [3], where the generalization is studied across datasets. Not only that, but the concept of generalization after a first memorization phase is witnessed for the grokking phenomenon [4] (not studied here), where a *family* of problems is first memorized before being able to reach perfect and stable generalization. This work, in particular, refers to a generalization phase that suddenly and abruptly develops after a long period of overfitting on the training set, which should sound counterintuitive from a *classical* perspective, since a model that has overfitted on the training set should eventually perform poorly on the test set.
> By extending this discussion to LLMs, we would also like to point out that the GPT-2 paper [5] has a whole chapter about “generalization vs memorization”, when referring to the ability of the model to perform unseen tasks after the pre-training phase. The study of the contamination between training and test set is still an open problem for the NLP community, indeed, in order to be able to recognize if ICL can actually generalize or if it simply memorizes training examples [6][7].
> Finally, we wish to clarify that we are not proposing a redefinition or reinterpretation of the term *generalization*. However, we believe that a certain degree of flexibility is warranted, given the shift in conditions between standard supervised classification and few-shot classification settings.
>
> ### About the model name
> The reason why we define another name for CAMeLU is that we wanted to highlight the difference in the results when the training is performed on a collection of datasets (properties are extensively explained in the text) rather than a large scale dataset (original CAMeLU), in the unsupervised case. In the same way, the supervised section defines two different names for the same model when trained on ImageNet-1k, i.e., GEOM-IN, and when trained on Meta-Album, i.e., GEOM. We find the names GEOM-U vs CAMeLU being more clear than GEOM-MA-U vs GEOM-IN-U to indicate the unsupervised training on the two collections. Modifications to the training set modify the final outcome of a model [35][36], this is extensively studied throughout the paper and since the focus is indeed on *how these modifications reflect the outcome*, we use different names when talking about different settings, although the model architecture is the same.

---

> ### Author Response · Authors · 2025-04-20
> **Response to reviewer TSog (part 2)**
>
> ### About human learning
>
> We appreciate the suggestion and we have now edited some parts related to the analogy with human learning that may have resulted in overly simplistic and premature conclusions. However, we still believe that keeping some analogies can provide a better understanding of the inspirational parallels that guided some of our design choices and the structuring of experiments. In the following we will better explain the analogies we kept:
> In Section 5.2 the goal was to investigate how performance scales as the model is exposed to an increasing number of datasets from the same set of domains, i.e., First (10 datasets), Second (20 datasets), and Third (30 datasets). Despite being a simplification, we believe it echoes an important dynamic observed in humans: progressive knowledge accumulation within familiar domains. For example, a child may initially recognize only a toy car (a limited example from the domain of vehicles) but gradually encounters a wider variety of instances—real cars, buses, trucks—which expand their conceptual understanding of the same domain. Similarly, in our setup, the model is exposed to a growing diversity of data within each domain, which helps it build a more generalizable representation of visual concepts.
> This analogy is meant to illustrate a learning trajectory that favors structured, incremental exposure and it is consistent with the rationale behind meta-learning [8, 9] and curriculum learning [10, 11].
> Furthermore, in the same section, the statement that “focusing on one domain at a time enhances cross-domain generalization” draws on a broader pedagogical insight: humans often benefit from mastering distinct concepts before integrating them into a more general understanding [12][13].
> When referring to the curriculum in Sect.6.3.1, consider the learning trend of a meta-learned transformer encoder, as shown in Fig.18. The transition between the first and the second phase of learning (identified as the memorization and learning in Fig.18) may be relevant for the following ability of the model to generalize. Starting with very simple datasets may lead the model to saturate its knowledge upon reaching the generalization phase (Fig. 18), beyond which no further improvement can be achieved. This drew its inspiration from neuroscience [14], which states that children deprived of visual capabilities in the very first months of their life - when faces and humans represent the center of their perception - are likely to exhibit deficits in visual perception as adults. Apart from this, we wanted to practically highlight the learning trend as also shown in Fig.10, where the E2H curriculum decreases its trend. Moreover, we would like to stress on the fact that the TL based curriculum computes a difficulty score from the same pre-trained features, but it cannot include the complexity that intercurs among datasets; this latter part is indeed studied in the following section of the paper.
> We recognize that some conclusions may have been drawn in a too simplistic way, yet we still assert that the inspiration from this paper came from the peculiar trend that a transformer-like architecture exhibits (already shown in [15, 16]) and the way humans are introduced to the world. For the former, we have discussed the three phases more in depth in Fig.18, where the first phase of memorization is followed by an abrupt change in the learning trend, where the model recognizes the task to be solved (N-way K-shot) and eventually acquires the ability to generalize to other tasks that share the same structure but on different, unseen classes and even unknown domains. When referring to humans, it has been long proved that the developmental trajectories in children are very slow [17]. Analyses of footage from head-mounted cameras reveal that infants' visual experiences are characterized by a right-skewed distribution, with frequent exposure concentrated on a limited set of individual faces [18] and objects [19]. During early development, children initially acquire visual invariances that enable them to recognize a small number of objects across varying viewing conditions. A similar process occurs in the context of learning to read: the brain first learn to distinguish lines, curves, and strokes before progressing to the recognition of individual characters and, ultimately, complete words [20]. This might resemble the memorization phase of the meta-learned transformer. The other analogy with human learning arises from the observation that, following the initial phase up to the second year of life, children undergo a rapid developmental surge in object recognition, eventually reaching a stage where a single instance is sufficient to generalize to new samples [21]. A comparable pattern can be observed in the learning trajectory of the transformer encoder, as illustrated in Fig. 18. In humans, this sharp developmental shift is closely linked to the emergence of language abilities [22][23][24].

---

> ### Author Response · Authors · 2025-04-20
> **Response to reviewer TSog (part 3)**
>
> ### Meta-learning vs in-context learning
> We are aware of the definition given by [25], which we often cite throughout our text. We also recognize that the reviewer is aware (but we still stress this to favour a broader audience) that the same paper specifies, in a footnote, that *they* «use the term "meta-learning" to capture the inner-loop / outer-loop structure of the general method, and the term “in context-learning” to refer to the inner loop of meta-learning.»
> That being said, it is important to clarify which was our intention, as the paper mentions and explains both terms to avoid the confusion that this current discussion is raising. When we talk about in-context learning, we specifically refer to the concept introduced by [25], as already mentioned in the manuscript. It may be true that previous work may have referred to learning *with*/*through* the context (a lot of few-shot learning methods make use of context to generalize their knowledge to new, unseen classes, here just a few and well-known methods [28][29], or even an attention based method developed prior to the in-context learning definition [27]). It is equally true that GPT-2 [5] also provides insights into what would finally be formally defined *in-context learning*. In light of this, our discussion specifies that its original definition [25] reflects the fact that a transformer-like architecture that has undergone a large-scale, autoregressive pre-training acquires abilities that allow it to solve few-shot tasks at inference time with the sole feeding of input-output pairs and no weights update [26]. Furthermore, as also witnessed in [26], the training task for LLMs differs from the ICL task, as we also evidence in our paper. Instead, when talking about meta-learning, we refer to a generic framework that proposes to solve new, unseen tasks, with few (or no) adaptation steps. For meta-learning, the structure of the training tasks is the same as the one resembled in the test tasks. That being said, after the rise of in-context learning [25], other studies [30][31] explicitly applied a meta-learning pipeline on top of a pre-trained LLM to enrich the few-shot performance of ICL. Under the same assumption, CAML [32], CAMeLU [33], and GPICL [34] meta-train a transformer to reach in-context, few-shot inference capabilities. The base of our model [32] inherits from the meta-learning field concepts like episodic training, label permutation, and fast adaptation (black box models). On the other hand, we borrow from the in-context learning literature the structure of the sequence (input-output pairs), the use of a transformer, which also characterizes its learning trend, and its built-in generalization abilities that allow the model to work on domains that were not seen, a problem that would require well-designed interventions to a staple model to work properly. Overall, our initial discussion in the paper was indeed due to specify what characterizes meta-learning, on the one hand, and in-context learning, on the other hand. We are still convinced that talking about in-context learning prior to the advent of GPT-3 would result in a more confusing setting, thus we specified that when mentioning that term, we were properly referring to anything that has been discussed after 2020 [25].
>
> ### Regarding 5.3
> (1) Both Chan [35] and Singh [36] had already proved that adding new classes to the training set increased the ICL capabilities, so this is considered to be true. Apart from that, since we explicitly mentioned that we wanted a fair and balanced comparison, due to the fact that Extended is the only Meta-Album size that do not include the OCR domain, we removed OCR from the other two sizes: this means that Mini gets 1577 training classes, compared to the 1597 in Extended. The difference between the two is negligible if compared to the number of images that Extended is granted with respect to Mini (roughly 8.5 times more, please refer to Appendix A.2 for more details). (2) We never defined Micro as a toy setting, nor the Meta-Album [43] paper did. Micro may be too small for the scale of our architecture and the extension of our experiments. However, smaller models may still benefit from an expanded, diverse, and balanced collection of 30 datasets, or even used for test-only purposes. (3) The mentioned configurations would be too simplistic, given the overall number of classes and the different nature of the datasets in Meta-Album, and the outcome would suffer from a too small difference to draw appropriate conclusions from.

---

> ### Author Response · Authors · 2025-04-20
> **Response to reviewer TSog (part 4)**
>
> ### Issue with experimental settings and conclusions
> Although few-shot learning has usually preserved boundaries of domains, nothing has ever prevented the use of a large-scale dataset that might include classes from very different domains (ImageNet-1k, for the quickest example). Indeed, just to cite recent work in meta-learning and few-shot learning [37][32][33], the use of a large-scale dataset to expand to the cross domain scenario during the inference phase is possible. Moreover, BECLR [38] even proves that this is possible even starting with a small dataset: please, notice that [38] makes use of *mini*ImageNet during the training phase, which contains 64 training classes that, however, belong to different domains. In light of this, we find the reviewer’s claim about the fact that boundaries among domains are preserved misleading. Other than that, all the previously cited papers use a training dataset that does **not** preserve domain boundaries (ImageNet-1k). In the same way, GEOM-M breaks the boundaries among the domains to emulate a single, large-scale dataset.
>
> Throughout the paper, in the introduction, in the conclusion and even in the Broader Impact Statement section, we highlight how the use of small-scale datasets introduces other benefits, such as more balance across domains, better interpretability (labels can be checked easier, for instance), replaceability either for quality upgrade, privacy or ethical concerns. Even the results in GEOM, GEOM-IN, GEOM-M do not show an always-better performance of GEOM over all the others, and we do not hide this. Despite the fact that the reviewer makes a strong statement about our attempt to break the soundness criterion of this journal, we would like to point out that even studies on LLMs have tried to pay attention to the balance of the different domains that were included into the training set, or the way they are used [39][40], which were already present in the original manuscript. Not only that: indeed, GPT-2 [5] prefers to build a web crawler from scratch (WebText), instead of using the already available CommonCrawl to perfect the quality of the training data. In addition to that, they even computed the amount of overlap between WebText and the test datasets, a problem that is still open among the NLP community [41][42]. Later, GPT-3 [25] even studied the weights in the training mix, and even recognized that a bug in their algorithm may have altered the overlap that was initially computed. Therefore, we strongly disagree with the reviewer’s claim that this study does not provide any interest to the NLP community. In light of this, the reviewer even objects that when pre-trained on a large-scale (or merged, in our case) dataset, it would be unfair to test the model on domain-based dataset, but this is a standard approach even for the ICL in the NLP field [5][25].
>
> When referring to Manufacturing, the key takeaway is that the images are very simple (i.e., characterized by patterns, so low level features). Every type of image can show low-level features, so the higher the number of images (independently of the class or the doman), the higher the knowledge that can be extracted. Meta-Album Mini contains much less images than Meta-Album Extended, indeed, when evaluated on Manufacturing, it shows a significant performance boost. About the other two domains: vehicles are present in ImageNet-1k, and small, microscopic animals are likely to be. As we explained, we checked the labels and *concept*, but CLIP embeddings struggle with latin nouns, as we specified. Given that our metric can still be improved, we had no other shared way to verify if Small Animals classes were present in ImageNet-1k.

---

> ### Author Response · Authors · 2025-04-20
> **Response to reviewer TSog (part 5)**
>
> ### Regarding Sec.6.1
>
> To be fair, we specified that we used the same number of overall epochs for both proportional and static settings (600, where each epoch runs over 500 tasks). The number of epochs varies per dataset. In our manuscript, we had already recognized that proportional was likely to be the most appropriate way of addressing sequential learning, but it also assumes more prior knowledge about the incoming datasets and the study of the perfect convergence point is still an open challenge. Since Meta-Album is a mostly balanced collection, apart from OCR, as we proved, our final choice fell on static for reasons that we already explained in 6.1. Adding even more epochs to a small dataset, may introduce overfitting. This is just a conjecture, as at the time of writing we cannot prove it, even for limited resources. We reserve to explore this as future work, as already outlined in the conclusion section.
> Furthermore, despite the disagreement with our assertion about the fact that BWT [44] is commonly used in Continual Learning, the paper cited by the reviewer [45] includes the concept of (absence of) forgetting [44] in a section named “Desirable Properties of CL models And Metrics”.
> As already explained in our paper, the reason for positive BWT values are not directly due to the intrinsic nature of Meta-Album: in the first phases, indeed, some negative BWT values do exist, as reported in Fig.8 and Tab.4. However, the more the number of classes and diversity increases, the more the generalization abilities on new, unseen tasks improve, and eventually make the model acquire positive BWT values. At this point, it should be even clearer what was our intention, even in the first answer of this rebuttal, when discussing generalization: we studied values of forgetting (in practical terms of backward transfer) as our hypothesis was based on the fact that adding new classes and enriching the whole training knowledge consequently increased the ICL abilities. We want to stress on the fact that, in any case, the model always solves new few-shot tasks, so it does not require to retain the memory of the seen classes, because they would not be seen during the inference phase. However, we can state that ICL can solve tasks for a domain that has been explored at the beginning of the training time, and this is because it has learned to generalize, rather than learning to solve a specific, dataset-related task.
> When requiring additional experiments with Triathlon and CTRL benchmarks, we would like to point out that Meta-Album is a much more complicated setting than those, other than DTD is already in the Manufacturing domain and CIFAR-FS shares a lot of similarities with other Meta-Album datasets, and that our intention is not comparing with existing continual learning approach, rather prove the generalization abilities over time for ICL, especially because we are not evaluating on the same classes that we have seen during the training phase.
>
> ### Regarding 6.3.2
> We sincerely apologize for the resulting lower clarity in the drawn conclusion of the curriculum section. TL-based curricula and OT-based curricula fundamentally differ for the fact that the former orders dataset complexity according to a unique point (the accuracy of a fine-tuned classifier that shares the same frozen backbone). Instead, OT-based curricula computes complexity among datasets. The latter is indeed more appropriate to be defined as a human-like curriculum, as the dataset encountered at time *t* is the closest to the one at time *t-1*. The same cannot be said for the TL-based curricula because two datasets may share the same difficulty with respect to a pre-trained feature extractor, but this gives no information on how dataset *t-1* can contribute to the knowledge of dataset *t*. Under this assumption, we also hope to make clearer the decision of starting with the same dataset in the OT-based curricula: we fixed the same starting point for both of them, then built the rest of the pipeline according to the *difficulty of the previous* dataset. For TL-based curricula, instead, it was more logical to go forward and backward. This is also the reason why we stated that the outcome was *predictable*. We have already explained this in our paper, but we have tried to make it more clear now.

---

> ### Author Response · Authors · 2025-04-20
> **Response to reviewer TSog (part 6)**
>
> ### Excessive references to Vettoruzzo et al.
> We apologize for the excessive references, and we thank the reviewer for pointing this out. We refined the bibliography by adding more recent and relevant references and removing less relevant ones.
>
> ### CAMeLU
> CAMeLU vs GEOM-U is motivated by extending the LOO in the unsupervised settings, to confirm how a collection of several small-scale datasets may provide comparable performance to a single large scale dataset. The test domain in GEOM-U is always excluded from the training phase, therefore, the model has zero notion of Vehicles (for instance) when evaluating vehicles, apart from the pre-acquired knowledge of the feature extractor, which is shared among all the models.
>
>
> ### Figure 4:
> We thank the reviewer for the suggestion, we have now refined Fig.4
>
> ### About the cost of collection
> The collection is not entirely free, despite the abundance of text over the entire web. Apart from filtering copyrighted material (non-trivial), several approaches have tried to refine the way the data is collected [39][40][25][5]. Other than that, analysing the whole data is still challenging due to its intrinsic vast nature.

---

> ### Author Response · Authors · 2025-04-20
> **References for reviewer TSog (part 1)**
>
> [1] Raventós, Paul, et al. Pretraining task diversity and the emergence of non-Bayesian in-context learning for regression. NeurIPS 2023
>
>
> [2] Zhang, et al.. Understanding deep learning requires rethinking generalization. ICLR 2017
>
>
> [3] Triantafillou, E., et al. (2021). Learning a Universal Template for Few-shot Dataset Generalization. In Proceedings of the 38th International Conference on Machine Learning (pp. 10424–10433). PMLR.
>
>
> [4] Power, A., et al. (2022). Grokking: Generalization beyond overfitting on small algorithmic datasets. arXiv preprint arXiv:2201.02177.
>
>
> [5] Radford, A., et al. 2019). Language models are unsupervised multitask learners. OpenAI blog, 1(8), 9.
>
>
> [6] Balloccu, S., et al. (2024). Leak, Cheat, Repeat: Data Contamination and Evaluation Malpractices in Closed-Source LLMs. In Proceedings of the 18th Conference of the European Chapter of the Association for Computational Linguistics (Volume 1: Long Papers) (pp. 67–93).
>
>
> [7] Sainz, O., et al. (2023). NLP Evaluation in trouble: On the Need to Measure LLM Data Contamination for each Benchmark. In Findings of the Association for Computational Linguistics: EMNLP 2023 (pp. 10776–10787).
>
>
> [8] Vettoruzzo, Anna, et al. "Advances and challenges in meta-learning: A technical review." IEEE transactions on pattern analysis and machine intelligence 46.7 (2024): 4763-4779.
>
>
> [9] Hospedales, Timothy, et al. "Meta-learning in neural networks: A survey." IEEE transactions on pattern analysis and machine intelligence 44.9 (2021): 5149-5169.
>
>
> [10] Sheybani, Saber, et al. "Curriculum learning with infant egocentric videos." Advances in Neural Information Processing Systems 36 (2023): 54199-54212.
>
>
> [11] Bengio, Yoshua, et al. "Curriculum learning." Proceedings of the 26th annual international conference on machine learning. 2009.
>
>
> [12] Feldman, Jacob. "The simplicity principle in human concept learning." Current directions in psychological science 12.6 (2003): 227-232.
>
>
> [13] Gagne, R. M. (1970). The conditions of learning (2nd ed.). Holt, Rinehart & Winston.
>
>
> [14] Maurer, D., et al. (2007). Sleeper effects. Dev. Sci. 10, 40–47. doi: 10.1111/j.1467-7687.2007.00562.x
>
>
> [15] Kirsch, Louis, et al. "General-purpose in-context learning by meta-learning transformers." arXiv preprint arXiv:2212.04458 (2022).
>
>
> [16] Anna Vettoruzzo, et al. Unsupervised meta-learning via in-context learning. In International Conference on Learning Representations, 2025.
>
>
> [17] MacNamara, J. (1982) Names for Things: A study of Child Language. Cambridge, MA: MIT Press.
>
>
> [18] Jayaraman, S., et al. (2017). Why are faces denser in the visual experiences of younger than older infants? Dev. Psychol. 53, 38–49. doi: 10.1037/dev0000230
>
>
> [19] Clerkin, E. M., et al. (2017). Real-world visual statistics and infants’ first-learned object names. Philos. Trans. R. Soc. B 372:20160055. doi: 10.1098/rstb.2016.0055
>
>
> [20] Stanislas Dehaene (2009). Reading in the Brain: The New Science of How We Read. New York: Penguin + 388 pp. ISBN: 978-0-14-311805-3
>
>
> [21] Smith, L. B. (2003). Learning to recognize objects. Psychol. Sci. 14, 244–250. doi: 10.1111/1467-9280.03439
>
>
> [22] Rosch, E., et al. (1976). Basic objects in natural categories. Cogn. Psychol. 8, 382–439. doi: 10.1016/0010-0285(76)90013-X
>
>
> [23] Samuelson, L. K., and Smith, L. B. (2005). They call it like they see it: spontaneous naming and attention to shape. Dev. Sci. 8, 182–198. doi: 10.1111/j.1467-7687.2005.00405.x
>
>
> [24] Smith, L. B., et al. (2002). Object name learning provides on-the-job training for attention. Psychol. Sci. 13, 13–19. doi: 10.1111/1467-9280.00403
>
>
> [25] Tom B. Brown, et al. (2020). Language Models are Few-Shot Learners.
>
>
> [26] Raventós, Paul, Chen, Ganguli. Pretraining task diversity and the emergence of non-Bayesian in-context learning for regression. NeurIPS 2023
>
>
> [27] Mishra, N., et al. (2018). A Simple Neural Attentive Meta-Learner. In International Conference on Learning Representations.
>
>
> [28] Snell, J., et al. (2017). Prototypical Networks for Few-shot Learning. Advances in Neural Information Processing Systems, 30.
>
>
> [29] Vinyals, O., et al. (2016). Matching networks for one shot learning. Advances in neural information processing systems, 29.
>
>
> [30] Chen, Y., et al. (2022). Meta-learning via Language Model In-context Tuning. In Proceedings of the 60th Annual Meeting of the Association for Computational Linguistics (Volume 1: Long Papers) (pp. 719–730). Association for Computational Linguistics.
>
>
> [31] Sewon Min, et al. (2022). MetaICL: Learning to Learn In Context.
>
>
> [32] Fifty, Christopher, et al. "Context-Aware Meta-Learning." The Twelfth International Conference on Learning Representations.

---

> ### Author Response · Authors · 2025-04-20
> **References for reviewer TSog (part 2)**
>
> [33] Vettoruzzo, Anna, et al. "Unsupervised Meta-Learning via In-Context Learning."  The Twelfth International Conference on Learning Representations. 2025.
>
>
> [34]  Kirsch, Louis, et al. "General-purpose in-context learning by meta-learning transformers." arXiv preprint arXiv:2212.04458 (2022).
>
>
> [35] Stephanie Chan, et al. 2022. Data distributional properties drive emergent in-context learning in transformers. In NeurIPS.
>
>
> [36] Singh, Aaditya, et al. "The transient nature of emergent in-context learning in transformers." Advances in Neural Information Processing Systems 36 (2023): 27801-27819.
>
>
> [37] Huiwon Jang, Hankook Lee, & Jinwoo Shin (2023). Unsupervised Meta-learning via Few-shot Pseudo-supervised Contrastive Learning. In The Eleventh International Conference on Learning Representations.
>
>
> [38] Poulakakis-Daktylidis, S., & Jamali-Rad, H. (2024). BECLR: Batch Enhanced Contrastive Few-Shot Learning. In The Twelfth International Conference on Learning Representations.
>
>
> [39] Aakanksha Chowdhery, et al. Palm: Scaling language modeling with pathways. Journal of Machine Learning Research, 24(240):1–113, 2023.
>
>
> [40] Sang Michael Xie, et al. Doremi: Optimizing data mixtures speeds up language model pretraining. Advances in Neural Information Processing Systems, 36, 2024.
>
>
> [41] Balloccu, S., et al. (2024). Leak, Cheat, Repeat: Data Contamination and Evaluation Malpractices in Closed-Source LLMs. In Proceedings of the 18th Conference of the European Chapter of the Association for Computational Linguistics (Volume 1: Long Papers) (pp. 67–93).
>
>
> [42] Sainz, O., et al. (2023). NLP Evaluation in trouble: On the Need to Measure LLM Data Contamination for each Benchmark. In Findings of the Association for Computational Linguistics: EMNLP 2023 (pp. 10776–10787).
>
>
> [43] Ullah, I., et al.. (2022). Meta-Album: Multi-domain Meta-Dataset for Few-Shot Image Classification. In Thirty-sixth Conference on Neural Information Processing Systems Datasets and Benchmarks Track.
>
>
> [44] Lopez-Paz, D. et al. Gradient episodic memory for continual learning. Advances in neural information processing systems, 30, 2017
>
>
> [45] Veniat et al. Efficient Continual Learning with Modular Networks and Task-Driven Priors. ICLR 2021

---

### Author Response · Authors · 2025-04-20
**General response to the reviewers**

We thank the reviewers for the time dedicated to our work. Here’s a list of modifications applied after their suggestions:

**Major**:
* Refined the human-like discussion: as we drew our inspiration from that, we wanted to provide stronger evidence of our intentions. This resulted in two modifications: (1) we added further explanations (with corresponding citations) in the introduction and (2) we lightened some conclusions that sounded rushed.
* We provided three more sets of experiments to enrich the evidence of our claims, following the reviewers’ suggestions: (1) a TL-based curriculum trained from scratch and (2) a LOO setting with a different feature extractor, and finally (3) a 5way-1shot scenario.


**Minor**:
* Refined Fig.4
* Added Fig.18
* Refined bibliography

---

### Decision · Action_Editor_2kgc · 2025-06-16

**Recommendation:** Reject

**Audience:**

Yes

**Audience Explanation:**

Despite the aforementioned concerns, the paper touches on a highly relevant and timely topic: how to better train in-context learners in a way that reflects natural learning principles, such as task compositionality and curriculum design. Researchers in continual learning, data-efficient training, or curriculum design may find partial value in the methodology, insights, and discussions, provided that the results are further substantiated.

**Claims And Evidence:**

No

**Claims Explanation:**

The central claim of the paper, i.e., training in-context learners on multiple small, domain-specific datasets improves generalization over large-scale unstructured data, is not adequately substantiated by the presented experiments. While the paper is thorough in coverage and presents numerous evaluations across supervised, sequential, and unsupervised scenarios, the experimental design often fails to align closely with the core claims.
- While the paper claims that organizing training data into multiple small, domain-specific datasets (GEOM) helps generalization, comparisons like GEOM vs. GEOM-M or GEOM vs. GEOM-IN are confounded.
   - Reviewer TSog pointed out that GEOM-M contains some mixture but retains much of the dataset organization, so the ablation doesn't effectively disentangle the role of structure.
   - Reviewer 7FoW further noted that the exact design of GEOM-M is underspecified, making it difficult to interpret what changes actually affect performance.
- The reliance on ImageNet-pretrained feature extractors in most settings introduces significant confounds (e.g., potential information leakage).
    - As Reviewer TSog and 7FoW noted, Meta-Album test datasets overlap with ImageNet classes, meaning the representation may already encode much of the test information, potentially inflating results. This undermines the claim that generalization stems from dataset structure or learning dynamics, rather than from pretrained knowledge leaking in.
- The authors suggest that curriculum learning (increasing task diversity or difficulty over time) improves ICL, yet no controlled curriculum learning experiments are done.
    - Reviewer 7FoW noted that the benefits attributed to curriculum effects may instead be due to better data alignment, not the order or structure of training.
- Furthermore, while the authors draw heavy analogies to human learning, these remain qualitative and are not convincingly validated by the results.

**Resubmission Of Major Revision:**

The authors may consider submitting a major revision at a later time.